# Gapped Phases in (2+1)d with Non-Invertible Symmetries: Part I

Lakshya Bhardwaj[1], Daniel Pajer[1], Sakura Schäfer-Nameki[1],
Apoorv Tiwari[2], Alison Warman[1], Jingxiang Wu[1]

[1] *Mathematical Institute, University of Oxford,*
*Andrew Wiles Building, Woodstock Road, Oxford, OX2 6GG, UK*
[2] *Niels Bohr International Academy, Niels Bohr Institute, University of Copenhagen,*
*Blegdamsvej 17, DK-2100, Copenhagen, Denmark*

We use the Symmetry Topological Field Theory (SymTFT) to study and classify gapped phases in (2+1)d for a class of categorical symmetries, referred to as being of bosonic type. The SymTFTs for these symmetries are given by twisted and untwisted (3+1)d Dijkgraaf-Witten (DW) theories for finite groups $G$. A finite set of boundary conditions (BCs) of these DW theories is well-known: these simply involve imposing Dirichlet and Neumann conditions on the (3+1)d gauge fields. We refer to these as minimal BCs. The key new observation here is that for each DW theory, there exists an infinite number of other BCs, that we call non-minimal BCs. These non-minimal BCs are all obtained by a 'theta construction', which involves stacking the Dirichlet BC with 3d TFTs having $G$ 0-form symmetry, and gauging the diagonal $G$ symmetry. On the one hand, using the non-minimal BCs as symmetry BCs gives rise to an infinite number of non-invertible symmetries having the same SymTFT, while on the other hand, using the non-minimal BCs as physical BCs in the sandwich construction gives rise to an infinite number of (2+1)d gapped phases for each such non-invertible symmetry. Our analysis is thoroughly exemplified for $G = \mathbb{Z}_2$ and more generally any finite abelian group, for which the resulting non-invertible symmetries and their gapped phases already reveal an immensely rich structure.

# 1 Introduction and Summary

One of the main challenges in quantum field theory is understanding the landscape of low-energy phases and phase transitions between them. While significant progress has been made in (1+1) dimensions, the (2+1) dimensional case remains a challenge. For example, bosonic gapped phases in (2+1)d have long been know to be characterized by modular tensor categories[1] [9–13] (for reviews see e.g. [10, 14–16]), but their classification is currently far out of reach, despite recent advances [17–19]. Furthermore, unlike in (1+1)d, where the infinite-dimensional Virasoro algebra is available, charting out the space of CFTs in (2+1)d remains an elusive goal.

The Symmetry Topological Field Theory (SymTFT) has become the standard approach for organizing the symmetry properties of quantum field theories (QFTs) [20–23] because it naturally separates symmetry information from the local degrees of freedom. Often the same underlying local degrees of freedom could manifest entirely different global structures. The SymTFT effectively organizes such global information into the choice of symmetry boundaries. This approach has been most successful in (1+1)d. In particular, gauging discrete (not necessarily invertible) symmetries can be studied equivalently by specifying different symmetry boundary conditions. This approach not only provides a more intuitive understanding but also allows for a rigorous study of the global structure of QFTs, i.e. to classify the gapped boundary conditions of a topological field theory (TFT). Furthermore, the SymTFT can also be utilized to study and construct lattice models realizing generalized symmetries and can

---

[1] The symmetry enrichments of these categories have been studied for instance in [1–8].

therefore provide important steps towards understanding unconventional phases in lattice systems [20, 24–28].

Extending this analysis to higher dimensions becomes a formidable task as the complexity of higher fusion categories, which describe the symmetry structure, increases. On a superficial level, higher dimensional topological defects can form complicated structures. In the present paper we will focus on the next step, namely (2+1)d theories, which have fusion 2-category symmetries[2], and a (3+1)d SymTFT.

Such higher categorical symmetries, also referred to as non-invertible symmetries, have recently been observed to be symmetries of quantum field theories as well [30–32] (for reviews, see [33, 34]). This has raised the interest in such symmetries also from the perspective of high energy physics, as they will constrain IR phases of QFTs.

Complementing this, there are several recent mathematical results, which help making progress in this specific dimension, see [35–39]. First of all, there is a putative classification of all fusion 2-categories, up to gauging (or Morita equivalence), with an in depth homotopy classification upcoming in [39]. The key insight in these works is that all fusion 2-categories can be divided into two types bosonic and fermionic[3], where the bosonic ones are gauge related to $2\mathsf{Vec}_G^\tau$ describing a $G$ 0-form symmetry possibly with a 't Hooft anomaly $\tau$, upto a decoupled fusion 2-category obtained as condensation completion of an MTC. In particular this allows starting with a (3+1)d DW theory for $G$ with twist as the most general SymTFT for any fusion 2-category of bosonic type!

The classification result furthermore implies the corresponding SymTFTs are then Dijkgraaf Witten theories for $G$ [36,37,40–42], and the Drinfeld centers are well documented in the mathematics literature [38,43]. Thus, from this perspective, in this dimension the situation is much simpler than in (1+1)d, where there are endless non-group-like fusion categories.

However, the complexity in (2+1)d, and (3+1)d SymTFTs, arises from elsewhere. Namely, a gapped boundary condition[4], which is central to the SymTFT paradigm of constructing phases, what used to be a relatively simple, algebraic condition, becomes now a much richer structure: there are still analogs of the boundary conditions familiar in lower dimensions, which we here refer to as **minimal** boundary conditions. These are specified by imposing

---

[2]The symmetries are generated by topological surfaces $D_2$, topological lines $D_1$ and points $D_0$, with fusion rules. Even one of the simplest fusion 2-category, that associated to a $\mathbb{Z}_2^{(1)}$ 1-form symmetry, $2\mathsf{Rep}(\mathbb{Z}_2)$ has non-invertible, so-called condensation, defects, which correspond to gauging the 1-form symmetry on a 2d subspace [29].

[3]Note that both bosonic and fermionic types of fusion 2-categories describe symmetries of bosonic systems. The fusion 2-categories of fermionic type are characterized by the presence of a topological line operator having topological spin $-1$ that is transparent to the other topological operators comprising the symmetry.

[4]In (2+1)d systems with (3+1)d SymTFT the question of gapped boundary conditions of DW theories were discussed e.g. in [44–49] and in high energy physics in [50, 51].

Dirichlet or Neumann boundary conditions (BC) on the topological defects in the SymTFT. The simplest is the one for the 0-form symmetry group $G^{(0)}$, which we universally call the **minimal Dirichlet BC**. All other minimal boundary conditions can be obtained by gauging a subgroup $H \subset G$ with some discrete torsion $H^3(H, U(1))$.

The genuinely new aspect in this dimension is however the existence of s that we call **non-minimal BCs**: these are obtained by stacking the universal Dirichlet BC with not just discrete torsion, but a TQFT $\mathfrak{T}$ with a symmetry $H \subset G$ and gauging the diagonal symmetry $H$. This construction is reminiscent of the theta-constructions of topological defects [52, 53] and the modified Neumann boundary conditions [54, 55]. This possibility immediately implies that there is an infinite number of gapped boundary conditions for any SymTFT in (3+1)d.

The goal of this paper is to initiate the exploration of all these gapped boundary conditions of (3+1)d SymTFTs, and to furthermore utilize them to construct gapped symmetric phases with fusion 2-category symmetries, following the SymTFT proposal for a categorical Landau paradigm in [56, 57]. We will revisit this in the subsequent sections in detail.

The abundance of gapped boundary conditions implies that even for a simple group such as $G = \mathbb{Z}_2$ the structure of gapped boundary conditions of its SymTFT, i.e. the Dijkgraaf-Witten theory for $\mathbb{Z}_2$, reveals an infinity of related symmetries, and an infinity of gapped phases for each of them. The first half of the paper will therefore illustrate this novel feature by focusing on this seemingly simple example.

We outline the general approach in section 5, explaining the construction of minimal and non-minimal boundary conditions as well as gapped phases. As an application we provide a detailed analysis for general abelian groups in section 6 and for non-abelian groups, including F2Cs of fermionic type, in future work [58]. Even for symmetries that arise as minimal BCs, e.g. 1-form symmetry groups, higher group symmetries and their representation category symmetries, the structure of gapped phases is interesting and in this generality – to our knowledge – not discussed elsewhere.

This framework will allow for an extension to gapless phases, which we intend to explore in the future. In particular it will be of interest to construct (intrinsically) gapless phases with topological order, generalizing such phases exhibiting SPT and SSB orders in (1+1)d. The SymTFT approach is particularly insightful to achieve this as demonstrated in (1+1)d [25, 57, 59–62] and has started being explored in (3+1)d [63, 64].

# 2 Symmetries and SymTFT of (2+1)d Theories

The most general finite, internal symmetry of a bosonic relativistic (2+1)d quantum field theory is a fusion 2-category, denoted $\mathcal{S}$. The goal of this series of papers is to systematically characterize all $\mathcal{S}$-symmetric phases, gapped and gapless. The starting point for this analysis is a comprehensive discussion of the possible symmetries, which, thanks to results in mathematics, are known in this instance. The tool to classify phases is the Symmetry Topological Field Theory (SymTFT).

In short the main result is that all fusion 2-categories are gauge related (or Morita equivalent) to $2\mathsf{Vec}_G^\tau$, upto decoupled factors which are (condensation completions of) MTCs. In particular this means that the SymTFT is the same for all symmetries of this type to the SymTFT for $2\mathsf{Vec}_G^\tau$, which is a Dijkgraaf Witten theory based on group $G$ and twist $\tau$. This hugely simplifies the analysis of fusion 2-category symmetries and their phases.

The interesting structure arises in the gapped boundary conditions. There is a canonical Dirichlet boundary condition which gives rise to the symmetry $2\mathsf{Vec}_G^\tau$. We will argue that there is an infinity of gapped boundary conditions, which is the key new feature in (2+1)d, and raises the level of complexity in this dimension.

## 2.1 Classification of Fusion 2-Category Symmetries

What makes the (2+1)d case particularly accessible is the fact that there is by now a classification of fusion 2-categories. A full homotopy classification of fusion 2-categories will appear shortly here [39] see also [36–38]. We will briefly summarize this, as it will be crucial in establishing the generality of the results we present in this paper:

Fusion 2-categories roughly fall into two classes: so-called bosonic and fermionic. Symmetries of bosonic type are all gauge-related to 0-form symmetries, possibly with 't Hooft anomalies, i.e. every fusion 2-category $\mathcal{S}$ of bosonic type is gauge or Morita equivalent to

$$\mathcal{S} \cong 2\mathsf{Vec}_G^\tau \boxtimes \Sigma\mathcal{M}\,, \tag{2.1}$$

with $\mathcal{M}$ is an MTC, or more precisely a nondegenerate braided fusion 1-category.[5] For every two multi-fusion 2-categories $\mathcal{C}$ and $\mathcal{D}$, we have $\mathcal{Z}(\mathcal{C}\boxtimes\mathcal{D}) \cong \mathcal{Z}(\mathcal{C})\boxtimes\mathcal{Z}(\mathcal{D})$ as braided monoidal 2-categories. Together with $\mathcal{Z}(\Sigma\mathcal{M}) \cong 2\mathsf{Vec}$, we arrive at the following corollary [38]:

> The Drinfeld center $\mathcal{Z}(\mathcal{S})$ of any fusion 2-category $\mathcal{S}$ of bosonic type is equivalent as a braided fusion 2-category to $\mathcal{Z}(2\mathsf{Vec}_G^\tau)$.

---

[5]A similar statement for the fusion 2-categories of fermionic type also holds by working with SVec and allowing $\mathcal{M}$ to be slightly-degenerate, i.e. Müger center of $\mathcal{M}$ = SVec.

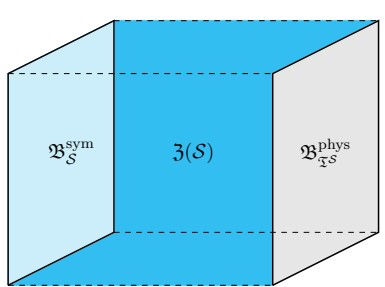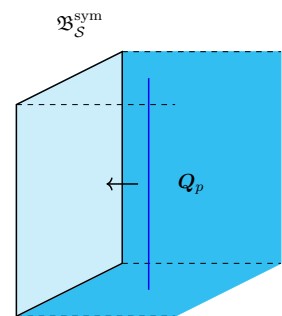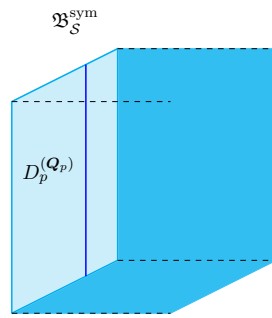

Figure 1: SymTFT picture: On the left hand side we show the interval compactification of the SymTFT with the symmetry and physical boundary. On the right hand side we show how the projection (Neumann b.c.s) of topological defects in the SymTFT, denoted $\boldsymbol{Q}_p$, gives rise to the symmetry generators $D_p$.

This means that for symmetries of bosonic type, the SymTFT in (3+1)d is always a DW theory based on a finite group $G$ with twist. The symmetry 2-categories of bosonic type are realized by topological defects living on topological boundary conditions of such (3+1)d DW theories.

## 2.2   2-Categorical Landau Paradigm

The SymTFT [20–23] provides a systematic and concise way to classify all gapped and gapless symmetric phases. This was suggested as a higher categorical Landau paradigm in any dimension in [57], and further substantiated in (1+1)d in [25, 59–62, 65].

The SymTFT $\mathfrak{Z}(\mathcal{S})$ for a $(2+1)$d theory with symmetry $\mathcal{S}$ is a (3+1)d TFT. A state sum model of it is provided by Douglas and Reutter [35]. The topological defects of the SymTFT form the so-called Drinfeld Center of the fusion 2-category $\mathcal{S}$, denoted by $\mathcal{Z}(\mathcal{S})$. These are the main players for the following analysis.

The SymTFT is put on a slab, with two boundary conditions – see figure 1:

- $\mathfrak{B}^{\text{sym}}$: the symmetry boundary, which is a gapped boundary condition. For fusion 2-categories, we specify these in terms of the surfaces and lines that can end on the boundary, along with additional lines localized along the boundary. These have to be mutually local and be a maximal set. In (1+1)d the analog are Lagrangian algebras.

- $\mathfrak{B}^{\text{phys}}$: the physical boundary, which may or may not be gapped. In the present paper

we focus on gapped phases, and $\mathfrak{B}^{\mathrm{phys}}$ is chosen to be a topological boundary condition as well.

We will often depict this schematically in projection to the plane:

$$
\begin{array}{ccc}
\mathfrak{B}^{\mathrm{sym}}_{\mathcal{S}} & \mathfrak{B}^{\mathrm{phys}}_{\mathfrak{T}\mathcal{S}} & \mathfrak{T}^{\mathcal{S}} \\
\end{array}
$$

$$ \tag{2.2} $$

The boundaries are $(2+1)$ dimensional, and the bulk is $(3+1)$ dimensional.

## 2.3 SymTFT for Fusion 2-Categories

In light of the gauge-equivalence of 2-fusion categories of bosonic type to $2\mathsf{Vec}^{\tau}_G$ (upto decoupled MTCs), it suffices to construct the Drinfeld center for this class of symmetries. This makes the case of such fusion 2-categories particularly accessible for the SymTFT approach.

From general considerations we know that the topological defects of the $(3+1)$d $G$ Dijkgraaf-Witten theory form the Drinfeld center of the fusion 2-category $2\mathsf{Vec}_G$ which can be organized as [43, 56][6]

$$
\mathcal{Z}(2\mathsf{Vec}_G) = \boxplus_{[g]} 2\mathsf{Rep}(H_g)\,. \tag{2.3}
$$

Here the sum is over conjugacy classes $[g]$ of $G$ and $H_g$ is the stabilizer group of $g \in [g]$. We denote the topological defects of dimension $k$ by $\boldsymbol{Q}^a_k$, where $a$ is a label for different defects. Each component in (2.3) has a non-trivial topological 2d surface defect, which we denote by the conjugacy class $[g]$

$$
\boldsymbol{Q}^{[g]}_2\,. \tag{2.4}
$$

All other surfaces in the component are related to it by condensations of lines on top of it. The topological lines living on this surface are labelled by

$$
\boldsymbol{Q}^{[g],\boldsymbol{R}}_1\,, \qquad \boldsymbol{R} \in \mathsf{Rep}(H_g)\,, \tag{2.5}
$$

To construct gapped boundary conditions we start with the one that realizes the symmetry $2\mathsf{Vec}^{\tau}_G$, which we call the canonical Dirichlet boundary condition. It is specified by imposing

---

[6]Here, we neglect the codimension-1 defects which are all in the same Schur component, i.e. constructable as condensations of the codimension-2 and co-dimension-3 defects. The 3-category of topological defects which includes the co-dimension-1 defects is the delooping of the 2-category denoted here as $\mathcal{Z}(2\mathsf{Vec}_G)$ [66].

Dirichlet boundary conditions onto the following topological defects:

$$\mathfrak{B}_{\text{Dir}} : \qquad \begin{cases} \boldsymbol{Q}_2^{[\text{id}]} \\ \displaystyle\bigoplus_{\boldsymbol{R}} \dim_{\boldsymbol{R}} \boldsymbol{Q}_1^{[\text{id}],\boldsymbol{R}} \end{cases} . \qquad (2.6)$$

**Boundary Conditions from Stacking and Gauging.** From the SymTFT philosophy it is clear that the classification of symmetries, and of gapped phases, relies on the classification of gapped boundary conditions. We can start with the boundary condition $\mathfrak{B}_{\text{Dir}}$ and gauge $H$, including discrete torsion $\omega$. The resulting boundary conditions will be called **minimal boundary conditions** $(\mathbf{Neu}(H), \omega)$. They are specified fully by imposing Dirichlet or Neumann boundary conditions onto the topological defects $\boldsymbol{Q}_2$ and $\boldsymbol{Q}_1$. The discrete torsion implies that the ends of surfaces $\boldsymbol{Q}_2^h$, $h \in H$ on the boundary can have non-trivial associators given by $\omega \in H^3(H, U(1))$.

The SymTFT will have topological defects that are lines $\boldsymbol{Q}_1$ and surfaces $\boldsymbol{Q}_2$ which we depict in terms of (dashed) lines in the SymTFT. E.g. if a surface $\boldsymbol{Q}_2$ can end on both boundaries this is shown by a solid line, and gives rise to a genuine line defect in the sandwich compactification

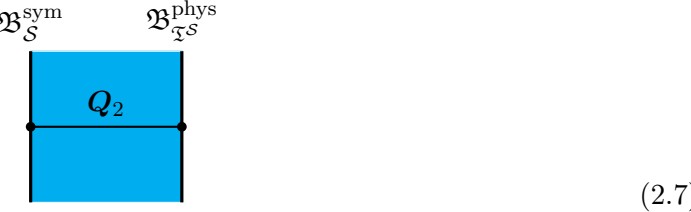

$$(2.7)$$

If a line $\boldsymbol{Q}_1$ can end on both boundaries this will be shown by a dashed line and gives rise to a genuine topological local operator in the compactification to (2+1)d

$$\begin{array}{cc} \mathfrak{B}_{\mathcal{S}}^{\text{sym}} & \mathfrak{B}_{\mathfrak{T}\mathcal{S}}^{\text{phys}} \\ & \\ \circ\text{-----}\boldsymbol{Q}_1\text{-----}\circ \end{array}$$

$$(2.8)$$

Lines and surfaces that have Neumann BC's, i.e. are projected parallel to the boundary, will be depicted in 3d as in figure 1. There are also twisted sector operators, which can only end on the physical boundary and form an L-shaped configuration in the SymTFT. We will discuss these at length, but will refrain from cluttering the pictures with these.

**Non-minimal Boundary Conditions.** The most interesting, dimension specific, new ingredient in (2+1)d is that we have the freedom to stack onto the boundary a (2+1)d $G$-symmetric TQFT $\mathfrak{T}^G$ (with anomaly $\tau$) before gauging. The resulting boundary conditions will be referred to as **non-minimal boundary conditions**. Studying this enrichment of boundary conditions, which yields for any $G$ an infinite number of symmetries and symmetric topological phases, is particularly interesting and will be discussed in detail in section 3 for $G = \mathbb{Z}_2$ and extended to general $(G, \tau)$ in section 5, with an analysis for abelian $G$ and trivial $\tau$ in section 6.

## 3 Gapped Boundary Conditions of (3+1)d $\mathbb{Z}_2$ DW Theory

In the previous section we discussed the classification of symmetry categories in (2+1)d. As explained there, such symmetries fall into two classes, that may be loosely referred to as bosonic and fermionic, though one should keep in mind that both types of symmetries are realizable in bosonic theories. Moreover, the symmetries of bosonic type are all gauge-related to 0-form symmetries, possibly with 't Hooft anomalies[7]. This means that the (3+1)d SymTFT for (2+1)d symmetries of bosonic type are (3+1)d DW gauge theories based on finite (0-form) groups, possibly with DW twists. The symmetry 2-categories of bosonic type are realized by topological defects living on topological boundary conditions of such (3+1)d DW theories.

In this and the following section, we will discuss the construction of such topological boundary conditions. Most notably, we will find that there is an infinite number of these boundary conditions (which are all irreducible). In fact, even if we regard two boundary conditions related by stacking of 3d TFTs to be equivalent, there are still an infinite number of equivalence classes of boundary conditions!

We will build up to these results in a pedagogical fashion starting first with non-anomalous $\mathbb{Z}_2$ group, and then accounting for more general abelian and non-abelian groups and anomalies. Most of the discussion can be straightforwardly generalized to any (possibly anomalous) higher-group symmetry in any spacetime dimension d, but for concreteness we will often restrict to 0-form symmetries in 3d.

---

[7]This result is true upto a decoupled MTC describing a fully anomalous possibly non-invertible 1-form symmetry. The inclusion or exclusion of such an MTC from the symmetry 2-category does not change the SymTFT, and hence can be ignored in the classification of (3+1)d SymTFTs.

## 3.1 Minimal Boundary Conditions

Let us begin with the simplest of cases where we have a non-anomalous $G = \mathbb{Z}_2$ 0-form symmetry in 3d. The (3+1)d DW theory is described by the action

$$S = \int a_1 \cup \delta b_2 \,, \tag{3.1}$$

where $a_1$ and $b_2$ are $\mathbb{Z}_2$-valued gauge fields of form degree specified by their subscript.

There are two well-known boundary conditions which are described as follows:

- $\mathfrak{B}_{\text{Dir}}$ on which $a_1$ is fixed to be a background field $A_1$, and $b_2$ is free to fluctuate. In other words, $a_1$ has Dirichlet BC and $b_2$ has Neumann BC.

- $\mathfrak{B}_{\text{Neu}}$ on which $b_2$ is fixed to be a background field $B_2$, and $a_1$ is free to fluctuate. In other words, $b_2$ has Dirichlet BC and $a_1$ has Neumann BC.

We refer to the two boundary conditions respectively as Dirichlet and Neumann, reflecting the condition imposed on the gauge field $a_1$. Note that the Dirichlet BC has a background $A_1$ field and hence carries a (non-anomalous) $\mathbb{Z}_2$ 0-form symmetry, while the Neumann BC has a background $B_2$ field and hence carries a (non-anomalous) $\mathbb{Z}_2$ 1-form symmetry. These two boundaries serve as symmetry boundaries for $\mathbb{Z}_2$ 0-form and 1-form symmetries respectively.

It is instructive to encode the information of these BCs in terms of the boundary behavior of the bulk topological defects. There are two key topological defects in the bulk (3+1)d theory

$$\boldsymbol{Q}_1 := \exp\left(\pi i \int a_1\right), \qquad \boldsymbol{Q}_2 := \exp\left(\pi i \int b_2\right), \tag{3.2}$$

which are respectively line and surface operators. In this paper, the spacetime dimension of topological defects are reflected in the subscripts of their label. Topological defects in (3+1)d are labeled by $\boldsymbol{Q}$ throughout this paper. In addition to the above-mentioned $\boldsymbol{Q}_1$ and $\boldsymbol{Q}_2$, there are other two and three-dimensional topological defects in the (3+1)d theory, which can all be produced as condensation defects for $\boldsymbol{Q}_1$ and $\boldsymbol{Q}_2$. Condensation defects do not play a significant role in this paper, and henceforth they are suppressed.

In terms of $\boldsymbol{Q}_1$ and $\boldsymbol{Q}_2$, the above two boundaries can be re-expressed as:

- $\mathfrak{B}_{\text{Dir}}$ is a boundary on which $\boldsymbol{Q}_1$ can end, and hence disappears when projected to the boundary. On the other hand, $\boldsymbol{Q}_2$ cannot end and projects to a surface $D_2$ on the boundary, which generates the $\mathbb{Z}_2$ 0-form symmetry.

- $\mathfrak{B}_{\text{Neu}}$ is a boundary on which $\boldsymbol{Q}_2$ can end, and hence disappears when projected to the boundary. On the other hand, $\boldsymbol{Q}_1$ cannot end and projects to a line $D_1$ on the boundary, which generates the $\mathbb{Z}_2$ 1-form symmetry.

Throughout this paper, topological defects in 3d will be labeled by $D$ and its subscript will denote the total spacetime dimension of the defect.

So far, all of the above extends straightforwardly to arbitrary $d$ by replacing $b_2 \to b_{d-1}$. For $d = 3$, there is an additional well-known BC that resembles $\mathfrak{B}_{\text{Neu}}$ discussed above. In order to describe this BC, let us note that $\mathfrak{B}_{\text{Neu}}$ can be obtained by gauging the $\mathbb{Z}_2$ 0-form symmetry of $\mathfrak{B}_{\text{Dir}}$ with the $\mathbb{Z}_2$ 1-form symmetry of $\mathfrak{B}_{\text{Neu}}$ being the dual symmetry obtained after gauging. That is, we can realize $\mathfrak{B}_{\text{Neu}}$ by using the action

$$S = \int_{M_4} a_1 \cup \delta b_2 + \int_{\partial M_4} a_1 \cup B_2 \,, \tag{3.3}$$

where $M_4$ is a 4-manifold and $\partial M_4$ is its boundary. Similarly, $\mathfrak{B}_{\text{Dir}}$ is obtained by gauging the $\mathbb{Z}_2$ 1-form symmetry of $\mathfrak{B}_{\text{Neu}}$ with the $\mathbb{Z}_2$ 0-form symmetry of $\mathfrak{B}_{\text{Dir}}$ being the dual symmetry obtained after gauging. That is, we can realize $\mathfrak{B}_{\text{Dir}}$ by using the action

$$S = \int_{M_4} a_1 \cup \delta b_2 + \int_{\partial M_4} A_1 \cup b_2 \,. \tag{3.4}$$

The above description (3.3) for $\mathfrak{B}_{\text{Neu}}$ can be modified by the addition of a topological term along the 3d boundary

$$S = \int_{M_4} a_1 \cup \delta b_2 + \int_{\partial M_4} \left( a_1 \cup B_2 + a_1 \cup a_1 \cup a_1 \right) \,, \tag{3.5}$$

which leads to another Neumann boundary condition that we denote by $\mathfrak{B}_{\text{Neu},\omega}$. The topological term $a_1^3$ describes the non-trivial element of the group cohomology

$$H^3(\mathbb{Z}_2, U(1)) = \mathbb{Z}_2 \,, \tag{3.6}$$

which we also denote by $\omega$. One says that $\mathfrak{B}_{\text{Neu},\omega}$ is obtained from $\mathfrak{B}_{\text{Dir}}$ by gauging the $\mathbb{Z}_2$ 0-form symmetry with discrete torsion characterized by $\omega$.

In terms of bulk topological defects $\boldsymbol{Q}_1$ and $\boldsymbol{Q}_2$, the BC $\mathfrak{B}_{\text{Neu},\omega}$ is characterized also by the properties that $\boldsymbol{Q}_2$ can end while $\boldsymbol{Q}_1$ cannot thus generating a non-anomalous $\mathbb{Z}_2$ 1-form symmetry. Thus, $\mathfrak{B}_{\text{Neu},\omega}$ can be equally well used as the symmetry boundary for $\mathbb{Z}_2$ 1-form symmetry. The difference between $\mathfrak{B}_{\text{Neu}}$ and $\mathfrak{B}_{\text{Neu},\omega}$ arises upon a closer analysis of how $\boldsymbol{Q}_2$ can end along these boundaries. The end of $\boldsymbol{Q}_2$ describes a topological line defect $D_1^{\text{ng}}$ living on the boundary (which is non-genuine as it is attached to the bulk surface $\boldsymbol{Q}_2$). The line defects $D_1^{\text{ng}}$ have a trivial F-symbol along $\mathfrak{B}_{\text{Neu}}$, but non-trivial F-symbols characterized by

$\omega$ along the boundary $\mathfrak{B}_{\text{Neu},\omega}$:

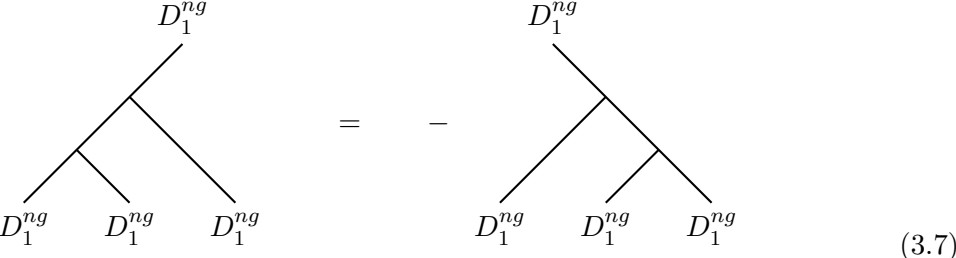

$$\tag{3.7}$$

## 3.2 Non-Minimal Boundary Conditions

The above three BCs are referred to as minimal BCs as they do not have any additional data on the boundary beyond a topological term for the bulk fields. In this subsection, we discuss the construction of non-minimal BCs.

One rather uninteresting class of non-minimal irreducible BCs is obtained by simply stacking the minimal BCs with irreducible[8] 3d TFTs, which we denote as

$$\mathfrak{B}_{\text{Dir}} \boxtimes \mathfrak{T} \equiv \mathfrak{B}_{\text{Dir}}^{\mathfrak{T}}$$
$$\mathfrak{B}_{\text{Neu}} \boxtimes \mathfrak{T} \tag{3.8}$$
$$\mathfrak{B}_{\text{Neu},\omega} \boxtimes \mathfrak{T} \,,$$

where $\mathfrak{T}$ is a 3d TFT. This gives rise to an infinite number of new irreducible topological BCs because there are an infinite number of irreducible 3d TFTs. Note that we have introduced a notation $\mathfrak{B}_{\text{Dir}}^{\mathfrak{T}}$ to denote $\mathfrak{B}_{\text{Dir}} \boxtimes \mathfrak{T}$ as these boundaries will play a significant role in the remaining discussion in this section.

As an example, one may choose $\mathfrak{T}$ to be a $\mathbb{Z}_2$ DW theory in 3d and stack it on top of $\mathfrak{B}_{\text{Dir}}$ without coupling it to the bulk (3+1)d $\mathbb{Z}_2$ DW theory. The combined action takes the form

$$S = \int_{M_4} a_1 \cup \delta b_2 + \int_{\partial M_4} \left( A_1 \cup b_2 + a_1' \cup \delta b_1' \right) \,, \tag{3.9}$$

where $a_1'$ and $b_1'$ are new $\mathbb{Z}_2$ gauge fields that live only on the 3d boundary.

To produce more interesting non-minimal irreducible BCs, we need to introduce couplings between the 3d and (3+1)d degrees of freedom. One way to do that is via the so-called theta construction [52, 53], where couplings are introduced by gauging a combined symmetry that acts both on the stacked 3d degrees of freedom and the boundary modes of the (3+1)d degrees of freedom. In more detail, let $\mathfrak{T}_{\mathbb{Z}_2}$ be an irreducible 3d TFT with a $\mathbb{Z}_2$ 0-form symmetry. Then we can obtain a non-minimal Neumann-type BC by first stacking $\mathfrak{B}_{\text{Dir}}$ with $\mathfrak{T}_{\mathbb{Z}_2}$, and

---

[8]That is, 3d TFTs having a single vacuum.

then gauging the diagonal $\mathbb{Z}_2$ 0-form symmetry

$$\frac{\mathfrak{B}_{\mathrm{Dir}} \boxtimes \mathfrak{T}_{\mathbb{Z}_2}}{\mathbb{Z}_2^{(0)}} = \mathfrak{B}_{\mathrm{Neu}}^{\mathfrak{T}_{\mathbb{Z}_2}} . \tag{3.10}$$

There are an infinite number of BCs of the form $\mathfrak{B}_{\mathrm{Neu}}^{\mathfrak{T}_{\mathbb{Z}_2}}$ as there are an infinite number of $\mathbb{Z}_2$ symmetric irreducible 3d TFTs. It should be noted that these boundary conditions cannot be obtained by stacking 3d TFTs on top of minimal BCs

$$\mathfrak{B}_{\mathrm{Neu}}^{\mathfrak{T}_{\mathbb{Z}_2}} \neq \mathfrak{B}_x \boxtimes \mathfrak{T}', \qquad x \in \{\mathrm{Dir}, \mathrm{Neu}, (\mathrm{Neu}, \omega)\} , \tag{3.11}$$

for any choices of $x$ and 3d TFT $\mathfrak{T}'$.

As an example, we can take $\mathfrak{T}_{\mathbb{Z}_2}$ to be $\mathbb{Z}_2$ DW theory with a $\mathbb{Z}_2^{(0)}$ symmetry chosen such that the action of $\mathfrak{T}_{\mathbb{Z}_2}$ involving background fields is

$$S = \int a_1' \cup \delta b_1' + C_1 \cup b_1' \cup b_1' , \tag{3.12}$$

where $C_1$ is background field for $\mathbb{Z}_2^{(0)}$. Let us feed it into the theta construction. Using the action for $\mathfrak{B}_{\mathrm{Dir}}$ from (3.4), the action for $\mathfrak{B}_{\mathrm{Dir}} \boxtimes \mathfrak{T}_{\mathbb{Z}_2}$ is

$$S = \int_{M_4} a_1 \cup \delta b_2 + \int_{\partial M_4} \left( A_1 \cup b_2 + a_1' \cup \delta b_1' + C_1 \cup b_1' \cup b_1' \right) . \tag{3.13}$$

The diagonal gauging is performed by identifying the background fields $A_1 = C_1$ and promoting them to a gauge field $a_1$ which is the restriction to boundary of the bulk $a_1$ gauge field. The resulting action for $\mathfrak{B}_{\mathrm{Neu}}^{\mathfrak{T}_{\mathbb{Z}_2}}$ is

$$S = \int_{M_4} a_1 \cup \delta b_2 + \int_{\partial M_4} \left( a_1 \cup b_2 + a_1' \cup \delta b_1' + a_1 \cup b_1' \cup b_1' \right) . \tag{3.14}$$

We can perform the theta construction starting from the Neumann minimal BCs as well. Let $\mathfrak{T}_{\mathbb{Z}_2^{(1)}}$ be a $\mathbb{Z}_2$ 1-form symmetric 3d TFT. We stack it with either $\mathfrak{B}_{\mathrm{Neu}}$ or $\mathfrak{B}_{\mathrm{Neu},\omega}$ and gauge the diagonal $\mathbb{Z}_2^{(1)}$ to produce boundaries

$$\frac{\mathfrak{B}_{\mathrm{Neu}} \boxtimes \mathfrak{T}_{\mathbb{Z}_2^{(1)}}}{\mathbb{Z}_2^{(1)}} , \qquad \frac{\mathfrak{B}_{\mathrm{Neu},\omega} \boxtimes \mathfrak{T}_{\mathbb{Z}_2^{(1)}}}{\mathbb{Z}_2^{(1)}} , \tag{3.15}$$

but both of these boundaries can be recognized as

$$\mathfrak{B}_{\mathrm{Dir}} \boxtimes \frac{\mathfrak{T}_{\mathbb{Z}_2^{(1)}}}{\mathbb{Z}_2^{(1)}} , \tag{3.16}$$

that is we obtain $\mathfrak{B}_{\mathrm{Dir}}$ upto a decoupled 3d TFT.[9]

---

[9]Note that the dual $\mathbb{Z}_2$ 0-form symmetries on $\mathfrak{B}_{\mathrm{Neu}}^{\mathfrak{T}_{\mathbb{Z}_2^{(1)}}}$ and $\mathfrak{B}_{\mathrm{Neu},\omega}^{\mathfrak{T}_{\mathbb{Z}_2^{(1)}}}$ are related by the stacking of a 3d $\mathbb{Z}_2^{(0)}$ SPT phase characterized by $\omega$.

## 3.3 Classification of all Boundary Conditions

Here we show that the boundaries $\mathfrak{B}_{\text{Dir}}^{\mathfrak{T}}$ and $\mathfrak{B}_{\text{Neu}}^{\mathfrak{T}_{\mathbb{Z}_2}}$ are all the possible topological BCs of the (3+1)d $\mathbb{Z}_2$ DW theory. The argument relies on the sandwich construction of $\mathcal{S}$-symmetric QFTs, according to which there is a one-to-one correspondence between BCs of the SymTFT $\mathfrak{Z}(\mathcal{S})$ and $\mathcal{S}$-symmetric QFTs. Using a BC of the SymTFT as physical boundary in the sandwich construction, with the symmetry boundary being $\mathfrak{B}_{\mathcal{S}}^{\text{sym}}$, the corresponding $\mathcal{S}$-symmetric QFT is the result of the interval compactification. In the topological/gapped setting, this specializes to a one-to-one correspondence between topological BCs of the SymTFT $\mathfrak{Z}(\mathcal{S})$ and $\mathcal{S}$-symmetric TFTs.

Let us apply this one-to-one correspondence to non-anomalous $\mathbb{Z}_2$ 0-form symmetry in 3d. The symmetry boundary is chosen to be $\mathfrak{B}_{\text{Dir}}$. Let $\mathfrak{T}_{\mathbb{Z}_2}$ be a $\mathbb{Z}_2$ 0-form symmetric 3d TFT. Note that we now allow the 3d TFT underlying 3d TFT to be reducible, i.e. have multiple vacua (or equivalently ground states on a sphere $S^2$). In other words, we are allowing for 3d TFTs exhibiting spontaneous breaking of $\mathbb{Z}_2$ 0-form symmetry. This is different from the situation in the previous subsection where we required the underlying 3d TFT to be irreducible.

The physical boundary condition corresponding to the 3d TFT $\mathfrak{T}_{\mathbb{Z}_2}$ can be constructed using a theta construction

$$\mathfrak{B}_{\mathfrak{T}_{\mathbb{Z}_2}}^{\text{phys}} = \frac{\mathfrak{B}_{\text{Dir}} \boxtimes \mathfrak{T}_{\mathbb{Z}_2}}{\mathbb{Z}_2^{(0)}}\,, \tag{3.17}$$

where we stack $\mathfrak{B}_{\text{Dir}}$ with $\mathfrak{T}_{\mathbb{Z}_2}$ and gauge the diagonal $\mathbb{Z}_2$ 0-form symmetry. Indeed, performing the sandwich interval compactification with this physical boundary and $\mathfrak{B}_{\text{Dir}}$ as symmetry boundary reverses the $\mathbb{Z}_2^{(0)}$ gauging appearing in (3.17) and projects out the bulk $b_2$ gauge field, while the bulk $a_1$ gauge field is frozen to become the background field $A_1$ for a $\mathbb{Z}_2$ 0-form symmetry. In total, only the $\mathfrak{T}_{\mathbb{Z}_2}$ degrees of freedom survive with $A_1$ acting as the background field for its $\mathbb{Z}_2$ 0-form symmetry.

Now let us recall that the 3d TFTs with $\mathbb{Z}_2$ 0-form symmetry are of two types:

- For the first type of TFTs, the $\mathbb{Z}_2$ symmetry is spontaneously broken and we can express

$$\mathfrak{T}^{\mathbb{Z}_2} = \mathfrak{T} \oplus \mathfrak{T}\,, \tag{3.18}$$

  i.e. $\mathfrak{T}^{\mathbb{Z}_2}$ is a theory with two vacua, with each vacuum governed by an irreducible 3d TFT $\mathfrak{T}$, which may carry topological order, or in other words, non-trivial topological line defects. The $\mathbb{Z}_2$ 0-form symmetry simply exchanges the two $\mathfrak{T}$-vacua.

- For the second type of TFTs, the $\mathbb{Z}_2$ symmetry is not spontaneously broken and $\mathfrak{T}^{\mathbb{Z}_2}$ is a TFT with one vacuum, which may carry non-trivial topological line defects. The $\mathbb{Z}_2$

0-form symmetry is realized by some condensation surface defect and may in general act non-trivially on the topological line defects.

For the first type of $\mathbb{Z}_2^{(0)}$ symmetric TFTs, we have

$$\mathfrak{B}_{\mathrm{Dir}} \boxtimes \mathfrak{T}_{\mathbb{Z}_2} = \left(\mathfrak{B}_{\mathrm{Dir}} \boxtimes \mathfrak{T}\right) \oplus \left(\mathfrak{B}_{\mathrm{Dir}} \boxtimes \mathfrak{T}\right), \tag{3.19}$$

with the diagonal $\mathbb{Z}_2^{(0)}$ symmetry exchanging the two $\mathfrak{B}_{\mathrm{Dir}} \boxtimes \mathfrak{T}$ factors. Hence, we have

$$\mathfrak{B}_{\mathfrak{T}_{\mathbb{Z}_2}}^{\mathrm{phys}} = \frac{\left(\mathfrak{B}_{\mathrm{Dir}} \boxtimes \mathfrak{T}\right) \oplus \left(\mathfrak{B}_{\mathrm{Dir}} \boxtimes \mathfrak{T}\right)}{\mathbb{Z}_2^{(0)}} = \mathfrak{B}_{\mathrm{Dir}} \boxtimes \mathfrak{T} = \mathfrak{B}_{\mathrm{Dir}}^{\mathfrak{T}}. \tag{3.20}$$

For the second type of $\mathbb{Z}_2^{(0)}$ symmetric TFTs, we have

$$\mathfrak{B}_{\mathfrak{T}_{\mathbb{Z}_2}}^{\mathrm{phys}} = \mathfrak{B}_{\mathrm{Neu}}^{\mathfrak{T}_{\mathbb{Z}_2}}, \tag{3.21}$$

simply by the definition (3.10) of $\mathfrak{B}_{\mathrm{Neu}}^{\mathfrak{T}_{\mathbb{Z}_2}}$.

Thus, we have justified the claim that $\mathfrak{B}_{\mathrm{Dir}}^{\mathfrak{T}}$ and $\mathfrak{B}_{\mathrm{Neu}}^{\mathfrak{T}_{\mathbb{Z}_2}}$ capture all the possible irreducible topological BCs of the (3+1)d $\mathbb{Z}_2$ DW theory. In particular, $\mathfrak{B}_{\mathrm{Dir}}$ lies in the family $\mathfrak{B}_{\mathrm{Dir}}^{\mathfrak{T}}$ of BCs and is obtained by choosing $\mathfrak{T}$ to be the trivial 3d TFT. On the other hand, $\mathfrak{B}_{\mathrm{Neu}}$ and $\mathfrak{B}_{\mathrm{Neu},\omega}$ lie in the family $\mathfrak{B}_{\mathrm{Neu}}^{\mathfrak{T}_{\mathbb{Z}_2}}$ of BCs, and are obtained by choosing $\mathfrak{T}_{\mathbb{Z}_2}$ to be the trivial and non-trivial $\mathbb{Z}_2^{(0)}$ SPT phases respectively.

### 3.4 Categorical Characterization of Boundary Conditions

Although we have classified all possible BCs in the previous subsection, we still lack a clear and useful understanding of the physical properties of these BCs. For instance, we have not even discussed the symmetries that are associated to these BCs. For this purpose, we need to understand the topological defects living on these BCs and how they interact with the bulk topological defects $\boldsymbol{Q}_1$ and $\boldsymbol{Q}_2$. All of this information is best encoded using category theory, which we will use throughout this subsection.

Let us first discuss the structure of $\mathfrak{B}_{\mathrm{Dir}}^{\mathfrak{T}}$ BCs. These are classified by irreducible 3d TFTs. Such a TFT $\mathfrak{T}$ is characterized by an MTC $\mathcal{M}$ describing topological line defects of $\mathfrak{T}$. These line defects live along the boundary $\mathfrak{B}_{\mathrm{Dir}}^{\mathfrak{T}}$ and are completely transparent to the projection $D_2$ of $\boldsymbol{Q}_2$ along the boundary. The symmetry associated to $\mathfrak{B}_{\mathrm{Dir}}^{\mathfrak{T}}$ is thus a direct product of a $\mathbb{Z}_2$ 0-form symmetry generated by $D_2$ with a possibly non-invertible anomalous 1-form symmetry described by the MTC $\mathcal{M}$. The resulting symmetry 2-category is denoted as

$$\mathcal{S}(\mathfrak{B}_{\mathrm{Dir}}^{\mathfrak{T}}) = 2\mathsf{Vec}_{\mathbb{Z}_2} \boxtimes \Sigma\mathcal{M}, \tag{3.22}$$

where $\Sigma\mathcal{M}$ is the fusion 2-category[10] obtained by adding all possible condensation surface defects constructed from lines in $\mathcal{M}$.

In particular, for $\mathfrak{B}_{\mathrm{Dir}}$ we have $\mathcal{M} = \mathsf{Vec}$ and

$$\mathcal{S}(\mathfrak{B}_{\mathrm{Dir}}) = 2\mathsf{Vec}_{\mathbb{Z}_2} \boxtimes \Sigma\mathsf{Vec} = 2\mathsf{Vec}_{\mathbb{Z}_2} \boxtimes 2\mathsf{Vec} = 2\mathsf{Vec}_{\mathbb{Z}_2} \,, \tag{3.23}$$

and for the example (3.9), we have $\mathcal{M} = \{D_1^{\mathrm{id}}, D_1^e, D_1^m, D_1^f\}$, namely the MTC associated to toric code, leading to

$$\mathcal{S}(\mathfrak{B}_{\mathrm{Dir}}^{\mathfrak{T}}) = \mathbb{Z}_2^{(0)} \times \mathbb{Z}_2^{(1),e} \times \mathbb{Z}_2^{(1),m} \,, \tag{3.24}$$

where we have two $\mathbb{Z}_2^{(1)}$ factors distinguished by $e$ and $m$. There is a mixed 't Hooft anomaly between these two factors described in terms of background fields as

$$B_2^e \cup B_2^m \,, \tag{3.25}$$

where the anomaly is a $\mathbb{Z}_2$ valued expression. Note that we have dropped the condensation defects for simplicity in the above expression.

Now let us discuss the structure of $\mathfrak{B}_{\mathrm{Neu}}^{\mathfrak{T}_{\mathbb{Z}_2}}$ BCs. Here $\mathfrak{T}_{\mathbb{Z}_2}$ is a $\mathbb{Z}_2^{(0)}$ symmetric 3d TFT whose underlying 3d TFT, that we label as $\mathfrak{T}$, is irreducible. Let $\mathcal{M}$ be the MTC characterizing the underlying 3d TFT $\mathfrak{T}$. A $\mathbb{Z}_2$ 0-form symmetry of $\mathfrak{T}$ is generated by an invertible surface defect $D_2$ of order 2 in $\mathfrak{T}$. The surface defects of an irreducible 3d TFT are all obtained as condensation defects of the line operators. As a consequence, they can all end topologically along some non-genuine topological line defects. In fact, a condensation defect can be completely characterized in terms of the properties of the non-genuine line operators living at its end. Thus in order to characterize a $\mathbb{Z}_2^{(0)}$ symmetry of $\mathfrak{T}$, we consider topological line defects living at the end of $D_2$, which form a category $\mathcal{M}_P$. The total category

$$\mathcal{M}_{\mathbb{Z}_2}^\times = \mathcal{M} \oplus \mathcal{M}_P \tag{3.26}$$

is a $\mathbb{Z}_2$-crossed braided extension of the MTC $\mathcal{M}$, which in particular captures the braiding of the non-genuine lines in $\mathcal{M}_P$ with the genuine lines in $\mathcal{M}$. In conclusion, $\mathfrak{T}_{\mathbb{Z}_2}$ is characterized by a $\mathbb{Z}_2$-crossed braided extension $\mathcal{M}_{\mathbb{Z}_2}^\times$ of $\mathcal{M}$.

The topological line operators of the boundary are obtained from $\mathcal{M}_{\mathbb{Z}_2}^\times$ by performing a $\mathbb{Z}_2$ gauging. The impact of such a gauging on the line operators is captured by a well-known mathematical procedure of $\mathbb{Z}_2$-equivariantization, which yields a new MTC

$$\widetilde{\mathcal{M}} \equiv \left(\mathcal{M}_{\mathbb{Z}_2}^\times\right)^{\mathbb{Z}_2} \tag{3.27}$$

---

[10]The objects of this 2-category are module categories over the MTC $\mathcal{M}$. As such one also uses the notation $\Sigma\mathcal{M} = \mathsf{Mod}(\mathcal{M})$.

from the data of $\mathcal{M}_{\mathbb{Z}_2}^{\times}$. The topological line defects of $\mathfrak{B}_{\mathrm{Neu}}^{\mathfrak{T}_{\mathbb{Z}_2}}$ are described by the MTC $\widetilde{\mathcal{M}}$, however some of these lines are non-genuine lines arising at the end of the bulk surface $\boldsymbol{Q}_2$.

In order to differentiate genuine lines from the non-genuine ones, let us note that there is a special genuine line $D_1$ in $\widetilde{\mathcal{M}}$ which is the projection of the bulk line $\boldsymbol{Q}_1$ onto the boundary. This line is also the generator of the dual $\mathbb{Z}_2$ 1-form symmetry obtained after the above gauging procedure. This line generates a special $\mathsf{Rep}(\mathbb{Z}_2)$ braided fusion sub-category of $\widetilde{\mathcal{M}}$. The entire MTC $\widetilde{\mathcal{M}}$ is $\mathbb{Z}_2$-graded, with the grade being the charge of a line under this $\mathbb{Z}_2$ 1-form symmetry

$$\widetilde{\mathcal{M}} = \widetilde{\mathcal{M}}_1 \oplus \widetilde{\mathcal{M}}_P \,, \tag{3.28}$$

where $\widetilde{\mathcal{M}}_1$ is the trivial grade capturing uncharged lines, and $\widetilde{\mathcal{M}}_P$ is the non-trivial grade capturing charged lines. The lines in $\widetilde{\mathcal{M}}_1$ are genuine, while the lines in $\widetilde{\mathcal{M}}_P$ are non-genuine, arising at the end of $\boldsymbol{Q}_2$.

The symmetry associated to the boundary $\mathfrak{B}_{\mathrm{Neu}}^{\mathfrak{T}_{\mathbb{Z}_2}}$ is thus a possibly non-invertible 1-form symmetry described by the category $\widetilde{\mathcal{M}}_1$. It should be noted that $\widetilde{\mathcal{M}}_1$ is necessarily a non-modular braided fusion category as the subcategory $\mathsf{Rep}(\mathbb{Z}_2) \subseteq \widetilde{\mathcal{M}}_1$ is transparent to all line operators in $\widetilde{\mathcal{M}}_1$. In fact, there are no other lines in $\widetilde{\mathcal{M}}_1$ that have this property. Mathematically, the sub-category formed by transparent lines, known as the Müger center of $\widetilde{\mathcal{M}}_1$, and denoted by $\mathcal{Z}_2(\widetilde{\mathcal{M}}_1)$, is

$$\mathcal{Z}_2(\widetilde{\mathcal{M}}_1) = \mathsf{Rep}(\mathbb{Z}_2) \,. \tag{3.29}$$

Thus the braided fusion categorical symmetries captured by topological BCs of the (3+1)d $\mathbb{Z}_2$ DW theory have the property that their Müger center is exactly $\mathsf{Rep}(\mathbb{Z}_2)$. Physically, this means that the associated possibly non-invertible 1-form symmetry carries an anomaly such that only a $\mathbb{Z}_2^{(1)}$ sub-symmetry is non-anomalous.

The symmetry 2-category associated to $\mathfrak{B}_{\mathrm{Neu}}^{\mathfrak{T}_{\mathbb{Z}_2}}$ is

$$\mathcal{S}\big(\mathfrak{B}_{\mathrm{Neu}}^{\mathfrak{T}_{\mathbb{Z}_2}}\big) = \Sigma\widetilde{\mathcal{M}}_1 \,, \tag{3.30}$$

which includes the condensation surface defects.

In particular, for $\mathfrak{B}_{\mathrm{Neu}}$ we have

$$\mathcal{M} = \mathcal{M}_P = \mathsf{Vec} \,, \tag{3.31}$$

such that the line in $\mathcal{M}_P$ has trivial self-braiding. This translates to

$$\widetilde{\mathcal{M}} = \{D_1^{\mathrm{id}}, D_1^e, D_1^m, D_1^f\} \tag{3.32}$$

being the MTC associated to the toric code with $\mathsf{Rep}(\mathbb{Z}_2)$ generated by the $D_1^e$ line, which is identified with the projection $D_1$ of the bulk line $\boldsymbol{Q}_1$. The trivial and non-trivial grades of $\widetilde{\mathcal{M}}$ are

$$\widetilde{\mathcal{M}}_1 = \{D_1^{\mathrm{id}}, D_1^e\}, \qquad \widetilde{\mathcal{M}}_P = \{D_1^m, D_1^f\}. \tag{3.33}$$

Note that both of the boundary ends $D_1^m$ and $D_1^f$ of $\boldsymbol{Q}_2$ have trivial F-symbol, which is consistent with what we discussed earlier. The symmetry associated to the boundary comes from $\widetilde{\mathcal{M}}_1$ which is clearly a non-anomalous $\mathbb{Z}_2^{(1)}$ symmetry, as the line $D_1^e$ is a boson.

For $\mathfrak{B}_{\mathrm{Neu},\omega}$ we again have

$$\mathcal{M} = \mathcal{M}_P = \mathsf{Vec}, \tag{3.34}$$

but now the line in $\mathcal{M}_P$ has non-trivial self-braiding. This translates to

$$\widetilde{\mathcal{M}} = \{D_1^{\mathrm{id}}, D_1^s, D_1^{\bar{s}}, D_1^{s\bar{s}}\} \tag{3.35}$$

being the MTC associated to the double semion model with $\mathsf{Rep}(\mathbb{Z}_2)$ generated by the $D_1^{s\bar{s}}$ line, which is identified with the projection $D_1$ of the bulk line $\boldsymbol{Q}_1$. The trivial and non-trivial grades of $\widetilde{\mathcal{M}}$ are

$$\widetilde{\mathcal{M}}_1 = \{D_1^{\mathrm{id}}, D_1^{s\bar{s}}\}, \qquad \widetilde{\mathcal{M}}_P = \{D_1^s, D_1^{\bar{s}}\}. \tag{3.36}$$

Note that both of the boundary ends $D_1^s$ and $D_1^{\bar{s}}$ of $\boldsymbol{Q}_2$ have non-trivial F-symbol, which is consistent with what we discussed earlier. The symmetry associated to the boundary comes from $\widetilde{\mathcal{M}}_1$ which is again a non-anomalous $\mathbb{Z}_2^{(1)}$ symmetry, as the line $D_1^{s\bar{s}}$ is a boson.

For the example (3.14), we have

$$\mathcal{M} = \{D_1^{\mathrm{id}}, D_1^e, D_1^m, D_1^f\} \tag{3.37}$$

being the toric code,

$$\mathcal{M}_P = \{D_1^M, D_1^{\bar{M}}, D_1^{eM}, D_1^{e\bar{M}}\} \tag{3.38}$$

with the braiding of $D_1^e$ and $D_1^M$ being $i$, and the braiding of $D_1^e$ and $D_1^{\bar{M}}$ being $-i$. This results in

$$\widetilde{\mathcal{M}} = \{D_1^{e^i m^j} | 0 \leq i, j \leq 3\}, \tag{3.39}$$

which is the MTC associated to the 3d $\mathbb{Z}_4$ DW theory without twist. The $\mathsf{Rep}(\mathbb{Z}_2)$ sub-category is generated by $D_1^{e^2}$ which is identified with the projection $D_1$ of the bulk line $\boldsymbol{Q}_1$. The trivial and non-trivial grades of $\widetilde{\mathcal{M}}$ are

$$\widetilde{\mathcal{M}}_1 = \{D_1^{e^i}, D_1^{e^i m^2} | 0 \leq i \leq 3\}, \qquad \widetilde{\mathcal{M}}_P = \{D_1^{e^i m}, D_1^{e^i m^3} | 0 \leq i \leq 3\} \tag{3.40}$$

The line $D_1^e \in \widetilde{\mathcal{M}}$ generates a $\mathbb{Z}_4^{(1)}$ symmetry associated to the boundary and the line $D_1^{m^2} \in \widetilde{\mathcal{M}}$ generates a $\mathbb{Z}_2^{(1)}$ symmetry associated to the boundary

$$\mathcal{S}(\mathfrak{B}_{\text{Neu}}^{\mathfrak{T}_{\mathbb{Z}_2}}) = \mathbb{Z}_4^{(1),e} \times \mathbb{Z}_2^{(1),m^2} . \tag{3.41}$$

There is a mixed 't Hooft anomaly between the two 1-form symmetry factors that can be expressed in terms of the background fields as

$$2B_2^e \cup B_2^{m^2} , \tag{3.42}$$

where the anomaly is a $\mathbb{Z}_4$ valued expression.

## 3.5 Symmetries and Boundaries Related by Gauging

A special property of (2+1)d SymTFTs is that all of their 2d topological BCs are related by gauging. In particular, all the symmetries captured by a (2+1)d SymTFT are all gauge related. This property does not extend to (3+1)d SymTFTs, as we now demonstrate with the example of (3+1)d $\mathbb{Z}_2$ DW theory.

Two irreducible boundaries that are gauge related have the property that there exists a topological interface between them. Physically, this interface may be viewed as a Dirichlet BC for the gauge fields used to transform one of the boundaries into the other. We now ask which of the boundaries $\mathfrak{B}_{\text{Dir}}^{\mathfrak{T}}$ and $\mathfrak{B}_{\text{Neu}}^{\mathfrak{T}_{\mathbb{Z}_2}}$ admit a topological interface to $\mathfrak{B}_{\text{Dir}}$, or in particular which of the symmetries described by (3+1)d $\mathbb{Z}_2$ DW theory are gauge related to $\mathbb{Z}_2$ 0-form symmetry.

Since we have

$$\mathfrak{B}_{\text{Dir}}^{\mathfrak{T}} = \mathfrak{B}_{\text{Dir}} \boxtimes \mathfrak{T} , \tag{3.43}$$

there is a topological interface between $\mathfrak{B}_{\text{Dir}}^{\mathfrak{T}}$ and $\mathfrak{B}_{\text{Dir}}$ if and only if the 3d TFT $\mathfrak{T}$ admits a 2d topological BC. For this to happen, the MTC $\mathcal{M}$ associated to $\mathfrak{T}$ must be the center of a fusion category $\mathcal{C}$

$$\mathcal{M} = \mathcal{Z}(\mathcal{C}) . \tag{3.44}$$

Indeed, the possibly non-invertible 1-form symmetry associated to the center of a fusion category is gauge equivalent to trivial symmetry[11], and hence symmetries $2\mathsf{Vec}_{\mathbb{Z}_2} \boxtimes \Sigma(\mathcal{Z}(\mathcal{C}))$ are gauge equivalent to $2\mathsf{Vec}_{\mathbb{Z}_2}$ symmetry.

Thus, boundaries $\mathfrak{B}_{\text{Dir}}^{\mathfrak{T}}$ of Dirichlet type gauge related to $\mathfrak{B}_{\text{Dir}}$ are characterized by centers of fusion categories. The fusion category $\mathcal{C}$ characterizes a specific topological interface between

---

[11] All of these gaugings are of non-faithfully acting symmetries. For example, the toric code is produced by gauging the non-faithfully acting trivial $\mathbb{Z}_2$ symmetry of the the trivial 3d TFT.

$\mathfrak{B}_{\mathrm{Dir}}^{\mathfrak{T}}$ and $\mathfrak{B}_{\mathrm{Dir}}$, which has the property that the genuine topological lines living on the interface form the fusion category $\mathcal{C}$. All of the examples of Dirichlet type BCs that we discussed above are of this kind. For $\mathfrak{B}_{\mathrm{Dir}}$, we have $\mathcal{C} = \mathsf{Vec}$, and for the example (3.9), we have $\mathcal{C} = \mathsf{Vec}_{\mathbb{Z}_2}$. The fact that the symmetry (3.24) with anomaly (3.25) is gauge related to $\mathbb{Z}_2^{(0)}$ can be seen by simply gauging the $\mathbb{Z}_2^{(1),e}$ symmetry, which forces the line $D_1^m$ generating $\mathbb{Z}_2^{(1),m}$ symmetry to lie in twisted sector of the dual $\mathbb{Z}_2^{(0)}$ symmetry obtained after gauging, due to the anomaly (3.25). This means that the dual $\mathbb{Z}_2^{(0)}$ symmetry acts non-faithfully and hence can be ignored, and we are left only with the faithfully acting $\mathbb{Z}_2^{(0)}$ symmetry factor of (3.24). In the reverse direction, the boundary (3.9) can be produced from $\mathfrak{B}_{\mathrm{Dir}}$ by gauging a non-faithfully acting $\mathbb{Z}_2^{(0)}$ symmetry on $\mathfrak{B}_{\mathrm{Dir}}$.

As for an example of a Dirichlet type boundary which is not gauge related to the bare Dirichlet BC, we can choose $\mathfrak{T}$ to be the $U(1)_2$ CS theory, for which we have $\mathcal{M} = \{D_1^{\mathrm{id}}, D_1^s\}$, namely the MTC associated to the semion model. This MTC is not the center of a fusion category. The associated symmetry is

$$\mathcal{S} = \mathbb{Z}_2^{(0)} \times \mathbb{Z}_2^{(1)}, \tag{3.45}$$

with a 't Hooft anomaly for $\mathbb{Z}_2^{(1)}$ characterized by the background field as

$$\mathcal{P}(B_2), \tag{3.46}$$

which is $\mathbb{Z}_4$ valued, where $\mathcal{P}(B_2)$ is the Pontryagin square. Clearly, we cannot gauge away such a $\mathbb{Z}_2^{(1)}$ symmetry due to the presence of this 't Hooft anomaly.

Let us now consider Neumann type BCs and analyze when they are gauge related to $\mathfrak{B}_{\mathrm{Dir}}$. From the definition (3.10), note that we can ignore the diagonal $\mathbb{Z}_2^{(0)}$ gauging, and can analyze when $\mathfrak{B}_{\mathrm{Dir}} \boxtimes \mathfrak{T}_{\mathbb{Z}_2}$ is gauge related to $\mathfrak{B}_{\mathrm{Dir}}$. Again by the same arguments as above, this is possible only when the 3d TFT $\mathfrak{T}$ underlying the $\mathbb{Z}_2$ symmetric 3d TFT $\mathfrak{T}_{\mathbb{Z}_2}$ admits a topological boundary condition, in which case the MTC $\mathcal{M}$ characterizing it is a center $\mathcal{Z}(\mathcal{C})$ for some fusion category $\mathcal{C}$

$$\mathcal{M} = \mathcal{Z}(\mathcal{C}). \tag{3.47}$$

In such a situation, a choice of $\mathbb{Z}_2^{(0)}$ symmetry of $\mathfrak{T}$ is encoded in an $\mathbb{Z}_2$-graded fusion category $\widetilde{\mathcal{C}}$ whose trivial grade is $\mathcal{C}$ and the non-trivial grade is denoted as $\mathcal{C}_P$

$$\widetilde{\mathcal{C}} = \mathcal{C} \oplus \mathcal{C}_P. \tag{3.48}$$

The $\mathbb{Z}_2$-graded fusion category $\widetilde{\mathcal{C}}$ describes genuine and non-genuine topological lines living on a specific topological interface between $\mathfrak{B}_{\mathrm{Neu}}^{\mathfrak{T}_{\mathbb{Z}_2}}$ and $\mathfrak{B}_{\mathrm{Dir}}$. The lines living in the trivial grade $\mathcal{C}$

are genuine, while the lines living in the non-trivial grade $\mathcal{C}_P$ are non-genuine as they lie at an end of the topological defect $D_2$ of $\mathfrak{B}_{\mathrm{Dir}}$ along the interface, where recall that $D_2$ is the topological surface arising from projection of the bulk surface $\boldsymbol{Q}_2$ on $\mathfrak{B}_{\mathrm{Dir}}$.

We can also recover the information of the MTC $\widetilde{\mathcal{M}}$ capturing genuine and non-genuine topological line defects of $\mathfrak{B}_{\mathrm{Neu}}^{\mathfrak{T}_{\mathbb{Z}_2}}$ as the center of $\widetilde{\mathcal{C}}$

$$\widetilde{\mathcal{M}} = \mathcal{Z}(\widetilde{\mathcal{C}}), \tag{3.49}$$

The line defect $D_1$ generating the $\mathsf{Rep}(\mathbb{Z}_2)$ subcategory of $\widetilde{\mathcal{M}}$, where recall that $D_1$ is the projection of the bulk line $\boldsymbol{Q}_1$ on $\mathfrak{B}_{\mathrm{Neu}}^{\mathfrak{T}_{\mathbb{Z}_2}}$, admits a simple description as an object of the Drinfeld center $\mathcal{Z}(\widetilde{\mathcal{C}})$. $D_1 \in \mathcal{Z}(\widetilde{\mathcal{C}})$ is an object whose projection $Z(D_1) \in \widetilde{\mathcal{C}}$ is the unit object in $\widetilde{\mathcal{C}}$, i.e. the line $D_1$ vanishes when it is projected onto the interface with $\mathfrak{B}_{\mathrm{Dir}}$. Moreover, its half-braiding with genuine lines in $\mathcal{C}$ living on the interface is $+1$ and with non-genuine lines in $\mathcal{C}_P$ living on the interface is $-1$. The genuine lines in $\widetilde{\mathcal{M}}_1$ on $\mathfrak{B}_{\mathrm{Neu}}^{\mathfrak{T}_{\mathbb{Z}_2}}$ are characterized by the property that they project to a genuine line in $\mathcal{C}$ on the interface, while the non-genuine lines in $\widetilde{\mathcal{M}}_P$ on $\mathfrak{B}_{\mathrm{Neu}}^{\mathfrak{T}_{\mathbb{Z}_2}}$ are characterized by the property that they project to a non-genuine line in $\mathcal{C}_P$ on the interface.

The mathematical relationship between various MTCs and fusion categories in the above situation is summarized in the following diagram

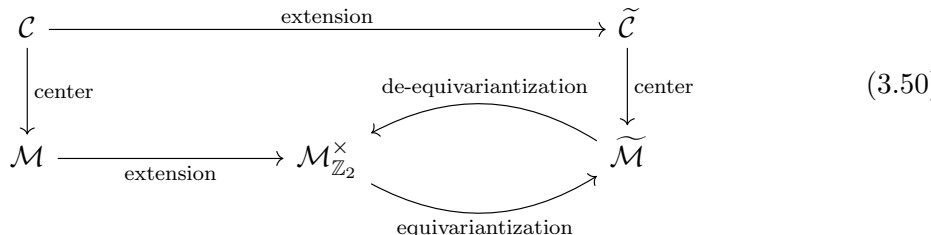

$$\tag{3.50}$$

Thus, boundaries $\mathfrak{B}_{\mathrm{Neu}}^{\mathfrak{T}_{\mathbb{Z}_2}}$ of Neumann type gauge related to $\mathfrak{B}_{\mathrm{Dir}}$ are characterized by centers of $\mathbb{Z}_2$-graded fusion categories. All of the examples of Neumann type BCs that we discussed above are of this kind. For $\mathfrak{B}_{\mathrm{Neu}}$ we have $\widetilde{\mathcal{C}} = \mathsf{Vec}_{\mathbb{Z}_2}$, and for $\mathfrak{B}_{\mathrm{Neu},\omega}$ we have $\widetilde{\mathcal{C}} = \mathsf{Vec}_{\mathbb{Z}_2}^\omega$ where we have a non-trivial associator. Indeed, these two boundaries are obtained from $\mathfrak{B}_{\mathrm{Dir}}$ by gauging the $\mathbb{Z}_2^{(0)}$ symmetry without and with discrete torsion respectively. On the other hand, for the example (3.14) we have $\widetilde{\mathcal{C}} = \mathsf{Vec}_{\mathbb{Z}_4}$. Indeed, this boundary is obtained by gauging a $\mathbb{Z}_4^{(0)}$ symmetry, whose $\mathbb{Z}_2$ subgroup acts non-faithfully, and whose generator is identified with the order 2 topological defect $D_2$ on $\mathfrak{B}_{\mathrm{Dir}}$. The dual $\mathbb{Z}_4^{(1)}$ symmetry is identified with $\mathbb{Z}_4^{(1),e}$ in (3.41), and we obtain the $\mathbb{Z}_2^{(1),m^2}$ factor from the flux sector of the $\mathbb{Z}_2^{(0)}$ subgroup of $\mathbb{Z}_4^{(0)}$ acting non-faithfully on $\mathfrak{B}_{\mathrm{Dir}}$. The line coming from flux sector is charged under the dual 1-form symmetry, which is captured by the anomaly (3.42). Examples of Neumann type BCs not gauge related to $\mathfrak{B}_{\mathrm{Dir}}$ are discussed in the next subsection.

## 3.6 More Examples of Neumann Non-Minimal Boundary Conditions

In this subsection, we discuss a few more examples of non-minimal BCs of Neumann type.

**BCs Gauge Related to $\mathfrak{B}_{\mathrm{Dir}}$.** First we discuss some BCs gauge related to $\mathfrak{B}_{\mathrm{Dir}}$. Recall that all such Neumann BCs are characterized by $\mathbb{Z}_2$ graded fusion categories. $\widetilde{\mathcal{C}} = \mathsf{Vec}_G$ for a finite possibly non-abelian group $G$ provides an example of a $\mathbb{Z}_2$ graded fusion category once we specify a surjective homomorphism

$$\rho : \; G \to \mathbb{Z}_2 = \{1, P\}\,. \tag{3.51}$$

The topological line defects on the corresponding boundary $\mathfrak{B}_{\mathrm{Neu}}^{\mathfrak{T}_{\mathbb{Z}_2}}$ are characterized by the MTC

$$\widetilde{\mathcal{M}} = \mathcal{Z}(\mathsf{Vec}_G)\,, \tag{3.52}$$

whose simple topological lines can be labeled as

$$D_1^{[g],R} \in \mathcal{Z}(\mathsf{Vec}_G)\,, \tag{3.53}$$

where $[g]$ is a conjugacy class in $G$ and $R$ is representation of the centralizer subgroup $H_g \subseteq G$ of an element $g \in [g]$. The genuine lines are

$$\widetilde{\mathcal{M}}_1 = \{D_1^{[g],R} | \rho(g) = 1\} \tag{3.54}$$

and the non-genuine lines are

$$\widetilde{\mathcal{M}}_P = \{D_1^{[g],R} | \rho(g) = P\} \tag{3.55}$$

with the projection $D_1$ of bulk line $\boldsymbol{Q}_1$ being identified as

$$D_1 = D_1^{[\mathrm{id}],R_\rho}\,, \tag{3.56}$$

where $R_\rho$ is the 1d sign representation of $G$ with respect to $\rho$, i.e. the action of $g \in G$ is $+1$ if $\rho(g) = 1$ and the action of $g \in G$ is $-1$ if $\rho(g) = P$.

   The corresponding 3d TFT $\mathfrak{T}_{\mathbb{Z}_2}$ is a 3d DW gauge theory based on the gauge group $H_\rho$ (without twist), where $H_\rho \subseteq G$ is the kernel of $\rho$. Indeed the topological lines of this 3d theory are described by the center $\mathcal{Z}(\mathcal{C})$ of the trivial grade

$$\mathcal{C} = \mathsf{Vec}_H \tag{3.57}$$

of the $\mathbb{Z}_2$-graded fusion category $\widetilde{\mathcal{C}} = \mathsf{Vec}_G$.

For an example of a Neumann type BC not based on groups, we can consider the $\mathbb{Z}_2$ graded fusion category

$$\widetilde{\mathcal{C}} = \mathsf{Ising}\,, \tag{3.58}$$

whose trivial and non-trivial grades are

$$\mathcal{C} = \{D_1^{\mathrm{id}}, D_1^{\psi}\} = \mathsf{Vec}_{\mathbb{Z}_2}\,, \qquad \mathcal{C}_P = \{D_1^{\sigma}\}\,. \tag{3.59}$$

The genuine and non-genuine lines on the boundary $\mathfrak{B}_{\mathrm{Neu}}^{\mathfrak{T}_{\mathbb{Z}_2}}$ are the same as for the doubled Ising 3d TFT. The genuine lines can be labeled as

$$\widetilde{\mathcal{M}}_1 = \{D_1^{\mathrm{id}}, D_1^{\psi\bar{\psi}}, D_1^{\psi}, D_1^{\bar{\psi}}, D_1^{\sigma\bar{\sigma}}\}\,. \tag{3.60}$$

The fusions of the four lines

$$D_1^{\mathrm{id}}, D_1^{\psi\bar{\psi}}, D_1^{\psi}, D_1^{\bar{\psi}} \tag{3.61}$$

are described by the group $\mathbb{Z}_2 \times \mathbb{Z}_2$. The fusion of any of these four lines with $D_1^{\sigma\bar{\sigma}}$ gives back $D_1^{\sigma\bar{\sigma}}$, which means that the latter line is non-invertible. Finally the self-fusion of the non-invertible line is

$$\left(D_1^{\sigma\bar{\sigma}}\right)^2 = D_1^{\mathrm{id}} \oplus D_1^{\psi\bar{\psi}} \oplus D_1^{\psi} \oplus D_1^{\bar{\psi}}\,. \tag{3.62}$$

Thus we may recognize $\widetilde{\mathcal{M}}_1$ as a braided fusion category whose underlying fusion category is Tambara-Yamagami $\mathsf{TY}_{\mathbb{Z}_2 \times \mathbb{Z}_2}$ category. We can say that the symmetry associated to the boundary is a "non-invertible $\mathsf{TY}_{\mathbb{Z}_2 \times \mathbb{Z}_2}$ 1-form symmetry" which may be denoted as

$$\mathcal{S}(\mathfrak{B}_{\mathrm{Neu}}^{\mathfrak{T}_{\mathbb{Z}_2}}) = \mathsf{TY}_{\mathbb{Z}_2 \times \mathbb{Z}_2}^{(1)}\,, \tag{3.63}$$

whose invertible 1-form sub-symmetry is $\mathbb{Z}_2^{(1)} \times \mathbb{Z}_2^{(1)}$. The braidings on $\widetilde{\mathcal{M}}_1$ capture a certain 't Hooft anomaly of $\mathsf{TY}_{\mathbb{Z}_2 \times \mathbb{Z}_2}^{(1)}$ non-invertible 1-form symmetry. Restricted to the $\mathbb{Z}_2^{(1)} \times \mathbb{Z}_2^{(1)}$ invertible 1-form sub-symmetry this 't Hooft anomaly can be expressed as

$$B_2^{\psi} \cup B_2^{\psi}\,, \tag{3.64}$$

where $B_2^{\psi}$ is background field for $\mathbb{Z}_2^{(1)}$ symmetry generated by $D_1^{\psi}$, where we have chosen the other $\mathbb{Z}_2^{(1)}$ symmetry to be generated by $D_1^{\psi\bar{\psi}}$. In fact, $D_1^{\psi\bar{\psi}}$ is completely transparent to all the $\widetilde{\mathcal{M}}_1$ lines, and hence is identified with the projection $D_1$ of the bulk line $\boldsymbol{Q}_1$

$$D_1 = D_1^{\psi\bar{\psi}}\,. \tag{3.65}$$

The underlying 3d TFT of the $\mathbb{Z}_2$ symmetric 3d TFT $\mathfrak{T}_{\mathbb{Z}_2}$ has lines described by

$$\mathcal{Z}(\mathcal{C}) = \mathcal{Z}(\mathsf{Vec}_{\mathbb{Z}_2}) \tag{3.66}$$

and hence can be identified as the 3d DW $\mathbb{Z}_2$ gauge theory without twist. The $\mathbb{Z}_2^{(0)}$ symmetry of $\mathfrak{T}_{\mathbb{Z}_2}$ is the electric magnetic duality symmetry that exchanges the lines $D_1^e$ and $D_1^m$ in $\mathcal{Z}(\mathsf{Vec}_{\mathbb{Z}_2})$, and the surface defect implementing this $\mathbb{Z}_2^{(0)}$ symmetry can be obtained as a condensation defect for $D_1^f$.

**BCs Not Gauge Related to $\mathfrak{B}_{\mathbf{Dir}}$.**  Now we discuss Neumann type BCs not gauge related to $\mathfrak{B}_{\mathrm{Dir}}$. Such BCs are described by MTCs $\widetilde{\mathcal{M}}$ that are not centers of fusion categories. Moreover, recall that $\widetilde{\mathcal{M}}$ must contain $\mathsf{Rep}(\mathbb{Z}_2)$ as a braided fusion subcategory.

In the examples discussed here, we additionally require that the 3d TFT associated to $\widetilde{\mathcal{M}}$ has vanishing gravitational 't Hooft anomaly, otherwise the corresponding boundary $\mathfrak{B}_{\mathrm{Neu}}^{\mathfrak{T}_{\mathbb{Z}_2}}$ is actually a topological interface between (3+1)d $\mathbb{Z}_2$ DW theory and an invertible (3+1)d TFT characterizing the gravitational anomaly.

We can construct an infinite class of such examples using abelian Chern Simons theories. The abelian Chern Simons theory $U(1)_k$ defines a bosonic TFT when $k$ is a positive even integer. It has chiral central charge $c_- = 1$ and abelian anyons are labeled by $a = 0, 1, \ldots, k-1$, forming a fusion ring $\mathbb{Z}_k$. The topological spins of the anyons are

$$h_a = \frac{a^2}{2k} \quad \mathrm{mod}\ 1 \tag{3.67}$$

and their braidings are

$$B_{ab} = \exp\left(2\pi i \frac{ab}{k}\right) . \tag{3.68}$$

The one-form symmetry $\mathbb{Z}_2 \subset \mathbb{Z}_k$ generated by the abelian anyon $a = k/2$ is non-anomalous if and only if $k$ is a multiple of 4. We denote the time reversal conjugate of $U(1)_k$ to be $U(1)_{-k}$.

We can use the above MTC to build MTCs with vanishing central charges by considering

$$\widetilde{\mathcal{M}} = U(1)_{2n} \times U(1)_{-2m}, \quad n, m \in \mathbb{Z}_{>0} . \tag{3.69}$$

This is a TFT with $4nm$ abelian anyons forming the group $\mathcal{A} = \mathbb{Z}_{2n} \times \mathbb{Z}_{2m}$ under fusion. Gapped boundary conditions of such a TFT are in 1-to-1 correspondence with Lagrangian subgroups $L$ of $\mathcal{A}$, which do not exist unless $nm$ is a perfect square since $|\mathcal{A}| = |L|^2$. The non-existence of topological BCs of these 3d TFTs implies that such MTCs $\widetilde{\mathcal{M}}$ cannot be expressed as centers of fusion categories. We assume that $nm$ is not a perfect square from this point onwards.

We also need to choose a non-anomalous $\mathbb{Z}_2$ 1-form symmetry in $\widetilde{\mathcal{M}}$. We pick this to be the diagonal $\mathbb{Z}_2$ one form symmetry generated by the anyon $(n, m)$. Since the spin of this anyon is $h = \frac{n-m}{4}$, requiring that this $\mathbb{Z}_2$ 1-form symmetry is anomalous imposes the condition $n - m \in 4\mathbb{Z}$, which we assume to hold from this point onwards.

The braiding of the anyon $(n, m)$ with anyon $(a, b)$ is given by the phase

$$B_{(n,m),(a,b)} = \exp 2\pi i \left( \frac{a - b}{2} \right), \tag{3.70}$$

which equips $\widetilde{\mathcal{M}}$ with a $\mathbb{Z}_2$ grading. The neutral component $\widetilde{\mathcal{M}}_1$ under the $\mathbb{Z}_2$ one form symmetry is given by $(a, b)$ with $a - b \in 2\mathbb{Z}$. The symmetry 2-category associated to $\mathfrak{B}_{\mathrm{Neu}}^{\mathfrak{T}_{\mathbb{Z}_2}}$ is thus the condensation of $\widetilde{\mathcal{M}}_1$

$$\mathcal{S}\big( \mathfrak{B}_{\mathrm{Neu}}^{\mathfrak{T}_{\mathbb{Z}_2}} \big) = \Sigma \widetilde{\mathcal{M}}_1, \tag{3.71}$$

The MTC $\mathcal{M}$ describing the 3d TFT underlying the $\mathbb{Z}_2$ symmetric 3d TFT $\mathfrak{T}_{\mathbb{Z}_2}$ involved in the construction (3.10) of the BC $\mathfrak{B}_{\mathrm{Neu}}^{\mathfrak{T}_{\mathbb{Z}_2}}$ is obtained by gauging the non-anomalous $\mathbb{Z}_2$ 1-form symmetry generated by the anyon $(n, m)$ in $\widetilde{\mathcal{M}}$. After condensing the bosonic abelian anyon $(n, m)$, we obtain the following anyon content

$$\{(a, b), \quad 0 \leq a < n, 0 \leq b < 2m, a - b \in 2\mathbb{Z}\}, \tag{3.72}$$

with the fusion rules

$$(0, 2)^m = 1, \quad (1, 1)^n = (0, 2)^{\frac{n+m}{2}}. \tag{3.73}$$

Looking closely at the spins and the braiding, one can recognize that it coincides with the abelian TFT described by the following K-matrix

$$K = \begin{pmatrix} \frac{n-m}{2} & -\frac{m+n}{2} \\ -\frac{m+n}{2} & \frac{n-m}{2} \end{pmatrix}. \tag{3.74}$$

As a sanity check, the diagonal element $\frac{n-m}{2}$ is indeed an even integer as required for the TFT to be bosonic, which is a consequence of the anomaly vanishing condition $n - m \in 4\mathbb{Z}$.

# 4 Gapped Phases from $\mathcal{Z}(2\mathsf{Vec}_{\mathbb{Z}_2})$

Now that we have classified the topological boundary conditions of the (3+1)d $\mathbb{Z}_2$ DW theory, we can construct all (2+1)d gapped phases having symmetries carried by these topological BCs. This is done via the sandwich construction whose bulk comprises of the (3+1)d TFT, with the symmetry boundary $\mathfrak{B}^{\mathrm{sym}}$ taken to be a BC realizing the desired symmetry fusion 2-category $\mathcal{S}$. Feeding in different topological BCs as the physical boundary, and performing the interval compactifications, we obtain all $\mathcal{S}$-symmetric (2+1)d gapped phases. In this section, we will describe the general structure of these gapped phases, along with symmetry realization, for all possible $\mathcal{S}$ whose SymTFT $\mathfrak{Z}(\mathcal{S})$ is the (3+1)d $\mathbb{Z}_2$ DW theory.

Broadly speaking, since we have two types of boundary conditions – Dirichlet type

$$\mathfrak{B}_{\mathrm{Dir}}^{\mathfrak{T}} \equiv \mathfrak{B}_{\mathrm{Dir}} \boxtimes \mathfrak{T}, \tag{4.1}$$

labelled by a choice of 3d TFT $\mathfrak{T}$, and Neumann type

$$\mathfrak{B}_{\text{Neu}}^{\mathfrak{T}_{\mathbb{Z}_2}} \equiv \frac{\mathfrak{B}_{\text{Dir}} \boxtimes \mathfrak{T}_{\mathbb{Z}_2}}{\mathbb{Z}_2^{(0)}} \, , \tag{4.2}$$

labelled by a choice of $\mathbb{Z}_2^{(0)}$ symmetric 3d TFT $\mathfrak{T}_{\mathbb{Z}_2}$, we have three different types of sandwiches.

- Dirichlet-Dirichlet type $(\mathfrak{B}_{\text{Dir}}^{\mathfrak{T}}, \mathfrak{B}_{\text{Dir}}^{\mathfrak{T}'})$

- Dirichlet-Neumann type $(\mathfrak{B}_{\text{Dir}}^{\mathfrak{T}}, \mathfrak{B}_{\text{Neu}}^{\mathfrak{T}_{\mathbb{Z}_2}'})$

- Neumann-Neumann type $(\mathfrak{B}_{\text{Neu}}^{\mathfrak{T}_{\mathbb{Z}_2}}, \mathfrak{B}_{\text{Neu}}^{\mathfrak{T}_{\mathbb{Z}_2}'})$

However, as we will see, depending on the choice of symmetry vs physical boundary, and depending on whether the boundaries are minimal or non-minimal, these three types of sandwiches lead to a rather rich structure of symmetric (2+1)d gapped phases.

## 4.1 $\mathbb{Z}_2$ 0-form Symmetry

We begin by choosing the symmetry boundary to be the Dirichlet BC

$$\mathfrak{B}^{\text{sym}} = \mathfrak{B}_{\text{Dir}} \, , \tag{4.3}$$

in which case the symmetry under consideration is a non-anomalous $\mathbb{Z}_2$ 0-form symmetry described by the fusion 2-category

$$\mathcal{S} = 2\mathsf{Vec}_{\mathbb{Z}_2} \, . \tag{4.4}$$

### 4.1.1 Minimal Phases

We have to now choose a physical boundary. Let us begin by choosing each of the minimal BCs to be the physical boundary.

$\mathbb{Z}_2^{(0)}$ **SSB Phase.** First of all, consider the case

$$\mathfrak{B}^{\text{phys}} = \mathfrak{B}_{\text{Dir}} \, . \tag{4.5}$$

The structure of the resulting (2+1)d gapped phase is easy to work out. Note that the bulk line $\boldsymbol{Q}_1$ can end on both boundaries, giving rise to a non-trivial topological local operator after the interval compactification:

$$\tag{4.6}$$

This means that the resulting 3d TFT has two vacua, differentiated by the vev of $D_0$, that are permuted by the $\mathbb{Z}_2^{(0)}$ symmetry. The interval compactification does not produce any genuine topological line operators, and hence we conclude that both vacua must be described by trivial 3d TFTs, and we denote the result of the interval compactification as

$$\text{Triv} \oplus \text{Triv} \,. \tag{4.7}$$

This is simply the standard Landau SSB phase for $\mathbb{Z}_2^{(0)}$ symmetry.

$\mathbb{Z}_2^{(0)}$ **Trivial Phase.**  Now consider the case

$$\mathfrak{B}^{\text{phys}} = \mathfrak{B}_{\text{Neu}} \,. \tag{4.8}$$

The bulk line $\boldsymbol{Q}_1$ ends on $\mathfrak{B}^{\text{sym}}$ but not on $\mathfrak{B}^{\text{phys}}$, and the bulk surface $\boldsymbol{Q}_2$ ends on $\mathfrak{B}^{\text{phys}}$ but not on $\mathfrak{B}^{\text{sym}}$.

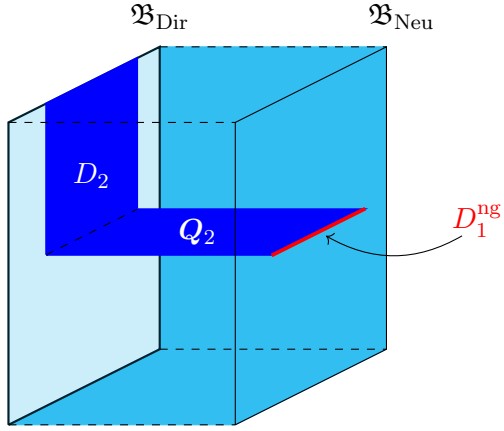

$$\tag{4.9}$$

The interval compactification thus does not produce any genuine topological defects, and we therefore obtain a trivial 3d TFT

$$\text{Triv} \,. \tag{4.10}$$

The $\mathbb{Z}_2$ 0-form symmetry of this 3d TFT is generated by a surface $D_2$ which can end in a topological line $D_1^{\text{ng}}$ arising from the end of the bulk surface $\boldsymbol{Q}_2$ along $\mathfrak{B}^{\text{phys}}$. See Figure 2.

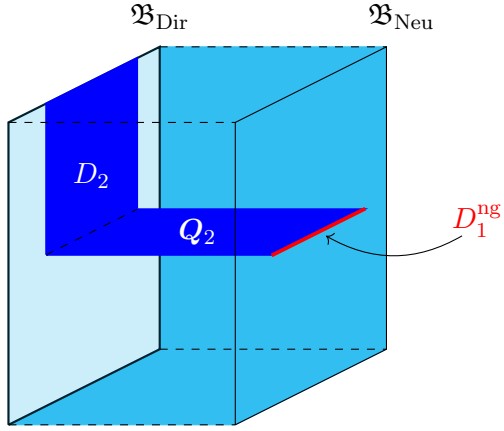

Figure 2: The surface $\boldsymbol{Q}_2$ in the SymTFT bulk is projected onto $\mathfrak{B}_{\text{Dir}}$ and ends along $\mathfrak{B}_{\text{Neu}}$, giving rise, after interval compactification, to a non-genuine boundary line $D_1^{\text{ng}}$. Recall that the bulk of the SymTFT is (3+1) dimensional and its boundaries are (2+1) dimensional, so on the left boundary $D_2$ is a 2-dimensional surface in (2+1)d.

From the property of $\mathfrak{B}_{\text{Neu}}$, we know that $D_1^{\text{ng}}$ has trivial F-symbols, which implies that the $\mathbb{Z}_2$ 0-form symmetry is realized trivially on this trivial 3d TFT. In other words, we obtain the standard Landau trivial phase for $\mathbb{Z}_2^{(0)}$ symmetry, in which the $\mathbb{Z}_2^{(0)}$ symmetry remains spontaneously unbroken.

**$\mathbb{Z}_2^{(0)}$ SPT Phase.**   Finally consider the case

$$\mathfrak{B}^{\text{phys}} = \mathfrak{B}_{\text{Neu}}^{\omega} \,. \tag{4.11}$$

By the same arguments as above, the 3d TFT obtained after interval compactification

$$\text{(4.12)}$$

is the trivial 3d TFT

$$\text{Triv} \,. \tag{4.13}$$

However, now from the property of $\mathfrak{B}_{\text{Neu}}^{\omega}$, we know that $D_1^{\text{ng}}$ has non-trivial F-symbols. Thus the $\mathbb{Z}_2^{(0)}$ symmetry is realized non-trivially on the trivial 3d TFT. In fact, we obtain the SPT phase for $\mathbb{Z}_2^{(0)}$ symmetry.

### 4.1.2   Non-Minimal Phases

We now consider non-minimal BCs as physical boundaries.

**$\mathbb{Z}_2^{(0)}$ Spontaneously Broken.**   Let us choose the physical boundary to be a general boundary of Dirichlet type

$$\mathfrak{B}^{\text{phys}} = \mathfrak{B}_{\text{Dir}}^{\mathfrak{T}} = \mathfrak{B}_{\text{Dir}} \boxtimes \mathfrak{T} \,, \tag{4.14}$$

characterized by an irreducible 3d TFT $\mathfrak{T}$ whose topological lines form an MTC $\mathcal{M}$. Since $\mathfrak{B}_{\text{Dir}}^{\mathfrak{T}}$ is simply obtained by stacking $\mathfrak{T}$ with $\mathfrak{B}_{\text{Dir}}$, the 3d TFT resulting from the interval compactification with $\mathfrak{B}^{\text{phys}} = \mathfrak{B}_{\text{Dir}}^{\mathfrak{T}}$ is obtained by stacking $\mathfrak{T}$ with the 3d TFT resulting from the interval compactification with $\mathfrak{B}^{\text{phys}} = \mathfrak{B}_{\text{Dir}}$, equation (4.7). Thus, the result of the interval compactification is

$$(\text{Triv} \oplus \text{Triv}) \boxtimes \mathfrak{T} = \mathfrak{T} \oplus \mathfrak{T} \,, \tag{4.15}$$

i.e. we have two vacua, with each vacuum carrying topological order described by the irreducible 3d TFT $\mathfrak{T}$. The $\mathbb{Z}_2^{(0)}$ symmetry is spontaneously broken as it exchanges the two $\mathfrak{T}$ vacua.

Note that we have topological lines described by the MTC $\mathcal{M}$ in each vacuum. These lines should be regarded as describing a (possibly non-invertible) emergent 1-form symmetry arising in the $\mathbb{Z}_2^{(0)}$ symmetric 3d gapped phase under discussion. That is, the $\mathcal{M}$ 1-form symmetry emerges in the IR of a $\mathbb{Z}_2^{(0)}$ symmetric (2+1)d system not having $\mathcal{M}$ symmetry in the UV but lying in this gapped phase.

Out of these, the non-minimal phases admitting a $\mathbb{Z}_2^{(0)}$ symmetric gapped boundary condition to a minimal phase are characterized by the condition

$$\mathcal{M} = \mathcal{Z}(\mathcal{C}) \,, \tag{4.16}$$

i.e. the MTC $\mathcal{M}$ being the center of a fusion category $\mathcal{C}$.

$\mathbb{Z}_2^{(0)}$ **Spontaneously Unbroken.** Let us choose the physical boundary to be a general boundary of Neumann type

$$\mathfrak{B}^{\text{phys}} = \mathfrak{B}_{\text{Neu}}^{\mathfrak{T}_{\mathbb{Z}_2}} \tag{4.17}$$

characterized by a $\mathbb{Z}_2^{(0)}$ symmetric irreducible 3d TFT $\mathfrak{T}_{\mathbb{Z}_2}$. Here we know from the discussion of section 3.3 that the result of the sandwich construction is the $\mathbb{Z}_2^{(0)}$ symmetric 3d TFT $\mathfrak{T}_{\mathbb{Z}_2}$. The 3d TFT $\mathfrak{T}$ underlying $\mathfrak{T}_{\mathbb{Z}_2}$ has topological lines forming an MTC $\mathcal{M}$, which forms the emergent 1-form symmetry associated to the $\mathbb{Z}_2^{(0)}$ symmetric (2+1)d gapped phase under discussion. The $\mathbb{Z}_2^{(0)}$ symmetry interacts non-trivially with this emergent symmetry. In particular, it can exchange some of the simple lines in $\mathcal{M}$, and its action may be fractionalized on lines in $\mathcal{M}$ left invariant by the $\mathbb{Z}_2^{(0)}$ action. These properties are captured in terms of a $\mathbb{Z}_2$ crossed braided fusion category $\mathcal{M}_{\mathbb{Z}_2}^{\times}$ extending the MTC $\mathcal{M}$, which specifies completely how the $\mathbb{Z}_2^{(0)}$ symmetry is realized on $\mathfrak{T}$. Since we have a single vacuum, and the $\mathbb{Z}_2^{(0)}$ symmetry acts within this vacuum, this is a (non-Landau) phase in which $\mathbb{Z}_2^{(0)}$ remains spontaneously unbroken.

Out of these, the non-minimal phases admitting a $\mathbb{Z}_2^{(0)}$ symmetric gapped boundary condition to a minimal phase are characterized by the condition

$$\mathcal{M} = \mathcal{Z}(\mathcal{C}) \,, \tag{4.18}$$

i.e. the MTC $\mathcal{M}$ being the center of a fusion category $\mathcal{C}$. An example of such a non-minimal phase is toric code with $\mathbb{Z}_2^{(0)}$ symmetry that does not permute any line but has symmetry fractionalization on $D_1^e$ and $D_1^f$. The associated category $\mathcal{M}_{\mathbb{Z}_2}^{\times}$ is provided in equations (3.37) and (3.38). Another example is toric code with $\mathbb{Z}_2^{(0)}$ symmetry that permutes $D_1^e$ and $D_1^m$. The trivial grade for the associated category $\mathcal{M}_{\mathbb{Z}_2}^{\times}$ is as in (3.37), while the non-trivial grade

is

$$\mathcal{M}_P = \{D_1^\sigma, D_1^{\bar\sigma}\}\,, \tag{4.19}$$

with fusion rules

$$
\begin{aligned}
D_1^e \otimes D_1^\sigma &= D_1^m \otimes D_1^\sigma = D_1^{\bar\sigma}\,, \\
D_1^e \otimes D_1^{\bar\sigma} &= D_1^m \otimes D_1^{\bar\sigma} = D_1^\sigma\,, \\
D_1^\sigma \otimes D_1^\sigma &= D_1^{\bar\sigma} \otimes D_1^{\bar\sigma} = D_1^{\mathrm{id}} \oplus D_1^f\,, \\
D_1^\sigma \otimes D_1^{\bar\sigma} &= D_1^{\bar\sigma} \otimes D_1^\sigma = D_1^e \oplus D_1^m\,.
\end{aligned}
\tag{4.20}
$$

As for non-minimal phases that do not admit such a gapped boundary condition, an example is provided by the 3d TFT with $K$-matrix specified in (3.74), whose $\mathbb{Z}_2^{(0)}$ symmetry is chosen such that upon gauging it we obtain the 3d CS theory $U(1)_{2n} \times U(1)_{-2m}$. See the discussion preceding equation (3.74) for more details.

## 4.2  Non-Anomalous $\mathbb{Z}_2$ 1-form Symmetry

Let us now choose the symmetry boundary to be the Neumann BC

$$\mathfrak{B}^{\mathrm{sym}} = \mathfrak{B}_{\mathrm{Neu}}\,, \tag{4.21}$$

in which case the symmetry under consideration is a non-anomalous $\mathbb{Z}_2$ 1-form symmetry described by the fusion 2-category

$$\mathcal{S} = 2\mathsf{Rep}(\mathbb{Z}_2)\,. \tag{4.22}$$

We could equally well choose the symmetry boundary to be the other Neumann BC $\mathfrak{B}^\omega_{\mathrm{Neu}}$, leading to similar results as below.

### 4.2.1  Minimal Phases

We have to now choose a physical boundary. Let us begin by choosing the minimal BCs to be the physical boundary.

$\mathbb{Z}_2^{(1)}$ **Trivial Phase.**  First of all, consider the case

$$\mathfrak{B}^{\mathrm{phys}} = \mathfrak{B}_{\mathrm{Dir}}\,. \tag{4.23}$$

We have already considered above a sandwich compactification involving $\mathfrak{B}_{\mathrm{Dir}}$ and $\mathfrak{B}_{\mathrm{Neu}}$ around equation (4.9). The resulting 3d TFT is thus the same, only which boundary is taken to be the symmetry boundary and which boundary is taken to be the physical one has been modified.

$$\tag{4.24}$$

Thus, the result of the interval compactification is a trivial 3d TFT

$$\text{Triv}. \tag{4.25}$$

The generator of $\mathbb{Z}_2^{(1)}$ symmetry is a line that can end topologically at a local operator. See Figure 3.

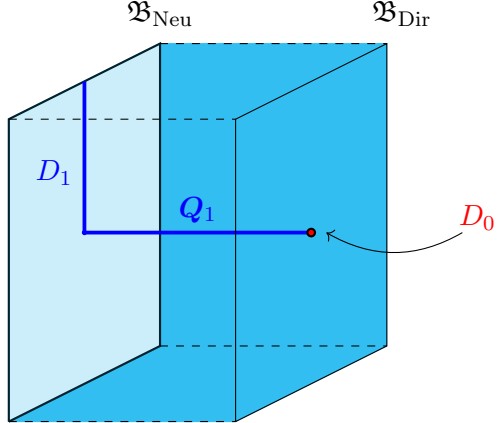

Figure 3: The line $\boldsymbol{Q}_1$ in the SymTFT bulk is projected onto $\mathfrak{B}_{\text{Neu}}$ and ends along $\mathfrak{B}_{\text{Dir}}$, giving rise, after interval compactification, to a boundary line $D_1$ that can end topologically at a local operator $D_0$.

Thus the $\mathbb{Z}_2^{(1)}$ symmetry is realized trivially, and hence we obtain the trivial phase for $\mathbb{Z}_2^{(1)}$ symmetry.

**$\mathbb{Z}_2^{(1)}$ SSB Phase: Non-semionic type.** Now consider the case

$$\mathfrak{B}^{\text{phys}} = \mathfrak{B}_{\text{Neu}}. \tag{4.26}$$

We obtain an order two topological line operator $D_1^e$ in the resulting 3d TFT by incorporating the bulk line $\boldsymbol{Q}_1$ parallel to boundaries in the sandwich construction. This line operator also generates the non-anomalous $\mathbb{Z}_2$ 1-form symmetry of the resulting $(2+1)$d gapped phase. We obtain another order two topological line operator $D_1^m$ in the resulting 3d TFT by compactifying the bulk surface $\boldsymbol{Q}_2$ in between the two boundaries.

$$\tag{4.27}$$

Since $\boldsymbol{Q}_1$ and $\boldsymbol{Q}_2$ have a non-trivial braiding, we learn that $D_1^e$ and $D_1^m$ must also braid non-trivially. Moreover, since the topological lines arising at the ends of $\boldsymbol{Q}_2$ along the $\mathfrak{B}_{\text{Neu}}$ boundaries have trivial F-symbol, we learn that $D_1^m$ also has a trivial F-symbol. Putting it all together, the resulting 3d TFT can be recognized as the 3d $\mathbb{Z}_2$ DW theory without twist, or equivalently as the toric code.

This can be recognized as an SSB phase for $\mathbb{Z}_2^{(1)}$ symmetry. The topological line $D_1^m$ can be identified as the image of a charged line operator that obtains a non-zero vev in such a gapped phase, and provides an emergent IR $\mathbb{Z}_2^{(1)}$ symmetry arising due to the spontaneous symmetry breaking of a UV $\mathbb{Z}_2^{(1)}$ symmetry. The emergent 1-form symmetry has the property that it has a trivial 't Hooft anomaly, reflected in the fact that $D_1^m$ is a boson. The combination

$$D_1^f = D_1^e \otimes D_1^m \tag{4.28}$$

generates another emergent $\mathbb{Z}_2^{(1)}$ symmetry which carries a 't Hooft anomaly described by the generator of the $\mathbb{Z}_2$ subgroup of the group $\mathbb{Z}_4$ of possible 't Hooft anomalies for $\mathbb{Z}_2^{(1)}$ symmetry in 3d. We refer to this $\mathbb{Z}_2^{(1)}$ SSB phase as being of non-semionic type.

$\mathbb{Z}_2^{(1)}$ **SSB Phase: Semionic type.** Finally, consider the case

$$\mathfrak{B}^{\text{phys}} = \mathfrak{B}_{\text{Neu}}^{\omega}. \tag{4.29}$$

Just like the previous case, we obtain an order two topological line operator $D_1^{s\bar{s}}$ in the resulting 3d TFT by incorporating the bulk line $\boldsymbol{Q}_1$ parallel to the boundaries in the sandwich construction. This line operator also generates the non-anomalous $\mathbb{Z}_2$ 1-form symmetry of the resulting (2+1)d gapped phase. We also obtain another order two topological line operator $D_1^s$ by compactifying the bulk surface $\boldsymbol{Q}_2$ in between the two boundaries.

$$\tag{4.30}$$

The non-trivial braiding between $\boldsymbol{Q}_1$ and $\boldsymbol{Q}_2$ implies that $D_1^{s\bar{s}}$ and $D_1^s$ also braid non-trivially. The F-symbol for $D_1^s$ is non-trivial since it comes from the non-trivial F-symbol for the end of $\boldsymbol{Q}_2$ along the $\mathfrak{B}_{\text{Neu}}^{\omega}$ boundary. Putting it all together, the resulting 3d TFT can be recognized as the 3d $\mathbb{Z}_2$ DW theory with twist, or equivalently as the double semion model.

This is another type of SSB phase for $\mathbb{Z}_2^{(1)}$ symmetry. The topological line $D_1^s$ can be identified as the image of a charged line operator obtaining a non-zero vev in such a gapped

phase, and provides an emergent IR $\mathbb{Z}_2^{(1)}$ symmetry arising due to the spontaneous symmetry breaking of a UV $\mathbb{Z}_2^{(1)}$ symmetry. The emergent 1-form symmetry has the property that it has a non-trivial 't Hooft anomaly described by the generator of the group $\mathbb{Z}_4$ of possible 't Hooft anomalies for $\mathbb{Z}_2^{(1)}$ symmetry in 3d, which is reflected in the fact that $D_1^s$ is a semion. The combination

$$D_1^{\bar{s}} = D_1^{s\bar{s}} \otimes D_1^s \tag{4.31}$$

generates another emergent $\mathbb{Z}_2^{(1)}$ symmetry which carries a 't Hooft anomaly described by the other generator of the $\mathbb{Z}_4$ anomaly group. We refer to this $\mathbb{Z}_2^{(1)}$ SSB phase as being of semionic type.

### 4.2.2 Non-Minimal Phases

Let us now choose non-minimal BCs as physical boundaries.

$\mathbb{Z}_2^{(1)}$ **Spontaneously Unbroken.** Let us choose the physical boundary to be a general boundary of Dirichlet type

$$\mathfrak{B}^{\text{phys}} = \mathfrak{B}_{\text{Dir}}^{\mathfrak{T}} = \mathfrak{B}_{\text{Dir}} \boxtimes \mathfrak{T}, \tag{4.32}$$

characterized by an irreducible 3d TFT $\mathfrak{T}$ whose topological lines form an MTC $\mathcal{M}$. Since $\mathfrak{B}_{\text{Dir}}^{\mathfrak{T}}$ is simply obtained by stacking $\mathfrak{T}$ with $\mathfrak{B}_{\text{Dir}}$, the 3d TFT resulting from interval compactification with $\mathfrak{B}^{\text{phys}} = \mathfrak{B}_{\text{Dir}}^{\mathfrak{T}}$ is obtained by stacking $\mathfrak{T}$ with the 3d TFT resulting from interval compactification with $\mathfrak{B}^{\text{phys}} = \mathfrak{B}_{\text{Dir}}$. Thus, the result of the interval compactification is

$$\text{Triv} \boxtimes \mathfrak{T} = \mathfrak{T}, \tag{4.33}$$

i.e. we have a single vacuum carrying topological order described by the irreducible 3d TFT $\mathfrak{T}$. The $\mathbb{Z}_2^{(1)}$ symmetry is spontaneously unbroken and realized by the identity line $D_1^{(\text{id})}$. The MTC $\mathcal{M}$ describes emergent 1-form symmetries.

Out of these, the non-minimal phases admitting a $\mathbb{Z}_2^{(1)}$ symmetric gapped boundary condition to a minimal phase are characterized by the property

$$\mathcal{M} = \mathcal{Z}(\mathcal{C}), \tag{4.34}$$

i.e. the MTC $\mathcal{M}$ being the center of a fusion category $\mathcal{C}$.

$\mathbb{Z}_2^{(1)}$ **Spontaneously Broken.** Let us choose the physical boundary to be a general boundary of Neumann type

$$\mathfrak{B}^{\text{phys}} = \mathfrak{B}_{\text{Neu}}^{\mathfrak{T}_{\mathbb{Z}_2}}, \tag{4.35}$$

characterized by a $\mathbb{Z}_2^{(0)}$ symmetric irreducible 3d TFT $\mathfrak{T}_{\mathbb{Z}_2}$. As discussed earlier, the genuine and non-genuine topological lines on the boundary $\mathfrak{B}_{\text{Neu}}^{\mathfrak{T}_{\mathbb{Z}_2}}$ are characterized by an MTC $\widetilde{\mathcal{M}}$ with a chosen $\mathsf{Rep}(\mathbb{Z}_2)$ subcategory generated by the boundary projection $D_1$ of the bulk line $\boldsymbol{Q}_1$. The lines in $\widetilde{\mathcal{M}}$ uncharged under $\mathsf{Rep}(\mathbb{Z}_2)$ are genuine, lying in braided fusion subcategory $\widetilde{\mathcal{M}}_1$, and the lines in $\widetilde{\mathcal{M}}$ charged under $\mathsf{Rep}(\mathbb{Z}_2)$ are non-genuine, lying in a subcategory $\widetilde{\mathcal{M}}_P$. The non-genuine lines arise at an end of the bulk surface $\boldsymbol{Q}_2$ along the boundary $\mathfrak{B}_{\text{Neu}}^{\mathfrak{T}_{\mathbb{Z}_2}}$. Upon interval compactification, the genuine lines on $\mathfrak{B}_{\text{Neu}}^{\mathfrak{T}_{\mathbb{Z}_2}}$ obviously descend to genuine lines of the resulting 3d TFT. On the other hand, the non-genuine lines on $\mathfrak{B}_{\text{Neu}}^{\mathfrak{T}_{\mathbb{Z}_2}}$ also descend to genuine lines of the resulting 3d TFT as $\boldsymbol{Q}_2$ ends also along $\mathfrak{B}^{\text{sym}}$. Thus, the resulting 3d TFT is characterized by the MTC $\widetilde{\mathcal{M}}$, which can be recognized as the 3d TFT

$$\mathfrak{T}_{\mathbb{Z}_2}/\mathbb{Z}_2^{(0)}\,, \tag{4.36}$$

obtained after gauging the $\mathbb{Z}_2^{(0)}$ symmetry of $\mathfrak{T}_{\mathbb{Z}_2}$.

The non-trivial line $D_1 \in \widetilde{\mathcal{M}}$ generates the $\mathbb{Z}_2^{(1)}$ symmetry of the resulting $(2+1)$d gapped phase. Hence, $\mathbb{Z}_2^{(1)}$ is spontaneously broken. Correspondingly, the subcategory $\widetilde{\mathcal{M}}_P$ of charged lines is non-empty[12]. All the line operators in $\widetilde{\mathcal{M}}$ are emergent except for $D_1$.

Out of these, the non-minimal phases admitting a $\mathbb{Z}_2^{(1)}$ symmetric gapped boundary condition to a minimal phase are characterized by the property

$$\mathcal{M} = \mathcal{Z}(\mathcal{C})\,, \tag{4.37}$$

i.e. the MTC $\mathcal{M}$ being the center of a fusion category $\mathcal{C}$, which is equivalent to the condition

$$\widetilde{\mathcal{M}} = \mathcal{Z}(\widetilde{\mathcal{C}})\,, \tag{4.38}$$

i.e. the MTC $\widetilde{\mathcal{M}}$ being the center of a $\mathbb{Z}_2$ graded fusion category $\widetilde{\mathcal{C}}$. This description as a center also uniquely fixes the chosen $\mathsf{Rep}(\mathbb{Z}_2)$ subcategory of $\widetilde{\mathcal{M}}$. An example of such a non-minimal phase is $\mathbb{Z}_4$ DW theory without twist for which we have

$$\widetilde{\mathcal{C}} = \mathsf{Vec}_{\mathbb{Z}_4}\,. \tag{4.39}$$

See the discussion around (3.39) for a description of $\widetilde{\mathcal{M}}$ and the $\mathsf{Rep}(\mathbb{Z}_2)$ subcategory. Another example of such a non-minimal phase is the doubled Ising model for which we have

$$\widetilde{\mathcal{C}} = \mathsf{Ising}, \qquad \widetilde{\mathcal{M}} = \mathsf{Ising} \boxtimes \overline{\mathsf{Ising}}\,. \tag{4.40}$$

See the discussion around (3.60) for more details.

As for non-minimal phases that do not admit such a gapped boundary condition, an example is provided by the 3d CS theory $U(1)_{2n} \times U(1)_{-2m}$. See the discussion preceding equation (3.69) for more details.

---

[12]This is guaranteed by $\widetilde{\mathcal{M}}_1$ being non-modular, since it has a non-trivial Müger center $\mathsf{Rep}(\mathbb{Z}_2)$.

## 4.3 Non-Minimal Symmetries of Dirichlet Type

We now choose a non-minimal BC of Dirichlet type as the symmetry boundary

$$\mathfrak{B}^{\mathrm{sym}} = \mathfrak{B}_{\mathrm{Dir}}^{\mathfrak{T}'} \tag{4.41}$$

for some irreducible 3d TFT $\mathfrak{T}'$. The associated symmetry fusion 2-category is

$$\mathcal{S} = 2\mathsf{Vec}_{\mathbb{Z}_2} \boxtimes \Sigma\mathcal{M}' \tag{4.42}$$

or in other words, we have a direct product of a $\mathbb{Z}_2$ 0-form symmetry, with a possibly non-invertible 1-form symmetry described by the MTC $\mathcal{M}'$ characterizing the 3d TFT $\mathfrak{T}'$. It should be noted that the 1-form symmetry is fully anomalous, i.e. all symmetries in $\mathcal{M}'$ must be spontaneously broken.

The phases for $\mathcal{S}$ symmetry follow simply from phases for $\mathbb{Z}_2^{(0)}$ symmetry as the symmetry boundary for $\mathcal{S}$ is obtained by stacking the symmetry boundary for $\mathbb{Z}_2^{(0)}$, which is $\mathfrak{B}_{\mathrm{Dir}}$, with the 3d TFT $\mathfrak{T}'$. The phases for $\mathcal{S}$ symmetry are thus simply obtained by stacking $\mathbb{Z}_2^{(0)}$ phases with $\mathfrak{T}'$.

### 4.3.1 Minimal Phases

Choosing each minimal BC as the physical boundary, we obtain the minimal phases where there are no emergent 1-form symmetries.

$\mathbb{Z}_2^{(0)}$ **Spontaneously Broken.** For $\mathfrak{B}^{\mathrm{phys}} = \mathfrak{B}_{\mathrm{Dir}}$, the (2+1)d gapped phase is described by the 3d TFT

$$\mathfrak{T}' \oplus \mathfrak{T}', \tag{4.43}$$

i.e. we have two vacua with each vacuum carrying $\mathfrak{T}'$. The two vacua are interchanged by $\mathbb{Z}_2^{(0)}$, which is hence spontaneously broken. The $\mathcal{M}'$ 1-form symmetry is realized diagonally on the two vacua.

$\mathbb{Z}_2^{(0)}$ **Realized Trivially.** For $\mathfrak{B}^{\mathrm{phys}} = \mathfrak{B}_{\mathrm{Neu}}$, the (2+1)d gapped phase is described by the 3d TFT

$$\mathfrak{T}' \tag{4.44}$$

with the $\mathbb{Z}_2^{(0)}$ symmetry realized trivially on it. The $\mathcal{M}'$ symmetry is realized by the lines of $\mathfrak{T}'$.

$\mathbb{Z}_2^{(0)}$ **Realized as an SPT Phase.** For $\mathfrak{B}^{\text{phys}} = \mathfrak{B}_{\text{Neu}}^{\omega}$, the (2+1)d gapped phase is described by the 3d TFT

$$\mathfrak{T}' \tag{4.45}$$

with the $\mathbb{Z}_2^{(0)}$ symmetry realized by a surface operator isomorphic to identity surface operator, but the line implementing the isomorphism having a non-trivial F-symbol. The $\mathbb{Z}_2^{(0)}$ symmetry is thus realized as in an SPT phase. The $\mathcal{M}'$ symmetry is generated by the lines of $\mathfrak{T}'$.

### 4.3.2 Non-Minimal Phases

Let us now choose non-minimal BCs as physical boundaries. These lead to non-minimal phases where we have emergent 1-form symmetries lying outside the MTC $\mathcal{M}'$.

$\mathbb{Z}_2^{(0)}$ **Spontaneously Broken.** For $\mathfrak{B}^{\text{phys}} = \mathfrak{B}_{\text{Dir}}^{\mathfrak{T}}$, the (2+1)d gapped phase is described by the 3d TFT

$$(\mathfrak{T}' \boxtimes \mathfrak{T}) \oplus (\mathfrak{T}' \boxtimes \mathfrak{T}), \tag{4.46}$$

i.e. we have two vacua with each vacuum carrying $\mathfrak{T}' \boxtimes \mathfrak{T}$. The two vacua are interchanged by $\mathbb{Z}_2^{(0)}$, which is hence spontaneously broken. The $\mathcal{M}'$ 1-form symmetry is realized diagonally on the $\mathfrak{T}'$ factors in the two vacua. The $\mathfrak{T}$ factors provide emergent $\mathcal{M}$ 1-form symmetry.

Out of these, the non-minimal phases admitting a $\mathcal{S}$-symmetric gapped boundary condition to a minimal phase are characterized by the condition

$$\mathcal{M} = \mathcal{Z}(\mathcal{C}). \tag{4.47}$$

$\mathbb{Z}_2^{(0)}$ **Spontaneously Unbroken.** For $\mathfrak{B}^{\text{phys}} = \mathfrak{B}_{\text{Neu}}^{\mathfrak{T}_{\mathbb{Z}_2}}$, the (2+1)d gapped phase is described by the 3d TFT

$$\mathfrak{T}' \boxtimes \mathfrak{T}, \tag{4.48}$$

i.e. we have a single vacuum left invariant by $\mathbb{Z}_2^{(0)}$, which is hence spontaneously unbroken. The $\mathbb{Z}_2^{(0)}$ symmetry is realized on the $\mathfrak{T}$ factor according to an extension $\mathcal{M}_{\mathbb{Z}_2}^{\times}$ of $\mathcal{M}$, and hence $\mathbb{Z}_2^{(0)}$ can mix with the emergent IR $\mathcal{M}$ 1-form symmetries. The UV $\mathcal{M}'$ 1-form symmetry is realized straightforwardly via the lines coming from the $\mathfrak{T}'$ factor.

Out of these, the non-minimal phases admitting an $\mathcal{S}$-symmetric gapped boundary condition to a minimal phase are again characterized by the property

$$\mathcal{M} = \mathcal{Z}(\mathcal{C}). \tag{4.49}$$

## 4.4  Non-Minimal Symmetries of Neumann Type

We now choose a non-minimal BC of Neumann type as the symmetry boundary

$$\mathfrak{B}^{\mathrm{sym}} = \mathfrak{B}_{\mathrm{Neu}}^{\mathfrak{T}'_{\mathbb{Z}_2}} \tag{4.50}$$

for some $\mathbb{Z}_2^{(0)}$ symmetric irreducible 3d TFT $\mathfrak{T}'_{\mathbb{Z}_2}$, whose underlying non-symmetric 3d TFT will be denoted as $\mathfrak{T}'$. We recall the various categories associated with such a BC. See the previous section for more details. Let $\mathcal{M}'$ be the MTC associated to $\mathfrak{T}'$, $\mathcal{M}'^{\times}_{\mathbb{Z}_2}$ be the $\mathbb{Z}_2$ extension of $\mathcal{M}'$ associated to $\mathfrak{T}'_{\mathbb{Z}_2}$, and $\widetilde{\mathcal{M}}'$ be the MTC obtained by $\mathbb{Z}_2$ equivariantization of $\mathcal{M}'^{\times}_{\mathbb{Z}_2}$. The MTC $\widetilde{\mathcal{M}}'$ is $\mathbb{Z}_2$ graded with the trivial grade being labeled as $\widetilde{\mathcal{M}}'_1$ and the non-trivial grade being labeled as $\widetilde{\mathcal{M}}'_P$. The associated symmetry fusion 2-category is

$$\mathcal{S} = \Sigma\widetilde{\mathcal{M}}'_1 \tag{4.51}$$

or in other words, we have a possibly non-invertible 1-form symmetry described by the braided fusion category $\widetilde{\mathcal{M}}'_1$. Recall that the its Müger center is

$$\mathcal{Z}_2(\widetilde{\mathcal{M}}'_1) = \mathsf{Rep}(\mathbb{Z}_2)\,, \tag{4.52}$$

i.e. the 1-form symmetry $\widetilde{\mathcal{M}}'_1$ contains a $\mathbb{Z}_2^{(1)}$ subsymmetry which is free of all pure and mixed 't Hooft anomalies, and moreover all the other 1-form symmetries in $\widetilde{\mathcal{M}}'_1$ are afflicted with some 't Hooft anomaly. The various phases for $\mathcal{S}$ symmetry are thus characterized by the spontaneous breaking or non-breaking $\mathbb{Z}_2^{(1)}$, as other elements of $\widetilde{\mathcal{M}}'_1$ must be spontaneously broken.

### 4.4.1  Minimal Phases

Choosing each of the minimal BCs as the physical boundary, we obtain the minimal phases where the emergent 1-form symmetries are minimal.

$\mathbb{Z}_2^{(1)}$ **Spontaneously Unbroken.**  For $\mathfrak{B}^{\mathrm{phys}} = \mathfrak{B}_{\mathrm{Dir}}$, the (2+1)d gapped phase is described by the 3d TFT

$$\mathfrak{T}'\,, \tag{4.53}$$

which follows from the discussion around (4.17), as we have simply exchanged the roles of symmetry and physical boundaries, which does not impact the result of the sandwich construction. The symmetry $\widetilde{\mathcal{M}}'_1$ is realized via the braided tensor functor

$$\phi: \ \widetilde{\mathcal{M}}'_1 \to \mathcal{M}'\,, \tag{4.54}$$

which is the forgetful functor that sends a $\mathbb{Z}_2$ equivariant object to its underlying object in $\mathcal{M}'$. Physically, this map can be understood by gauging (also kown as condensation) $\mathbb{Z}_2^{(1)} \subset \widetilde{\mathcal{M}}'_1$, whose generator is referred to as $D_1$. By definition, all lines in $\widetilde{\mathcal{M}}'_1$ are uncharged under this $\mathbb{Z}_2^{(1)}$, and hence remain genuine lines after gauging. As is well-known, there are two possible fates of a simple line $D_1^a \in \widetilde{\mathcal{M}}'_1$:

- If $D_1^a \otimes D_1 \cong D_1^a$, then $D_1^a$ splits into two simple lines $D_1^{a,+}$ and $D_1^{a,-}$ in the category $\mathcal{M}'$ obtained after gauging $\mathbb{Z}_2^{(1)}$. In this case we have

$$\phi(D_1^a) = D_1^{a,+} \oplus D_1^{a,-} . \tag{4.55}$$

- If $D_1^a \otimes D_1 = D_1^b \not\cong D_1^a$, then $D_1^a$ becomes a simple line $D_1^{a,b}$ in the category $\mathcal{M}'$ obtained after gauging $\mathbb{Z}_2^{(1)}$. In this case we have

$$\phi(D_1^a) = \phi(D_1^b) = D_1^{a,b} . \tag{4.56}$$

In particular, we have

$$D_1 \otimes D_1 = D_1^{\mathrm{id}} \tag{4.57}$$

and hence

$$\phi(D_1) = D_1^{\mathrm{id}} , \tag{4.58}$$

i.e. $\mathbb{Z}_2^{(1)} \subset \widetilde{\mathcal{M}}'_1$ is spontaneously unbroken in this $\widetilde{\mathcal{M}}'_1$-symmetric (2+1)d gapped phase.

$\mathbb{Z}_2^{(1)}$ **Spontaneously Broken: Type I.** For $\mathfrak{B}^{\mathrm{phys}} = \mathfrak{B}_{\mathrm{Neu}}$, the (2+1)d gapped phase is described by the 3d TFT

$$\mathfrak{T}'_{\mathbb{Z}_2}/\mathbb{Z}_2^{(0)} , \tag{4.59}$$

namely the theory obtained after gauging the $\mathbb{Z}_2^{(0)}$ symmetry of $\mathfrak{T}'_{\mathbb{Z}_2}$. This follows from the discussion around (4.36), as we have simply exchanged the roles of symmetry and physical boundaries, which does not impact the result of the sandwich construction. The lines of this 3d TFT form the MTC $\widetilde{\mathcal{M}}'$. The symmetry $\widetilde{\mathcal{M}}'_1$ is realized via the inclusion

$$\phi : \widetilde{\mathcal{M}}'_1 \hookrightarrow \widetilde{\mathcal{M}}' . \tag{4.60}$$

Since $\phi(D_1)$ is a non-trivial line, the $\mathbb{Z}_2^{(1)}$ subsymmetry is spontaneously broken. The lines in $\widetilde{\mathcal{M}}'_P \subset \widetilde{\mathcal{M}}'$ are all charged under $\mathbb{Z}_2^{(1)}$. The $\widetilde{\mathcal{M}}'_P$ generate emergent 1-form symmetries. Note that this is a minimal amount of emergent 1-form symmetry, as the emergence of at least one line in $\widetilde{\mathcal{M}}'_P$ is required for spontaneous breaking of $\mathbb{Z}_2^{(1)}$, but this implies the emergence of all lines in $\widetilde{\mathcal{M}}'_P$ by taking fusions with the lines in $\widetilde{\mathcal{M}}'_1$.

$\mathbb{Z}_2^{(1)}$ **Spontaneously Broken: Type II.** For $\mathfrak{B}^{\text{phys}} = \mathfrak{B}^{\omega}_{\text{Neu}}$, the (2+1)d gapped phase is described by the 3d TFT

$$\mathfrak{T}'_{\mathbb{Z}_2}/(\mathbb{Z}_2^{(0)}, \omega), \tag{4.61}$$

namely the theory obtained after gauging the $\mathbb{Z}_2^{(0)}$ symmetry of $\mathfrak{T}'_{\mathbb{Z}_2}$ with a discrete torsion $\omega$. The lines of this 3d TFT form a $\mathbb{Z}_2$ graded MTC $\widetilde{\mathcal{M}}'_{\omega}$ whose trivial grade is

$$(\widetilde{\mathcal{M}}'_{\omega})_1 = \widetilde{\mathcal{M}}'_1, \tag{4.62}$$

while the simple lines in non-trivial grade $(\widetilde{\mathcal{M}}'_{\omega})_P$ can be expressed as

$$D_1^a \otimes D_1^s, \tag{4.63}$$

where $D_1^a$ is a simple line in $\widetilde{\mathcal{M}}'_P$ and $D_1^s$ is a semion that is invisible to the lines in $\widetilde{\mathcal{M}}'$. The inclusion of the $D_1^s$ factor modifies the spin $\theta$ and braidings $B$ of lines in the non-trivial grade:

$$\theta(D_1^a \otimes D_1^s) = i\theta(D_1^a), \qquad B(D_1^a \otimes D_1^s, D_1^b \otimes D_1^s) = -B(D_1^a, D_1^b). \tag{4.64}$$

The $\mathbb{Z}_2^{(1)}$ subsymmetry is again spontaneously broken, and the lines in $(\widetilde{\mathcal{M}}'_{\omega})_P$ provide a minimal emergent symmetry arising due to this spontaneous breaking.

**Examples.** Let us discuss a few examples:

- Consider the symmetry boundary discussed around (3.41) giving rise to a $\mathbb{Z}_4^{(1),e} \times \mathbb{Z}_2^{(1),m^2}$ symmetry with mixed 't Hooft anomaly (3.42). The fully non-anomalous $\mathbb{Z}_2^{(1)}$ subsymmetry is given by the $\mathbb{Z}_2$ subgroup of $\mathbb{Z}_4^{(1),e}$, and denoted as $\mathbb{Z}_2^{(1),e^2}$. As discussed in previous section, this symmetry is gauge related to the minimal $\mathbb{Z}_2^{(0)}$ and $\mathbb{Z}_2^{(1)}$ symmetries. The minimal phases are

  1. The phase with $\mathbb{Z}_2^{(1),e^2}$ spontaneously unbroken is given by the toric code. The $\mathbb{Z}_4^{(1),e}$ is realized by the line $D_1^e$ of TC and $\mathbb{Z}_2^{(1),m^2}$ is realized by the line $D_1^m$ of TC.

  2. The type I phase with $\mathbb{Z}_2^{(1),e^2}$ spontaneously broken is given by the $\mathbb{Z}_4$ DW theory without twist. The lines of $\mathbb{Z}_4$ DW theory form

     $$\mathbb{Z}_4^{(1),e} \times \mathbb{Z}_4^{(1),m}, \tag{4.65}$$

     where the generators of $\mathbb{Z}_4^{(1),e}$ and $\mathbb{Z}_4^{(1),m}$ are bosons, and braid with the phase $+i$. The $\mathbb{Z}_4^{(1),e}$ symmetry is realized by the $\mathbb{Z}_4^{(1),e}$ factor of (4.65), and the $\mathbb{Z}_2^{(1),m^2}$ symmetry is realized by the $\mathbb{Z}_2$ subgroup of the $\mathbb{Z}_4^{(1),m}$ factor of (4.65).

3. The type II phase with $\mathbb{Z}_2^{(1),e^2}$ spontaneously broken is given by the $\mathbb{Z}_4$ DW theory with twist given by the generator of the $\mathbb{Z}_2$ subgroup of $H^3(\mathbb{Z}_4, U(1)) = \mathbb{Z}_4$. The lines of this twisted $\mathbb{Z}_4$ DW theory form

$$\mathbb{Z}_4^{(1),e} \times \mathbb{Z}_4^{(1),ms}, \tag{4.66}$$

where the generator of $\mathbb{Z}_4^{(1),e}$ is a boson, the generator of $\mathbb{Z}_4^{(1),m}$ is a semion, and the two braid with the phase $+i$. The $\mathbb{Z}_4^{(1),e}$ symmetry is realized by the $\mathbb{Z}_4^{(1),e}$ factor of (4.66), and the $\mathbb{Z}_2^{(1),m^2}$ symmetry is realized by the $\mathbb{Z}_2$ subgroup of the $\mathbb{Z}_4^{(1),ms}$ factor of (4.66).

- Consider the symmetry boundary discussed around (3.60) giving rise to a non-invertible $\mathsf{TY}^{(1)}_{\mathbb{Z}_2 \times \mathbb{Z}_2}$ symmetry, see (3.63). The fully non-anomalous $\mathbb{Z}_2^{(1)}$ subsymmetry is generated by $D_1^{\psi\bar{\psi}}$. As discussed there, this symmetry is gauge related to the minimal $\mathbb{Z}_2^{(0)}$ and $\mathbb{Z}_2^{(1)}$ symmetries. Its minimal phases are

  1. The phase with $\mathbb{Z}_2^{(1)}$ spontaneously unbroken is given by the toric code. The $\mathsf{TY}^{(1)}_{\mathbb{Z}_2 \times \mathbb{Z}_2}$ symmetry is realized as

     $$\begin{aligned} \phi(D_1^{\mathrm{id}}) = \phi(D_1^{\psi\bar{\psi}}) &= D_1^{\mathrm{id}} \\ \phi(D_1^{\psi}) = \phi(D_1^{\bar{\psi}}) &= D_1^{f} \\ \phi(D_1^{\sigma\bar{\sigma}}) &= D_1^{e} \oplus D_1^{m} \,. \end{aligned} \tag{4.67}$$

  2. The type I phase with $\mathbb{Z}_2^{(1)}$ spontaneously broken is given by the doubled Ising model, or the center $\mathcal{Z}(\mathsf{TY}_{\mathbb{Z}_2}^{+})$ of the $\mathsf{TY}_{\mathbb{Z}_2}$ fusion category with trivial Frobenius-Schur indicator. On the other hand, the type II phase with $\mathbb{Z}_2^{(1)}$ spontaneously broken is given by the center $\mathcal{Z}(\mathsf{TY}_{\mathbb{Z}_2}^{-})$ of the $\mathsf{TY}_{\mathbb{Z}_2}$ fusion category with non-trivial Frobenius-Schur indicator.

- Consider the symmetry boundary discussed around (3.69) giving rise to the symmetry (3.71) where $\widetilde{\mathcal{M}}_1$ is generated by anyons $(a, b)$ with $a - b \in 2\mathbb{Z}$.

  The phase with $\mathfrak{B}^{\mathrm{phys}} = \mathfrak{B}_{\mathrm{Dir}}$ has spontaenously unbroken $\mathbb{Z}_2^{(1)}$ symmetry. This is a 3d TFT described by the K matrix (3.74). In terms of charge label of $U(1) \times U(1)$ theory, the symmetry is realized via the braided tensor functor

  $$\begin{aligned} \phi : (a, b) &\mapsto (a, b), & \forall 0 \le a < n \\ (a, b) &\mapsto (a + n \mod 2n, b + m \mod 2m), & \forall n \le a < 2n \end{aligned} \tag{4.68}$$

  Choosing $\mathfrak{B}^{\mathrm{phys}} = \mathfrak{B}_{\mathrm{Neu}}$, we have the Type I phase with $\mathbb{Z}_2^{(1)}$ spontaneously broken recovering the abelian TFT $U(1)_{2n} \times U(1)_{-2m}$. On the other hand, choosing $\mathfrak{B}^{\mathrm{phys}} =$

$\mathfrak{B}_{\mathrm{Neu}}^{\omega}$, the lines in the resulting 3d TFT is identical to that of $U(1)_{2n} \times U(1)_{-2m}$ except the lines in the odd sector is further accompanied by $D_1^s$, thus the spin and the braiding are modified according to (4.64).

### 4.4.2 Non-Minimal Phases

Let us now choose non-minimal BCs as physical boundaries. These lead to non-minimal phases where we have non-minimal amount of emergent 1-form symmetries.

$\mathbb{Z}_2^{(1)}$ **Spontaneously Unbroken.** For $\mathfrak{B}^{\mathrm{phys}} = \mathfrak{B}_{\mathrm{Dir}}^{\mathfrak{T}}$, the (2+1)d gapped phase is described by the 3d TFT

$$\mathfrak{T}' \boxtimes \mathfrak{T}, \tag{4.69}$$

with the $\widetilde{\mathcal{M}}_1'$ symmetry being realized on the $\mathfrak{T}'$ factor via the braided tensor functor (4.54). The MTC $\mathcal{M}$ coming from the $\mathfrak{T}$ factor is an emergent 1-form symmetry.

Such a phase admits a $\widetilde{\mathcal{M}}_1'$-symmetric gapped boundary to a minimal phase only if $\mathcal{M} = \mathcal{Z}(\mathcal{C})$.

$\mathbb{Z}_2^{(1)}$ **Spontaneously Broken.** Consider now $\mathfrak{B}^{\mathrm{phys}} = \mathfrak{B}_{\mathrm{Neu}}^{\mathfrak{T}_{\mathbb{Z}_2}}$. We have two MTCs $\widetilde{\mathcal{M}}'$ and $\widetilde{\mathcal{M}}$ associated to the symmetry and physical boundaries respectively. The genuine lines on $\mathfrak{B}^{\mathrm{sym}}$ and $\mathfrak{B}^{\mathrm{phys}}$ give rise to lines in the resulting 3d TFT, which are valued in the category

$$\widetilde{\mathcal{M}}_1' \boxtimes \widetilde{\mathcal{M}}_1. \tag{4.70}$$

Moreover, the bulk surface $Q_2$ can end on both boundaries, and thus we obtain another set of lines valued in the category

$$\widetilde{\mathcal{M}}_P' \boxtimes \widetilde{\mathcal{M}}_P, \tag{4.71}$$

which can be depicted as

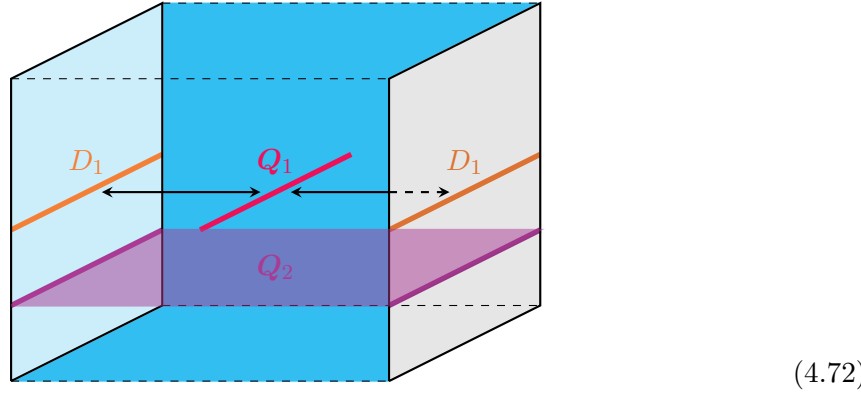

$$\tag{4.72}$$

In total, we have obtained a braided fusion category

$$\widetilde{\mathcal{M}}' \boxtimes_{\mathbb{Z}_2} \widetilde{\mathcal{M}} = (\widetilde{\mathcal{M}}_1' \boxtimes \widetilde{\mathcal{M}}_1) \boxplus (\widetilde{\mathcal{M}}_P' \boxtimes \widetilde{\mathcal{M}}_P) \tag{4.73}$$

worth of lines in the 3d TFT. However, this is not an MTC: if $D_1' \in \widetilde{\mathcal{M}}_1'$ and $D_1 \in \widetilde{\mathcal{M}}_1$ are projections of the bulk line $\boldsymbol{Q}_1$ on the two boundaries, then all lines in (4.73) are transparent to $D_1' \otimes D_1$. Indeed, we have slightly overcounted the total number of lines in (4.73): the sandwich construction with bulk line $\boldsymbol{Q}_1$ placed parallel to boundaries provides a single line in the resulting 3d TFT, but (4.73) involves two images $D_1'$ and $D_1$ of $\boldsymbol{Q}_1$. This overcounting can be removed by gauging the $\mathbb{Z}_2^{(1)}$ subsymmetry of (4.73) generated by $D_1' \otimes D_1$, which results in an MTC

$$\frac{\widetilde{\mathcal{M}}' \boxtimes_{\mathbb{Z}_2} \widetilde{\mathcal{M}}}{\mathbb{Z}_2^{(1)}} . \tag{4.74}$$

This is the MTC associated to the 3d TFT resulting from the interval compactification.

In order to describe the realization of $\widetilde{\mathcal{M}}_1'$ symmetry on this 3d TFT, note that we have a braided tensor functor

$$\widetilde{\mathcal{M}}' \boxtimes_{\mathbb{Z}_2} \widetilde{\mathcal{M}} \to \frac{\widetilde{\mathcal{M}}' \boxtimes_{\mathbb{Z}_2} \widetilde{\mathcal{M}}}{\mathbb{Z}_2^{(1)}} \tag{4.75}$$

because $\widetilde{\mathcal{M}}' \boxtimes_{\mathbb{Z}_2} \widetilde{\mathcal{M}}$ can be obtained as $\mathbb{Z}_2$ equivariantization of (4.74), or in other words by gauging the dual $\mathbb{Z}_2^{(0)}$ symmetry. This functor sends a $\mathbb{Z}_2$ equivariant object to its underlying object, in the way described in detail after equation (4.54). Combining this with the obvious functor

$$\widetilde{\mathcal{M}}_1' \to \widetilde{\mathcal{M}}_1' \boxtimes \widetilde{\mathcal{M}}_1 \hookrightarrow \widetilde{\mathcal{M}}' \boxtimes_{\mathbb{Z}_2} \widetilde{\mathcal{M}}, \tag{4.76}$$

we obtain the realization

$$\widetilde{\mathcal{M}}_1' \to \frac{\widetilde{\mathcal{M}}' \boxtimes_{\mathbb{Z}_2} \widetilde{\mathcal{M}}}{\mathbb{Z}_2^{(1)}} \tag{4.77}$$

of the $\widetilde{\mathcal{M}}_1'$ on the resulting 3d TFT.

In fact the resulting 3d TFT can be identified with

$$\mathfrak{T}_{\mathbb{Z}_2}' \boxtimes \mathfrak{T}_{\mathbb{Z}_2}/\mathbb{Z}_2^{(0)} \tag{4.78}$$

where we gauge the diagonal $\mathbb{Z}_2^{(0)}$ symmetry of $\mathfrak{T}_{\mathbb{Z}_2}' \boxtimes \mathfrak{T}_{\mathbb{Z}_2}$.

Such a phase admits a $\widetilde{\mathcal{M}}_1'$-symmetric gapped boundary to a minimal phase only if $\widetilde{\mathcal{M}} = \mathcal{Z}(\widetilde{\mathcal{C}})$, where $\widetilde{\mathcal{C}}$ is a $\mathbb{Z}_2$-graded fusion category. If we furthermore assume that $\mathfrak{B}^{\mathrm{sym}}$ is gauge related to $\mathfrak{B}_{\mathrm{Dir}}$, i.e. if $\widetilde{\mathcal{M}}' = \mathcal{Z}(\widetilde{\mathcal{C}}')$ where $\widetilde{\mathcal{C}}'$ is a $\mathbb{Z}_2$-graded fusion category, then we can express the MTC resulting after interval compactification simply as

$$\frac{\widetilde{\mathcal{M}}' \boxtimes_{\mathbb{Z}_2} \widetilde{\mathcal{M}}}{\mathbb{Z}_2^{(1)}} = \frac{\mathcal{Z}(\widetilde{\mathcal{C}}') \boxtimes_{\mathbb{Z}_2} \mathcal{Z}(\widetilde{\mathcal{C}})}{\mathbb{Z}_2^{(1)}} = \mathcal{Z}\big(\widetilde{\mathcal{C}}' \boxtimes_{\mathbb{Z}_2} \widetilde{\mathcal{C}}\big) . \tag{4.79}$$

Let us discuss a couple of examples:

- We take $\mathfrak{B}^{\mathrm{sym}}$ to be the BC discussed around (4.65), for which we have

$$\begin{aligned}
\widetilde{\mathcal{M}}' &= \mathcal{Z}(\mathsf{Vec}_{\mathbb{Z}_4}) \\
&= \left( \{D_1^{\mathrm{id}}, D_1^e, D_1^{e^2}, D_1^{e^3}\} \boxtimes \{D_1^{\mathrm{id}}, D_1^{m^2}\} \right) \boxplus \left( \{D_1^{\mathrm{id}}, D_1^e, D_1^{e^2}, D_1^{e^3}\} \boxtimes \{D_1^m, D_1^{m^3}\} \right),
\end{aligned}$$
(4.80)

where we have also described the $\mathbb{Z}_2$ grading. This carries a $\mathbb{Z}_4^{(1),e} \times \mathbb{Z}_2^{(1),m^2}$ symmetry with mixed 't Hooft anomaly (3.42). Let us consider a non-minimal (2+1)d gapped phase with such a symmetry associated to choosing the physical boundary to be the BC discussed around (3.60), for which we have

$$\widetilde{\mathcal{M}} = \mathcal{Z}(\mathsf{Ising}) = \{D_1^{\mathrm{id}}, D_1^\psi, D_1^{\bar\psi}, D_1^{\psi\bar\psi}, D_1^{\sigma\bar\sigma}\} \boxplus \{D_1^\sigma, D_1^{\bar\sigma}, D_1^{\bar\psi\sigma}, D_1^{\psi\bar\sigma}\},$$
(4.81)

where we have also described the $\mathbb{Z}_2$ grading. We have

$$\frac{\widetilde{\mathcal{M}}' \boxtimes_{\mathbb{Z}_2} \widetilde{\mathcal{M}}}{\mathbb{Z}_2^{(1)}} \supset \frac{\widetilde{\mathcal{M}}_1' \boxtimes \widetilde{\mathcal{M}}_1}{\mathbb{Z}_2^{(1)}} = \{D_1^{\mathrm{id}}, D_1^e\} \boxtimes \{D_1^{\mathrm{id}}, D_1^{m^2}\} \boxtimes \{D_1^{\mathrm{id}}, D_1^\psi, D_1^{\bar\psi}, D_1^{\psi\bar\psi}, D_1^{\sigma\bar\sigma}\}$$
(4.82)

with

$$D_1^e \otimes D_1^e = D_1^{\psi\bar\psi}$$
(4.83)

Thus the lines $\{D_1^{\mathrm{id}}, D_1^e\} \boxtimes \{D_1^{\mathrm{id}}, D_1^\psi, D_1^{\bar\psi}, D_1^{\psi\bar\psi}, D_1^{\sigma\bar\sigma}\}$ form an extension of $\{D_1^{\mathrm{id}}, D_1^e\} = \mathsf{Rep}(\mathbb{Z}_2)$ by $\mathsf{TY}_{\mathbb{Z}_2 \times \mathbb{Z}_2} = \{D_1^{\mathrm{id}}, D_1^\psi, D_1^{\bar\psi}, D_1^{\psi\bar\psi}, D_1^{\sigma\bar\sigma}\}$. The symmetry $\mathbb{Z}_4^{(1),e}$ is generated by $D_1^e$ in (4.82), and the symmetry $\mathbb{Z}_2^{(1),m^2}$ is generated by $D_1^{m^2}$ in (4.82).

To complete the description of the resulting 3d TFT, we also have lines

$$\frac{\widetilde{\mathcal{M}}' \boxtimes_{\mathbb{Z}_2} \widetilde{\mathcal{M}}}{\mathbb{Z}_2^{(1)}} \supset \frac{\widetilde{\mathcal{M}}_P' \boxtimes \widetilde{\mathcal{M}}_P}{\mathbb{Z}_2^{(1)}} = \{D_1^{\mathrm{id}}, D_1^e\} \boxtimes \{D_1^m, D_1^{m^3}\} \boxtimes \{D_1^\sigma, D_1^{\bar\sigma}, D_1^{\bar\psi\sigma}, D_1^{\psi\bar\sigma}\}$$
(4.84)

- Now we interchange the roles of symmetry and physical boundary in the above example. This leads to a non-minimal phase for non-invertible $\mathsf{TY}_{\mathbb{Z}_2 \times \mathbb{Z}_2}^{(1)}$ symmetry, which is realized by $\{D_1^{\mathrm{id}}, D_1^e\} \boxtimes \{D_1^{\mathrm{id}}, D_1^\psi, D_1^{\bar\psi}, D_1^{\psi\bar\psi}, D_1^{\sigma\bar\sigma}\}$ in (4.82).

- Again the MTCs $\widetilde{\mathcal{M}}$ and $\widetilde{\mathcal{M}}'$ do not have to be a center. For example take $\widetilde{\mathcal{M}}' = \mathcal{Z}(\mathsf{Ising})$ and $\widetilde{\mathcal{M}} = U(1)_{2n} \times U(1)_{-2m}$ discussed around (3.69). In the first part of $\frac{\widetilde{\mathcal{M}}' \boxtimes_{\mathbb{Z}_2} \widetilde{\mathcal{M}}}{\mathbb{Z}_2^{(1)}}$, we have:

$$\frac{\widetilde{\mathcal{M}}_1' \boxtimes \widetilde{\mathcal{M}}_1}{\mathbb{Z}_2^{(1)}} = \{D_1^{\mathrm{id}}, D_1^\psi, D_1^{\bar\psi}, D_1^{\psi\bar\psi}, D_1^{\sigma\bar\sigma}\} \boxtimes \{(a,b), 0 \le a < n, 0 \le b < 2m, a - b \in 2\mathbb{Z}\},$$
(4.85)

where $D_1^{\psi\bar\psi} \otimes (n,m)$ is identified with the identity line. In addition, we also have:

$$\frac{\widetilde{\mathcal{M}}'_P \boxtimes \widetilde{\mathcal{M}}_P}{\mathbb{Z}_2^{(1)}} = \{D_1^\sigma, D_1^{\bar\sigma}, D_1^{\bar\psi\sigma}, D_1^{\psi\bar\sigma}\} \boxtimes \{(a,b), 0 \le a < n, 0 \le b < 2m, a - b \in 2\mathbb{Z} + 1\}. \tag{4.86}$$

# 5 Generalization to any (3+1)d SymTFTs of Bosonic Type

In this section, we generalize the discussion of the previous section to SymTFTs for other 3d symmetries of bosonic type. These are (3+1)d DW gauge theories based on a finite, possibly non-abelian, 0-form group $G$ possibly with a twist

$$\tau \in H^4(G, U(1)). \tag{5.1}$$

In this section, we will provide a theta construction for arbitrary topological boundary conditions of these (3+1)d TFTs.

## 5.1 Gapped Boundary Conditions

**Theta Construction of BCs.** We begin with a topological boundary condition realizing $G$ 0-form symmetry in 3d with 't Hooft anomaly $\tau$, and refer to this BC as $\mathfrak{B}_{\mathrm{Dir}}$. Performing sandwich constructions with $\mathfrak{B}_{\mathrm{Dir}}$ as the symmetry boundary yields 3d TFTs with $G$ 0-form symmetry having 't Hooft anomaly $\tau$. Let $\mathfrak{T}_G^\tau$ be such an irreducible $(G, \tau)$ symmetric[13] 3d TFT. The corresponding physical boundary condition can be obtained as

$$\mathfrak{B}_{\mathfrak{T}_G^\tau}^{\mathrm{phys}} = \frac{\mathfrak{B}_{\mathrm{Dir}} \boxtimes \overline{\mathfrak{T}_G^\tau}}{G^{(0)}}, \tag{5.2}$$

where we first stack $\mathfrak{B}_{\mathrm{Dir}}$ with the orientation reversal[14] $\overline{\mathfrak{T}_G^\tau}$ of the 3d TFT $\mathfrak{T}_G^\tau$, and then gauge the diagonal $G$ 0-form symmetry. Note that the diagonal $G$ is anomaly-free since the $G$ symmetry of $\overline{\mathfrak{T}_G^\tau}$ carries $\tau^{-1}$ 't Hooft anomaly, which cancels against the anomaly of the $G$ symmetry living on $\mathfrak{B}_{\mathrm{Dir}}$.

**Classification of BCs.** Since the sandwich construction provides a one-to-one correspondence between 3d symmetric TFTs and topological BCs of the SymTFT, the above BCs $\mathfrak{B}_{\mathfrak{T}_G^\tau}^{\mathrm{phys}}$ are all the irreducible topological BCs of the (3+1)d $(G, \tau)$ DW theory. These are in correspondence with irreducible $(G, \tau)$ symmetric 3d TFTs, which are classified by

---

[13]The corresponding symmetry fusion 2-category is denoted as $2\mathsf{Vec}_G^\tau$.

[14]We have suppressed the orientation reversal operation in the previous subsection on non-anomalous $\mathbb{Z}_2$ symmetry for brevity.

1. A subgroup $H \subseteq G$, such that $\tau$ restricted to $H$ trivializes.

2. An MTC $\widetilde{\mathcal{M}}$ with a choice of $\mathsf{Rep}(H)$ braided fusion subcategory, or equivalently an $H$-crossed extension $\mathcal{M}_H^{\times}$ of an MTC $\mathcal{M}$. The two are related by $H$-equivariantization

$$\widetilde{\mathcal{M}} = (\mathcal{M}_H^{\times})^H . \tag{5.3}$$

We will denote such a $(G, \tau)$-symmetric 3d TFT as

$$\mathfrak{T}_G^{\tau} = \mathfrak{T}^{\mathcal{M}_H^{\times}} \tag{5.4}$$

and the corresponding physical boundary as

$$\mathfrak{B}_{\mathfrak{T}_G^{\tau}}^{\mathrm{phys}} = \mathfrak{B}_{\mathrm{Neu}_H}^{\mathcal{M}_H^{\times}} . \tag{5.5}$$

Physically, a $(G, \tau)$ symmetric 3d TFT characterized by the tuple $(H, \widetilde{\mathcal{M}})$ has vacua $v_{gH}$ identified with $H$-cosets $gH$ in $G$. The $G$ symmetry in general permutes these vacua according to how the left multiplication by group elements permutes the $H$-cosets. In a particular vacuum $v_{gH}$, some subgroup of $G$ isomorphic to $H$ remains spontaneously unbroken, i.e. leaves the vacuum invariant. In particular, the vacuum $v_H$ corresponding to the trivial $H$ coset has the property that $H$ is the subgroup spontaneously unbroken in this vacuum. The MTC $\mathcal{M}$ characterizes the topological order in the vacuum $v_H$, or in other words the topological line operators of the irreducible (non-symmetric) 3d TFT, that we label as $\mathfrak{T}$, describing the vacuum $v_H$. The extension $\mathcal{M}_H^{\times}$ of $\mathcal{M}$ characterizes the surface operators of $\mathfrak{T}$ realizing the action of $H$ on $\mathfrak{T}$. The crossed braided category $\mathcal{M}_H^{\times}$ is $H$-graded, and an object lying in grade $h \in H$ describes a topological line living at the end of the surface realizing the action of $h$.

**Topological Defects on Boundary and Associated Symmetry 2-Category.** The MTC $\widetilde{\mathcal{M}}$ on the other hand characterizes genuine and non-genuine topological lines living on the resulting irreducible topological BC $\mathfrak{B}_{\mathrm{Neu}_H}^{\mathcal{M}_H^{\times}}$. The lines living in the $\mathsf{Rep}(H)$ subcategory are all genuine and capture projections of the bulk Wilson lines $\boldsymbol{Q}_1^R$ forming the symmetric braided fusion category $\mathsf{Rep}(G)$. The projection map relating the bulk lines to the boundary lines is given by the functor

$$\mathsf{Rep}(G) \rightarrow \mathsf{Rep}(H) , \tag{5.6}$$

decomposing each $G$-representation into $H$-representations by restricting the $G$-action to subgroup $H \subseteq G$. The existence of $\mathsf{Rep}(H)$ subcategory allows one to grade $\widetilde{\mathcal{M}}$ by the set of

conjugacy classes of the group $H$, as these conjugacy classes capture the charges of line operators under a $\mathsf{Rep}(H)$ non-invertible 1-form symmetry. The charge, or the braiding phase, is simply the character of a conjugacy class in a representation. We can thus write

$$\widetilde{\mathcal{M}} = \bigoplus_{[h]} \widetilde{\mathcal{M}}_{[h]}. \tag{5.7}$$

Lines in $\widetilde{\mathcal{M}}_{[\mathrm{id}]}$ are genuine and generate a possibly non-invertible 1-form symmetry that is associated to the boundary $\mathfrak{B}_{\mathrm{Neu}_H}^{\mathcal{M}_H^\times}$. The Müger center of the braided fusion category $\widetilde{\mathcal{M}}_{[\mathrm{id}]}$ is

$$\mathcal{Z}_2\left(\widetilde{\mathcal{M}}_{[\mathrm{id}]}\right) = \mathsf{Rep}(H), \tag{5.8}$$

which means that the part of the $\widetilde{\mathcal{M}}_{[\mathrm{id}]}$ 1-form symmetry that is free of all pure and mixed 't Hooft anomalies is $\mathsf{Rep}(H)$.

The other lines in $\widetilde{\mathcal{M}}$ are attached to bulk topological surfaces. In particular, a line in $\mathcal{M}_{[h]}$ is attached to the bulk surface $\boldsymbol{Q}_2^{[g]}$ where $[g]$ is a conjugacy class of $G$ such that $[h] \subseteq [g]$. The bulk topological surfaces $\boldsymbol{Q}_2^{[g]}$ for $G$ conjugacy classes $[g]$ having the property $[g] \cap H$ is the empty set do not end on the boundary $\mathfrak{B}_{\mathrm{Neu}_H}^{\mathcal{M}_H^\times}$ and instead project to become non-condensation surface defects on the boundary generating possibly non-invertible 0-form symmetries. In total, these boundary surface defects combine with $\widetilde{\mathcal{M}}_{[\mathrm{id}]}$ to generate a fusion 2-category $\mathcal{S}(\mathfrak{B}_{\mathrm{Neu}_H}^{\mathcal{M}_H^\times})$ which is the symmetry 2-category associated to the boundary $\mathfrak{B}_{\mathrm{Neu}_H}^{\mathcal{M}_H^\times}$. In fact, we can provide a gauging construction for $\mathcal{S}(\mathfrak{B}_{\mathrm{Neu}_H}^{\mathcal{M}_H^\times})$ as

$$\mathcal{S}(\mathfrak{B}_{\mathrm{Neu}_H}^{\mathcal{M}_H^\times}) = \frac{2\mathsf{Vec}_G^\tau \boxtimes \Sigma\mathcal{M}}{H^{(0)}}, \tag{5.9}$$

where we stack the symmetries $2\mathsf{Vec}_G^\tau$ and $\Sigma\mathcal{M}$, and gauge a diagonal $H^{(0)}$ symmetry. The diagonal $H^{(0)}$ is realized in the $2\mathsf{Vec}_G^\tau$ factor as the subgroup $H \subseteq G$, and in the $\Sigma\mathcal{M}$ factor as the condensation surfaces such that topological lines living at their end form the crossed braided fusion category $\mathcal{M}_H^\times$.

**Minimal BCs.** For each choice of subgroup $H$, there are special minimal boundary conditions for which there are no genuine topological lines on the boundary, except the ones coming from the projection of bulk lines. For these we have

$$\mathcal{M} = \mathsf{Vec}, \qquad \mathcal{M}_H^\times = \mathsf{Vec}_H^\omega, \tag{5.10}$$

where $\omega \in H^3(H, U(1))$, which results in $\widetilde{\mathcal{M}}$ being the MTC associated to the 3d DW theory with $H$ gauge group and twist $\omega$. In particular, we have for all choices of $\omega$

$$\widetilde{\mathcal{M}}_1 = \mathsf{Rep}(H). \tag{5.11}$$

We denote such a minimal BC as

$$\mathfrak{B}^\omega_{\mathrm{Neu}_H} \, . \tag{5.12}$$

In fact, all these minimal BCs are gauge related to $\mathfrak{B}_{\mathrm{Dir}}$. We can obtain $\mathfrak{B}^\omega_{\mathrm{Neu}_H}$ by gauging the $H$ subgroup of the $G$ 0-form symmetry of $\mathfrak{B}_{\mathrm{Dir}}$ with discrete torsion $\omega$.

**Non-Minimal BCs Gauge-related to Minimal BCs.** More generally, the BCs gauge related to $\mathfrak{B}_{\mathrm{Dir}}$ are characterized by the condition

$$\widetilde{\mathcal{M}} = \mathcal{Z}(\widetilde{\mathcal{C}}) \, , \tag{5.13}$$

i.e. the MTC $\widetilde{\mathcal{M}}$ being the center of an $H$-graded fusion category $\widetilde{\mathcal{C}}$

$$\widetilde{\mathcal{C}} = \bigoplus_{h \in H} \mathcal{C}_h \, . \tag{5.14}$$

We can also identify the MTC $\mathcal{M}$ as

$$\mathcal{M} = \mathcal{Z}(\mathcal{C}) \, , \tag{5.15}$$

where $\mathcal{C}$ denotes the fusion category formed by the unit grade of $\widetilde{\mathcal{C}}$.

## 5.2 Gapped Phases

Let us now consider sandwich constructions involving the topological BCs discussed in the previous subsection. Let us first choose the symmetry boundary to be

$$\mathfrak{B}^{\mathrm{sym}} = \mathfrak{B}_{\mathrm{Dir}} \tag{5.16}$$

and the physical boundary to be

$$\mathfrak{B}^{\mathrm{phys}} = \mathfrak{B}^{\mathcal{M}'^\times_{H'}}_{\mathrm{Neu}_{H'}} \tag{5.17}$$

associated to a $(G, \tau)$-symmetric 3d TFT $\mathfrak{T}^{\mathcal{M}'^\times_{H'}}$. We know from the discussion of the previous subsection that in this case the result of the sandwich construction is the $(G, \tau)$-symmetric 3d TFT $\mathfrak{T}^{\mathcal{M}'^\times_{H'}}$ itself.

We now modify the symmetry boundary to be

$$\mathfrak{B}^{\mathrm{sym}} = \mathfrak{B}^{\mathcal{M}^\times_H}_{\mathrm{Neu}_H} \tag{5.18}$$

associated to a $(G, \tau)$-symmetric 3d TFT $\mathfrak{T}^{\mathcal{M}^\times_H}$. Recall that this $\mathfrak{B}^{\mathrm{sym}}$ is obtained from $\mathfrak{B}_{\mathrm{Dir}}$ as

$$\mathfrak{B}^{\mathrm{sym}} = \frac{\mathfrak{B}_{\mathrm{Dir}} \boxtimes \overline{\mathfrak{T}^{\mathcal{M}^\times_H}}}{G^{(0)}} \, , \tag{5.19}$$

i.e. by stacking $\mathfrak{B}_{\mathrm{Dir}}$ with $\overline{\mathfrak{T}^{\mathcal{M}_H^\times}}$ and then gauging the diagonal $G^{(0)}$ symmetry. Consequently, the result of the sandwich compactification for $\mathfrak{B}^{\mathrm{sym}}$ given by (5.18) can be obtained by beginning with the result of the sandwich compactification for $\mathfrak{B}^{\mathrm{sym}} = \mathfrak{B}_{\mathrm{Dir}}$, stacking it with $\overline{\mathfrak{T}^{\mathcal{M}_H^\times}}$ and then gauging the diagonal $G^{(0)}$ symmetry. Hence, the result of sandwich construction with $\mathfrak{B}^{\mathrm{sym}}$ given by (5.18) and $\mathfrak{B}^{\mathrm{phys}}$ given by (5.17) is the 3d TFT

$$\frac{\overline{\mathfrak{T}^{\mathcal{M}_H^\times}} \boxtimes \mathfrak{T}^{\mathcal{M}'^\times_{H'}}}{G^{(0)}} \, . \tag{5.20}$$

The symmetry associated to the boundary (5.18) can be expressed as

$$\mathcal{S} = \frac{2\mathsf{Vec}_G^\tau \boxtimes \mathcal{S}(\overline{\mathfrak{T}^{\mathcal{M}_H^\times}})}{G^{(0)}} \tag{5.21}$$

where $\mathcal{S}(\overline{\mathfrak{T}^{\mathcal{M}_H^\times}})$ is the multi-fusion 2-category formed by topological defects of the 3d TFT $\overline{\mathfrak{T}^{\mathcal{M}_H^\times}}$, which follows from the construction (5.19). Despite the appearance of a multi-fusion 2-category in this construction, the final category $\mathcal{S}$ is a fusion 2-category. From this construction, it is easy to deduce how $\mathcal{S}$ symmetry is realized on the 3d TFT (5.20). The $2\mathsf{Vec}_G^\tau$ part is realized by surface defects of $\mathfrak{T}^{\mathcal{M}'^\times_{H'}}$ factor implementing its $(G, \tau)$ symmetry, and the $\mathcal{S}(\overline{\mathfrak{T}^{\mathcal{M}_H^\times}})$ part is realized by the topological defects of the $\overline{\mathfrak{T}^{\mathcal{M}_H^\times}}$ factor. Finally we just gauge the diagonal $G^{(0)}$ and track the effect of this gauging on these topological defects of $\overline{\mathfrak{T}^{\mathcal{M}_H^\times}} \boxtimes \mathfrak{T}^{\mathcal{M}'^\times_{H'}}$.

We implement the analysis of this general section for abelian $G$ and trivial $\tau$ in the next section, and leave the detailed implementation for examples of non-abelian $G$ for a future work [58].

# 6 Gapped Boundary Conditions and Phases from $\mathcal{Z}(2\mathsf{Vec}_{\mathbb{A}})$

In this section, we purpose the (3+1)d Dijkgraaf-Witten theory corresponding to a general finite Abelian group $\mathbb{A}$ as the SymTFT. We concretely classify and characterize the gapped boundary conditions (both minimal and non-minimal) of this TFT and then use these to study gapped phases via the SymTFT sandwich construction.

**The SymTFT.** The topological surface and line defects of the (3+1)d $\mathbb{A}$ DW theory forms the Drinfeld center $\mathcal{Z}(2\mathsf{Vec}_{\mathbb{A}})$. Specifically there are topological (flux) surfaces corresponding to each group element $a \in \mathbb{A}$ which we denote by $\boldsymbol{Q}_2^a$ and topological (charge) lines labelled by representations $\hat{a} \in \widehat{\mathbb{A}} \equiv \hom(\mathbb{A}, U(1))$, which we denote by $\boldsymbol{Q}_1^{\hat{a}}$. These surfaces and lines braid with the braiding phase given by

$$B(\boldsymbol{Q}_2^a, \boldsymbol{Q}_1^{\hat{a}}) = \hat{a}(a) \, . \tag{6.1}$$

Additionally there are also condensation surface defects which arise from higher-gaugings of subgroups of $\mathsf{Rep}(\mathbb{A})$ with choices of discrete torsion on a the surfaces $\boldsymbol{Q}_2^a$. The condensation defects along with the lines and local operators on them that descend from $\boldsymbol{Q}_2^a$ in this manner form the Fusion 2-category $2\mathsf{Rep}(\mathbb{A})$. The complete set of topological defects of dimension-2 and lower can therefore be organized as

$$\mathcal{Z}(2\mathsf{Vec}_{\mathbb{A}}) = \boxplus_{a \in \mathbb{A}} \, 2\mathsf{Rep}(\mathbb{A}) \,. \tag{6.2}$$

The condensation defects will not play a significant role for the present purposes therefore we will suppress them in what follows.

## 6.1 Minimal Boundary Conditions

### 6.1.1 General Abelian Group $\mathbb{A}$

We again start with the so-called Dirichlet boundary condition which realizes the $\mathbb{A}^{(0)}$ 0-form symmetry. We can generate the other minimal boundary conditions by gauging subgroups $\mathbb{B} \subset \mathbb{A}$ with a choice of discrete torsion $\omega \in H^3(\mathbb{B}, U(1))$.

**Dirichlet Boundary Condition:** The Dirichlet BC as always realizes the 0-form symmetry that we start with, which in this case is $\mathbb{A}^{(0)}$. The boundary condition is specified by the following SymTFT topological defects having Dirichlet BC (i.e. can end)

$$\mathfrak{B}_{\text{Dir}} = \begin{cases} \boldsymbol{Q}_2^{\text{id}} \,, \\ \boldsymbol{Q}_1^{\hat{a}} \,, & \hat{a} \in \widehat{\mathbb{A}} \,. \end{cases} \tag{6.3}$$

We depict this by the quiche

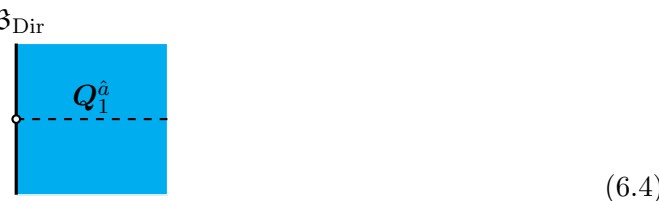

$$\tag{6.4}$$

The surface defects $\boldsymbol{Q}_2^a$, $a \in \mathbb{A}$ project (with Neumann BC) to the boundary and give rise to the 0-form symmetry generators

$$\boldsymbol{Q}_2^a \quad \longrightarrow \quad D_2^a \,, \qquad a \in \mathbb{A} \,, \tag{6.5}$$

which are the objects of $2\mathsf{Vec}_{\mathbb{A}}$. The topological lines can all end on the boundary. The end-point of a bulk line $\boldsymbol{Q}_1^{\hat{a}}$ is a non-genuine topological local operator, which we denote as

$$\partial \boldsymbol{Q}_1^{\hat{a}} = \widetilde{D}_0^{\hat{a}} \,. \tag{6.6}$$

As a consequence of the bulk braiding (6.1), $\widetilde{D}_0^{\hat{a}}$ is charged under the $\mathbb{A}^{(0)}$ symmetry on the boundary.

**Neumann Boundary Condition:**  Gauging the $\mathbb{A}^{(0)}$ symmetry on the Dirichlet boundary condition $\mathfrak{B}_{\mathrm{Dir}}$ with discrete torsion $\omega \in H^3(\mathbb{A}, U(1))$ results in the pure Neumann boundary condition which we denote by $\mathfrak{B}_{\mathrm{Neu},\omega}$. Upon such a gauging the non-genuine charged operators $\widetilde{D}_0^{\hat{a}}$ go into the twisted sector, i.e., they live at the end of a symmetry defect, and therefore $\boldsymbol{Q}_1^{\hat{a}}$ cannot end on $\mathfrak{B}_{\mathrm{Neu},\omega}$. The bulk lines therefore project to the boundary as

$$\boldsymbol{Q}_1^{\hat{a}} \quad \longrightarrow \quad D_1^{\hat{a}}, \ \hat{a} \in \widehat{\mathbb{A}}. \tag{6.7}$$

These fuse according to $\mathsf{Rep}(\mathbb{A})$, and generate the 1-form symmetry on $\mathfrak{B}_{\mathrm{Neu},\omega}$. Conversely, a line in the twisted sector of $D_2^a$ becomes an untwisted sector line charged under $\mathsf{Rep}(\mathbb{A})$ after gauging $\mathbb{A}$. In other words, $\boldsymbol{Q}_2^a$ can end on $\mathfrak{B}_{\mathrm{Neu},\omega}$ on a non-genuine line, which we denote by $\widetilde{D}_1^a$ that braids with $D_1^{\hat{a}}$ with the braiding phase

$$B(\widetilde{D}_1^a, D_1^{\hat{a}}) = B(\boldsymbol{Q}_2^a, \boldsymbol{Q}_1^{\hat{a}}) = \hat{a}(a). \tag{6.8}$$

To summarize the following bulk SymTFT defects can end on the boundary $\mathfrak{B}_{\mathrm{Neu},\omega}$

$$\mathfrak{B}_{\mathrm{Neu},\omega} = \begin{cases} \boldsymbol{Q}_2^a, \ a \in \mathbb{A} \\ \boldsymbol{Q}_1^1. \end{cases} \tag{6.9}$$

The quiche is

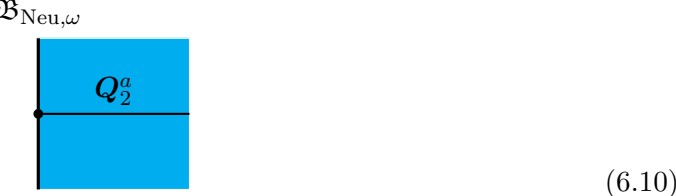

$$\tag{6.10}$$

Meanwhile, the ends of the surfaces give rise to (non-genuine) lines

$$\partial \boldsymbol{Q}_2^a = \widetilde{D}_1^a, \tag{6.11}$$

which have F-symbols determined by $\omega$ as

$$F(\widetilde{D}_1^{a_1}, \widetilde{D}_1^{a_2}, \widetilde{D}_1^{a_3}) = \omega(a_1, a_2, a_3), \tag{6.12}$$

and are charged under the 1-form symmetry. The category of topological defects on the boundary is $2\mathsf{Rep}(\mathbb{A})$ which is insensitive to the discrete torsion $\omega$. In contrast, the gapped phases arising via a SymTFT sandwich will depend sensitively on the choice of discrete torsion on both the symmetry and physical boundary.

**Partial Neumann Boundary Conditions:** The most general boundary conditions are obtained by starting from $\mathfrak{B}_{\text{Dir}}$ and gauging a subgroup $\mathbb{B} \subset \mathbb{A}$ of the 0-form symmetry with discrete torsion $\omega \in H^3(\mathbb{B}, U(1))$. We denote such a boundary by $\mathfrak{B}_{\text{Neu}(\mathbb{B}),\omega}$. When $\mathbb{B} = \mathbb{A}$, we drop the subgroup label for brevity. Following such a gauging, $D_2^b$ for $b \in \mathbb{B}$ become trivial or equivalently $\mathbf{Q}_2^b$ can end on $\mathfrak{B}_{\text{Neu}(\mathbb{B}),\omega}$ on a non-genuine line $\widetilde{D}_1^b$. As a consequence any SymTFT line that braids non-trivially with $\mathbf{Q}_2^b$ for any $b \in \mathbb{B}$ cannot end on $\mathfrak{B}_{\text{Neu}(\mathbb{B}),\omega}$. The topological defects with Dirichlet boundary conditions are

$$\mathfrak{B}_{\text{Neu}(\mathbb{B}),\omega} = \begin{cases} \mathbf{Q}_2^b, & b \in \mathbb{B} \\ \mathbf{Q}_1^{\hat{a}}, & \hat{a} \in \widehat{\mathbb{A}/\mathbb{B}}, \end{cases} \tag{6.13}$$

where $\widehat{\mathbb{A}/\mathbb{B}}$ is the subgroup of $\widehat{\mathbb{A}}$ comprising representations $\hat{a}$ that satisfy $\hat{a}(b) = 1$ for all $b \in \mathbb{B}$. The associated quiche is given by

$$\tag{6.14}$$

The lines $\mathbf{Q}_1^{\hat{a}}$ that carry representations that do not trivialize under a restriction to $\mathbb{B}$ generate a 1-form symmetry

$$\widehat{\mathbb{A}}/\widehat{\mathbb{A}/\mathbb{B}} \simeq \widehat{\mathbb{B}}, \tag{6.15}$$

on the $\mathfrak{B}_{\text{Neu}(\mathbb{B}),\omega}$. This can also be seen from the fact that the non-genuine local operator at the end of such a $\mathbf{Q}_1^{\hat{a}}$ was charged under the $\mathbb{B}$ 0-form symmetry on $\mathfrak{B}_{\text{Dir}}$ and therefore goes into the twisted sector upon gauging $\mathbb{B}$. The ends of the surfaces $\mathbf{Q}_2^b$ give rise to (non-genuine) lines

$$\partial \mathbf{Q}_2^b = \widetilde{D}_1^b, \tag{6.16}$$

which have F-symbols determined by $\omega$

$$F(\widetilde{D}_1^{b_1}, \widetilde{D}_1^{b_2}, \widetilde{D}_1^{b_3}) = \omega(b_1, b_2, b_3), \tag{6.17}$$

and are charged under the 1-form symmetry. The surfaces $\mathbf{Q}_2^a$ for $a \notin B$ project to the boundary as non-trivial surface defects which generate a 0-form symmetries $\mathbb{A}/\mathbb{B}$

$$\mathbf{Q}_2^a \quad \longrightarrow \quad D_2^a, \; a \notin \mathbb{B}. \tag{6.18}$$

To summarize, the boundary has a symmetry

$$(\mathbb{A}/\mathbb{B})^{(0)} \times \widehat{B}^{(1)}, \tag{6.19}$$

with possibly a mixed anomaly between the 0-form and 1-form symmetry due to the fact that $\mathbb{B}$ may not be a proper subgroup. More precisely consider that $\mathbb{B}$ sits in the short exact sequence

$$1 \longrightarrow \mathbb{B} \longrightarrow \mathbb{A} \longrightarrow \mathbb{A}/\mathbb{B} \longrightarrow 1, \tag{6.20}$$

with the extension class $\epsilon \in H^2(\mathbb{A}/\mathbb{B}, \mathbb{B})$. Then the anomaly can be described by the following (3+1)d topological action

$$S_{\text{anom}}^\beta = \int_{M_4} B_2 \cup \epsilon(A_1), \tag{6.21}$$

where $B_2 \in H^2(M_4, \widehat{\mathbb{B}})$ and $A_1 \in H^1(M_4, \mathbb{A}/\mathbb{B})$ are background gauge fields for the 1-form and 0-form symmetry respectively. Physically this anomaly encodes the fact that the fusion product $D_2^{a_1} \otimes D_2^{a_2}$ with $a_1, a_2 \in \mathbb{A}/\mathbb{B}$ is isomorphic to $D_2^{a_1 a_2}$ such that the isomorphism is implemented by the line $\widehat{D}_1^{\epsilon(a_1,a_2)}$ which braids non-trivially with the 1-form symmetry. We denote this fusion 2-category of topological defects on $\mathbb{B}_{\text{Neu}(\mathbb{B}),\omega}$ as

$$2\text{Vec}_{(\mathbb{A}/\mathbb{B})^{(0)} \times \mathbb{B}^{(1)}}^\beta, \quad \beta \in H^4((\mathbb{A}/\mathbb{B})^{(0)} \times \mathbb{B}^{(1)}, U(1)). \tag{6.22}$$

### 6.1.2 Example: $\mathbb{A} = \mathbb{Z}_4$

As a concrete example consider $\mathbb{A} = \mathbb{Z}_4 = \{a^m; a^4 = \text{id}\}$. Denote by $[g] = g$ the conjugacy classes for all $g \in \mathbb{Z}_4$. The stabilizer group for each element is $H_g = \mathbb{Z}_4$. The Drinfeld center (2.3) specialized to to this group is

$$\mathcal{Z}(2\text{Vec}_{\mathbb{Z}_4}) = 2\text{Rep}(\mathbb{Z}_4) \boxplus 2\text{Rep}(\mathbb{Z}_4) \boxplus 2\text{Rep}(\mathbb{Z}_4) \boxplus 2\text{Rep}(\mathbb{Z}_4). \tag{6.23}$$

Each is the stabilizer group of $a^m$, and gives rise to a simple object, i.e. topological surface operator,

$$\boldsymbol{Q}_2^{[a^m]} \equiv \boldsymbol{Q}_2^m, \qquad m = 0, \cdots, 3. \tag{6.24}$$

The topological lines, or 1-morphisms, are $\boldsymbol{Q}_1^{[a^m], \boldsymbol{R}_e}$ and for $m = 0$, $\boldsymbol{Q}_1^{[\text{id}], \boldsymbol{R}_e} =: \boldsymbol{Q}_1^e$. The minimal gapped boundary conditions are easily determined as follows: We again start with the so-called Dirichlet boundary condition, and for the minimal BC, gauge subgroups $H$ with discrete torsion $H^3(H, U(1))$. In the present case we can gauge a $\mathbb{Z}_2$ or all of the $\mathbb{Z}_4$, and in total we get seven minimal boundary conditions for the SymTFT $\mathcal{Z}(2\text{Vec}_{\mathbb{Z}_4})$ which one can group into three separate classes:

- The boundary $\mathfrak{B}_{\mathrm{Dir}}$ is given by

$$\mathfrak{B}_{\mathrm{Dir}} = \begin{cases} \boldsymbol{Q}_2^{\mathrm{id}} \\ \boldsymbol{Q}_1^1 \oplus \boldsymbol{Q}_1^\eta \oplus \boldsymbol{Q}_1^{\eta^2} \oplus \boldsymbol{Q}_1^{\eta^3} \,, \end{cases} \tag{6.25}$$

where $\boldsymbol{Q}_1^{\eta^i}$ are the lines generating a $\mathbb{Z}_4$, which have Dirichlet BC and $\eta = i$. The quiche for this BC is

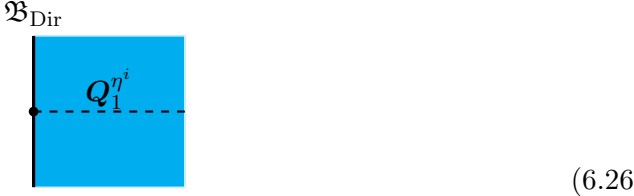

$$\tag{6.26}$$

The surfaces $\boldsymbol{Q}_2^m$ are projected parallel to the boundary, i.e. have Neumann boundary conditions

$$\boldsymbol{Q}_2^m \Big|_{\mathfrak{B}^{\mathrm{sym}}} = (D_2^m) \,, \tag{6.27}$$

and gives rise to an order four topological surface $D_2^m$ generating a $\mathbb{Z}_4$ 0-form symmetry.

- The BCs $\mathfrak{B}_{m^2}^\omega$ for a choice of discrete torsion $\omega \in H^3(\mathbb{Z}_2, U(1)) = \mathbb{Z}_2$ has the property that $\boldsymbol{Q}_1^e$ projects to an order two line $D_1^e$ while $\boldsymbol{Q}_2^m$ is projected to an order two surface $D_2^m$, and thus $\boldsymbol{Q}_1^{\eta^2}$ and $\boldsymbol{Q}_2^2$ can end on the boundary

$$\mathfrak{B}_{\mathrm{Neu}(\mathbb{Z}_2),\omega} = \begin{cases} \boldsymbol{Q}_2^{\mathrm{id}} \oplus \boldsymbol{Q}_2^2 \\ \boldsymbol{Q}_1^1 \oplus \boldsymbol{Q}_1^{\eta^2} \oplus \boldsymbol{Q}_1^{2,1} \oplus \boldsymbol{Q}_1^{2,\eta^2} \,. \end{cases} \tag{6.28}$$

The quiche is

$$\begin{array}{c} \mathfrak{B}_{\mathrm{Neu}(\mathbb{Z}_2),\omega} \\[4pt] \boxed{\begin{array}{c} \boldsymbol{Q}_2^2 \\[8pt] \boldsymbol{Q}_1^{\eta^2} \oplus \boldsymbol{Q}_1^{2,\eta^2} \end{array}} \end{array} \tag{6.29}$$

There are two minimal boundaries distinguished by the choice of discrete torsion, which determines the associator for the topological lines of $\mathcal{Z}(\mathsf{Vec}_{\mathbb{Z}_2}^\omega)$ living on the boundary. The identifications are[15]

$$\boldsymbol{Q}_2^0 \sim \boldsymbol{Q}_2^2 \sim D_2^{\mathrm{id}} \,, \qquad \boldsymbol{Q}_2^1 \sim \boldsymbol{Q}_2^3 \sim D_2^- \,. \tag{6.30}$$

---

[15]We will assume that in these identification it is always understood that $\boldsymbol{Q}_2$ is restricted to the boundary to then identify with the topological surface defects $D_2$ after the sandwich compactification.

which generates a $\mathbb{Z}_2$ 0-form symmetry. Likewise the lines are identified as

$$Q_1^\eta \sim Q_1^{\eta^3} \sim D_1^\eta \,, \tag{6.31}$$

which generates a $\mathbb{Z}_2^{(1)}$ 1-form symmetry on the boundary. These two boundaries carry a $\mathbb{Z}_2^{(0)} \times \mathbb{Z}_2^{(1)}$ symmetry with mixed anomaly

$$\int A_1^2 \cup B_2 \,, \tag{6.32}$$

where $A_1$ and $B_2$ are the background gauge fields corresponding to $\mathbb{Z}_2^{(0)}$ and $\mathbb{Z}_2^{(1)}$ symmetry. This arises in the symTFT bulk through the non-trivial braiding of $Q_2^2$ and $Q_1^\eta$.

- The boundary $\mathfrak{B}_{\mathrm{Neu},\omega}$ for $\omega \in H^3(\mathbb{Z}_4, U(1)) = \mathbb{Z}_4$

$$\mathfrak{B}_{\mathrm{Neu},\omega} = \begin{cases} Q_2^{\mathrm{id}} \oplus Q_2^1 \oplus Q_2^2 \oplus Q_2^3 \\ Q_1^1 \,, \end{cases} \tag{6.33}$$

has the property that $Q_1^\eta$ projects to an order four line $D_1^e$ while $Q_2^m$ can end on the boundary. The quiche is

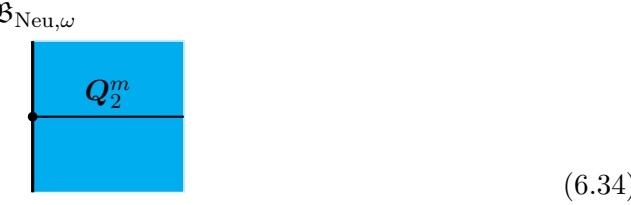

$$\tag{6.34}$$

There are four minimal boundaries distinguished by the choice of discrete torsion (when thinking about this as arising from gauging) or associator $\omega \in \{0, 1, 2, 3\} = H^3(\mathbb{Z}_4, U(1))$ for the topological lines of $\mathcal{Z}(\mathsf{Vec}_{\mathbb{Z}_4}^\omega)$ which carry a $\mathbb{Z}_4$ 1-form symmetry. Again, the choice of $\omega$ will determine the associators of the non-genuine lines that are the boundaries of $Q_2^m$.

| | Dir | Neu | $(\mathrm{Neu}, \nu)$ | $(\mathrm{Neu}(\mathbb{B}), \nu)$ |
|---|---|---|---|---|
| Dir : $2\mathrm{Vec}_\mathbb{A}$ | $\mathbb{A}^{(0)}$ SSB | $\mathbb{A}^{(0)}$ Trivial | $\mathbb{A}^{(0)}$ $\mathrm{SPT}_\nu$ | $(\mathbb{A}/\mathbb{B})^{(0)}$ SSB $\ \mathbb{B}^{(0)}$ $\mathrm{SPT}_\nu$ |
| Neu : $2\mathrm{Rep}_\mathbb{A}$ | $\widehat{\mathbb{A}}^{(1)}$ Trivial | $\widehat{\mathbb{A}}^{(1)}$ SSB $\ \mathrm{DW}_\mathbb{A}$ | $\widehat{\mathbb{A}}^{(1)}$ SSB $\ \mathrm{DW}_\mathbb{A}^\nu$ | $\widehat{\mathbb{B}}^{(1)}$ SSB $\ \mathrm{DW}_\mathbb{B}^\nu$ |
| $(\mathrm{Neu}, \omega)$ : $2\mathrm{Rep}_\mathbb{A}$ | $\widehat{\mathbb{A}}^{(1)}$ Trivial | $\widehat{\mathbb{A}}^{(1)}$ SSB $\ \mathrm{DW}_\mathbb{A}^{\omega^{-1}}$ | $\widehat{\mathbb{A}}^{(1)}$ SSB $\ \mathrm{DW}_\mathbb{A}^{\omega^{-1}\cdot\nu}$ | $\widehat{\mathbb{B}}^{(1)}$ SSB $\ \mathrm{DW}_\mathbb{B}^{\omega^{-1}\cdot\nu}$ |
| $(\mathrm{Neu}(\mathbb{D}), \omega)$ : $2\mathrm{Vec}^\beta\left((\mathbb{A}/\mathbb{D})^{(0)} \times \widehat{\mathbb{D}}^{(1)}\right)$ | $(\mathbb{A}/\mathbb{D})^{(0)}$ SSB | $\widehat{\mathbb{D}}^{(1)}$ SSB $\ \mathbb{A}/\mathbb{D} \ \mathrm{SPT} \boxtimes \mathrm{DW}_\mathbb{D}^{\omega^{-1}}$ | $\mathbb{D}^{(1)}$ SSB $\ \left(\mathbb{A}/\mathbb{D} \ \mathrm{SPT} \boxtimes \mathrm{DW}_\mathbb{D}^{\omega^{-1}\cdot\nu}\right)$ | $\left(\mathbb{B}/(\mathbb{B}\cap\mathbb{D}) \ \mathrm{SPT}\right) \boxtimes \mathrm{DW}_{\mathbb{D}\cap\mathbb{B}}^{\omega^{-1}\cdot\nu}$ |

Table 1: The minimal gapped Phases for the symmetries determined by the vertical axis, as well as the associated symmmetry $\mathcal{S}$. The gapped phases are characterized by the physical boundary condition, as specified horizontally. For a detailed discussion of the SPT phases see the main text.

## 6.2 Minimal Gapped Phases

Having described the minimal boundaries of the $\mathbb{A}$ SymTFT and their associated symmetry categories, we can now study the gapped phases obtained by a SymTFT sandwich with both the physical and symmetry boundaries $\mathfrak{B}^{\mathrm{phys}}$ and $\mathfrak{B}^{\mathrm{sym}}$ chosen from the set of minimal gapped boundary conditions. We will denote the phases by

$$(\mathfrak{B}^{\mathrm{sym}}, \mathfrak{B}^{\mathrm{phys}}). \tag{6.35}$$

We summarize them in table 1.

### 6.2.1 Minimal Phases for $\mathbb{A}^{(0)}$

Starting with $\mathfrak{B}^{\mathrm{sym}} = \mathfrak{B}_{\mathrm{Dir}}$ such that the symmetry is $\mathcal{S} = 2\mathsf{Vec}_{\mathbb{A}}$, we find the following phases:

- $\mathbb{A}^{(0)}$ **SSB Phase:** $(\mathfrak{B}_{\mathbf{Dir}}, \mathfrak{B}_{\mathbf{Dir}})$. The phase where the $\mathbb{A}^{(0)}$ symmetry is spontaneously broken to the trivial group corresponds to SymTFT sandwich $(\mathfrak{B}_{\mathrm{Dir}}, \mathfrak{B}_{\mathrm{Dir}})$, depicted as

$$\tag{6.36}$$

In the SymTFT pictures we only show the genuine order parameters, not the twisted sector ones. The order parameters for this phase are provided by the compactification of the representation lines $Q_1^{\hat{a}}$ that end on $\mathfrak{B}^{\mathrm{phys}}$. Since these lines can also end on $\mathfrak{B}^{\mathrm{sym}}$, after compactifying the SymTFT, they become topological local operators denoted by $\mathcal{O}^{\hat{a}}$. These are charged under $\mathbb{A}^{(0)}$ and transform as

$$D_2^a : \mathcal{O}^{\hat{a}} \longmapsto \hat{a}(a)\mathcal{O}^{\hat{a}} . \tag{6.37}$$

One can construct idempotent linear combinations of these local operators which are the projectors into the different vacua of the SSB phase. The vacua have the form

$$v_a = \frac{1}{|\mathbb{A}|} \sum_{\hat{a} \in \widehat{\mathbb{A}}} \hat{a}(a)\mathcal{O}^{\hat{a}} , \tag{6.38}$$

which transform as the regular representation space for $\mathbb{A}$, i.e.

$$D_2^a : v_{a'} \to v_{aa'} . \tag{6.39}$$

- $\mathbb{A}^{(0)}$ **Preserving SPT Phase:** $(\mathfrak{B}_{\mathbf{Dir}}, \mathfrak{B}_{\mathbf{Neu},\omega})$. The phase where $\mathbb{A}^{(0)}$ symmetry is spontaneously preserved is given by picking a Neumann boundary condition $\mathfrak{B}_{\mathrm{Neu},\omega}$ for the physical boundary, i.e., by the SymTFT sandwich

$$\begin{array}{ccc} \mathfrak{B}_{\mathrm{Dir}} & \mathfrak{B}_{\mathrm{Neu},\omega} & \mathfrak{T}^{\mathcal{S}} \\ & = & \end{array} \tag{6.40}$$

The bulk SymTFT surfaces $\boldsymbol{Q}_2^a$ which can end on $\mathfrak{B}_{\mathrm{Neu},\omega}$ provide the set of order parameters. Since none of these can end on the symmetry they all correspond to twisted sector lines after compactification. We denote the twisted sector line obtained from $\boldsymbol{Q}_2^a$ upon compactifying the SymTFT as $\widetilde{\mathcal{L}}^a$. These lines have a non-trivial F-symbol

$$F(\widetilde{\mathcal{L}}^{a_1}, \widetilde{\mathcal{L}}^{a_2}, \widetilde{\mathcal{L}}^{a_3}) = \omega(a_1, a_2, a_3), \tag{6.41}$$

coming from the ends of the lines on the physical boundary. In this phase, since all the symmetry defects $D_2^a$ are isomorphic to the identity, it corresponds to an $\mathbb{A}^{(0)}$ symmetry preserving phase. Meanwhile since the isomorphism is provided by $\widetilde{\mathcal{L}}^a$ which has a non-trivial associator, it corresponds to an SPT labelled by $\omega \in H^3(\mathbb{A}, U(1))$.

- **Partial Symmetry Breaking SPT Phase:** $(\mathfrak{B}_{\mathbf{Dir}}, \mathfrak{B}_{\mathbf{Neu}(\mathbb{B}),\omega})$. To obtain a partial symmetry broken phase, we pick the SymTFT sandwich

$$\left(\mathfrak{B}_{\mathrm{Dir}}, \mathfrak{B}_{\mathrm{Neu}(\mathbb{B}),\omega}\right). \tag{6.42}$$

The sandwich picture is

$$\begin{array}{ccc} \mathfrak{B}_{\mathrm{Dir}} & \mathfrak{B}_{\mathrm{Neu}(\mathbb{B}),\omega} & \mathfrak{T}^{\mathcal{S}} \\ Q_1^{\hat{a} \in \widehat{\mathbb{A}/\mathbb{B}}} & = & \mathcal{O}^{\hat{a}} \end{array} \tag{6.43}$$

Only the representation lines $Q_1^{\hat{a}}$ that braid trivially with the $\boldsymbol{Q}_2^b$ for $b \in \mathbb{B}$ end on $\mathfrak{B}^{\mathrm{phys}}$, these are indicated by $\hat{a} \in \widehat{\mathbb{A}/\mathbb{B}}$ in the sandwich. After compactifying the SymTFT, these become local order parameters $\mathcal{O}^{\hat{a}}$ that transform trivially under $\mathbb{B}^{(0)} \subset \mathbb{A}^{(0)}$ but non-trivially under the quotient $(\mathbb{A}/\mathbb{B})^{(0)}$. These local operators implement a spontaneously symmetry breaking to $\mathbb{B}$, i.e., $\mathbb{B} \subset \mathbb{A}$ is spontaneously preserved in this gapped phase.

There are $|\mathbb{A}/\mathbb{B}|$ vacua that correspond to idempotent combinations of the topological local operators. The bulk surfaces $\boldsymbol{Q}_2^b$ end on $\mathfrak{B}^{\mathrm{phys}}$ and become non-genuine lines attached to $D_2^a$ on $\mathfrak{B}^{\mathrm{sym}}$. Upon compactifying the SymTFT, these become twisted sector order parameters $\widehat{L}^b$, which have the associator

$$F(\widetilde{\mathcal{L}}^{b_1}, \widetilde{\mathcal{L}}^{b_2}, \widetilde{\mathcal{L}}^{b_3}) = \omega(b_1, b_2, b_3). \tag{6.44}$$

Therefore this corresponds to the gapped phase where each of the $|\mathbb{A}/\mathbb{B}|$ SSB vacua realize a $\mathbb{B}$ SPT labelled by $\omega \in H^3(\mathbb{B}, U(1))$.

### 6.2.2 Minimal Phases for $\widehat{\mathbb{A}}^{(1)}$

Starting with $\mathfrak{B}^{\mathrm{sym}} = \mathfrak{B}_{\mathrm{Neu},\omega}$ such that the symmetry is $\mathcal{S} = 2\mathsf{Rep}_{\mathbb{A}}$, we find the following phases:

- **Trivial Phase:** $(\mathfrak{B}_{\mathbf{Neu},\omega}, \mathfrak{B}_{\mathbf{Dir}})$. The $2\mathsf{Rep}(\mathbb{A})$ trivial phase is obtained from the SymTFT sandwich

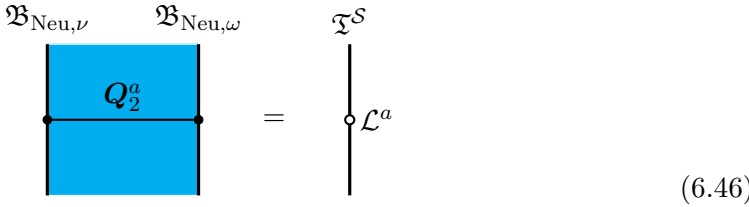

$$\tag{6.45}$$

The twisted sector order parameters for the IR gapped phase are given by the lines $\boldsymbol{Q}_1^{\hat{a}}$ that end on $\mathfrak{B}^{\mathrm{phys}}$ and become twisted sector local operators on the symmetry boundary. After compactifying the SymTFT, these order paramaters become twisted sector local operators $\widetilde{\mathcal{O}}^{\hat{a}}$ attached to the 1-form symmetry generator $D_1^{\hat{a}}$. This implies that in this phase, all the 1-form symmetry generators are isomorphic to the identity and this is therefore the $2\mathsf{Rep}(\mathbb{A})$ trivial phase.

- **SSB Phase:** $(\mathfrak{B}_{\mathbf{Neu},\omega}, \mathfrak{B}_{\mathbf{Neu},\nu})$. The 1-form symmetry breaking phase is constructed from the SymTFT sandwich

$$\tag{6.46}$$

where $\omega, \nu \in H^3(\mathbb{A}, U(1))$. The bulk surfaces $\boldsymbol{Q}_2^a$ for all $a \in \mathbb{A}$ can end on both the boundaries and therefore become genuine topological lines that serve as order parameters

for this gapped phase upon compactifying the SymTFT. We denote the line obtained from compactifying $\boldsymbol{Q}_2^a$ stretched between the two boundaries as $\mathcal{L}^a$. These lines are charged under the $\widehat{\mathbb{A}}$ 1-form symmetry such that the braiding phase is given by

$$B(\mathcal{L}^a, D_1^{\hat{a}}) = B(\boldsymbol{Q}_2^a, \boldsymbol{Q}_1^{\hat{a}}) = \hat{a}(a) \,. \tag{6.47}$$

Furthermore, the associators of $\mathcal{L}^a$ may be non-trivial and depends on $\omega$ and $\nu$ as

$$F(\mathcal{L}^{a_1}, \mathcal{L}^{a_2}, \mathcal{L}^{a_3}) = \omega^{-1}(a_1, a_2, a_3)\nu(a_1, a_2, a_3) \,. \tag{6.48}$$

Together the symmetry and charged lines form the modular tensor category

$$\mathcal{Z}(\mathsf{Vec}_{\mathbb{A}}^{\omega^{-1} \cdot \nu}) \,, \tag{6.49}$$

which is the 3d DW theory for the group $\mathbb{A}$ with a topological action given by $\omega^{-1} \cdot \nu$. Correspondingly all the minimal $2\mathsf{Rep}(\mathbb{A})$ SSB phases are 3d $\mathbb{A}$ DW theories which are minimal modular completions of the $2\mathsf{Rep}(\mathbb{A})$ symmetry and are constructed by picking different choices of $\nu$.

- **Partial Symmetry Breaking Phase:** $(\mathfrak{B}_{\mathbf{Neu},\omega}, \mathfrak{B}_{\mathbf{Neu}(\mathbb{B}),\nu})$.

  The $2\mathsf{Rep}_{\mathbb{A}}$ partial symmetry breaking phase is constructed from the SymTFT sandwich

$$\tag{6.50}$$

where $\nu \in H^3(\mathbb{B}, U(1))$. In this gapped phase, the lines $\boldsymbol{Q}_1^{\hat{a}}$ for all $\hat{a}$ in $\widehat{\mathbb{A}/\mathbb{B}}$ and $\boldsymbol{Q}_2^b$ for all $b \in \mathbb{B}$ provide the order parameters. Firstly, the bulk lines $\boldsymbol{Q}_1^{\hat{a}}$ for $\hat{a} \in \widehat{\mathbb{A}/\mathbb{B}}$ become twisted sector local operators $\widetilde{\mathcal{O}}^{\hat{a}}$ which are attached to the 1-form symmetry generators $D_1^{\hat{a}}$. Therefore the 1-form symmetry subgroup $\widehat{\mathbb{A}/\mathbb{B}}$ is trivial as all its generators are isomorphic to the identity line in the corresponding IR TQFT. Meanwhile the SymTFT surfaces $\boldsymbol{Q}_2^b$ become topological line operators $\mathcal{L}^b$ which are charged under the symmetry generators $D_1^{\hat{a}'}$ with $\hat{a}' \in \widehat{\mathbb{A}}/\widehat{\mathbb{A}/\mathbb{B}} \simeq \widehat{\mathbb{B}}$. The resulting phase is the $\widehat{\mathbb{B}}$ 1-form SSB phase which nothing but the 3d $\mathbb{B}$ DW theory. The charged lines $\mathcal{L}^b$ have a potentially non-trivial associator given by $\omega^{-1} \cdot \nu$ evaluated on $\mathbb{B}$.

### 6.2.3 Minimal Phases for $(\mathbb{A}/\mathbb{B})^{(0)} \times \widehat{\mathbb{B}}^{(1)}$ with Anomaly

Starting with $\mathfrak{B}^{\mathrm{sym}} = \mathfrak{B}_{\mathrm{Neu}(\mathbb{B}),\omega}$ such that the symmetry is $\mathcal{S} = 2\mathsf{Vec}^{\beta}((\mathbb{A}/\mathbb{B})^{(0)} \times \widehat{\mathbb{B}}^{(1)})$, where $\beta$ is the anomaly given by the anomaly action (6.21), we find the following phases:

- **0-form SSB Phase:** $(\mathfrak{B}_{\mathrm{Neu}(\mathbb{B}),\omega}, \mathfrak{B}_{\mathrm{Dir}})$. The phase where the 0-form subsymmetry is completely broken corresponds to the SymTFT sandwich

$$ \tag{6.51} $$

The order parameters are given by all the bulk SymTFT lines as these can end on the physical boundary. The bulk SymTFT lines

$$ \boldsymbol{Q}_1^{\hat{a}}, \quad \hat{a} \in \widehat{\mathbb{A}/\mathbb{B}}, \tag{6.52} $$

can end on the symmetry boundary as well and therefore become topological local operators

$$ \mathcal{O}^{\hat{a}}, \quad \hat{a} \in \widehat{\mathbb{A}/\mathbb{B}}, \tag{6.53} $$

upon the SymTFT compactification. These are precisely the complete set of charged local operators for $(\mathbb{A}/\mathbb{B})^{(0)}$ and therefore play the role of 0-form SSB order parameters. The remaining bulk lines

$$ \boldsymbol{Q}_1^{\hat{a}}, \quad \hat{a} \in \widehat{\mathbb{A}}/\widehat{\mathbb{A}/\mathbb{B}} \simeq \widehat{\mathbb{B}}, \tag{6.54} $$

become the non-genuine local operators

$$ \widetilde{\mathcal{O}}^{\hat{a}}, \quad \hat{a} \in \widehat{\mathbb{A}}/\widehat{\mathbb{A}/\mathbb{B}} \simeq \widehat{\mathbb{B}}, \tag{6.55} $$

upon compactifying the SymTFT. Specifically the operator $\widetilde{\mathcal{O}}^{\hat{a}}$ provides a topological end of the symmetry defect $D_1^{\hat{a}}$ in the IR TQFT describing this gapped phase. Therefore the 1-form symmetry remains unbroken.

- **1-form SSB Phases:** $(\mathfrak{B}_{\mathrm{Neu}(\mathbb{B}),\omega}, \mathfrak{B}_{\mathrm{Neu},\nu})$. Various phases that spontaneously break the 1-form symmetry $\widehat{\mathbb{B}}^{(1)}$ and spontaneously preserve the 0-form symmetry $(\mathbb{A}/\mathbb{B})^{(0)}$

are given by the SymTFT sandwich

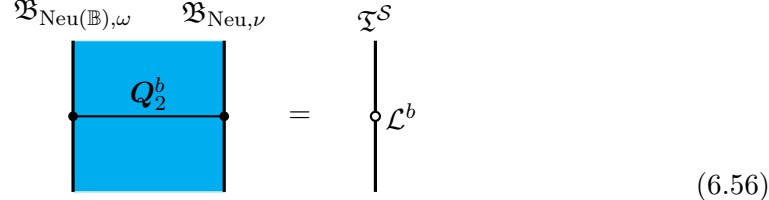

$$(6.56)$$

The order parameters are provided by the bulk SymTFT surfaces $\boldsymbol{Q}_2^a$ for all $a \in \mathbb{A}$ as all of these can end on $\mathfrak{B}^{\text{phys}}$. From these the bulk surfaces with $b \in \mathbb{B}$ can also end on the symmetry boundary and therefore become genuine topological lines in the IR TQFT describing this gapped phase. We denote these lines as $\mathcal{L}^b$. These lines braid with the 1-form symmetry defects $D_1^{\hat{b}}$ with $\hat{b} \in \widehat{\mathbb{B}}$ with the braiding phase $\hat{b}(b)$. Additionally these lines also have a non-trivial associators given by the restriction of $\omega^{-1} \cdot \nu$ to $\mathbb{B}$.

The bulk surfaces $\boldsymbol{Q}_2^a$ for $a \in \mathbb{A}/\mathbb{B}$ provide non-genuine lines $\widetilde{\mathcal{L}}^a$ upon compactifying the SymTFT. This implies that the 0-form symmetry generators are isomorphic to the identity surface in the IR TQFT and therefore the $(\mathbb{A}/\mathbb{B})^{(0)}$ symmetry is preserved.

The lines $\mathcal{L}$ braid non-trivially, given by the pull-back of the class $\nu \in H^3(\mathbb{A}, U(1))$ to $\mathbb{A}/\mathbb{B}$. This pullback is specified by the extension class of the short exact sequence

$$1 \to \mathbb{B} \to \mathbb{A} \to \mathbb{A}/\mathbb{B} \to 1 \tag{6.57}$$

which is the group cohomology class $\alpha \in H^2(\mathbb{A}/\mathbb{B}, \mathbb{B})$. The cohomology groups $H^3(G, U(1))$ are related by a spectral squence [67–69], which starts with $E_2^{p,q} = H^p(\mathbb{A}/\mathbb{B}, H^q(\mathbb{B}, U(1)))$. The first differential is computed by (Theorem 4 in [67])

$$d_2 : E_2^{p,1} = H^p(\mathbb{A}/\mathbb{B}, \text{Hom}(\mathbb{B}, U(1))) \to E_2^{p+2,0} = H^{p+2}(\mathbb{A}/\mathbb{B}, U(1))$$
$$\beta \mapsto -\alpha \cup \beta \, . \tag{6.58}$$

This provides the explicit connection between the discrete torsions for $\mathbb{B}$ and $\mathbb{A}/\mathbb{B}$ and the extension class $\alpha$, with the discrete torsion for $\mathbb{A}$. This relation also provides the pull-back of the class $\nu$ to $\mathbb{A}/\mathbb{B}$, which characterizes the SPT for the preserved 0-form symmetry $\mathbb{A}/\mathbb{B}$.

- **Partial Symmetry Broken Phases** $(\mathfrak{B}_{\text{Neu}(\mathbb{B}),\omega}, \mathfrak{B}_{\text{Neu}(\mathbb{D}),\nu})$: The SymTFT for these phases are

$$\mathfrak{B}_{\mathrm{Neu}(\mathbb{B}),\omega} \quad \mathfrak{B}_{\mathrm{Neu}(\mathbb{D}),\nu} \qquad \mathfrak{T}^{\mathcal{S}}$$

$$\tag{6.59}$$

The order parameters for the corresponding gapped phase are given by the set of bulk SymTFT surfaces and lines that can end on $\mathfrak{B}^{\mathrm{phys}}$. These are

$$\boldsymbol{Q}_2^d, \quad d \in \mathbb{D}, \qquad \boldsymbol{Q}_1^{\hat{a}}, \quad \hat{a} \in \widehat{\mathbb{A}/\mathbb{D}}. \tag{6.60}$$

First we discuss the order parameters corresponding to the bulk surfaces. Among these, a subset corresponding to $d \in \mathbb{D} \cap \mathbb{B}$ can also end on $\mathfrak{B}^{\mathrm{sym}}$ and therefore become genuine lines after the SymTFT compactification. We denote these lines as $\mathcal{L}^d$. These lines have an associator given by $\omega^{-1} \cdot \nu$ restricted to $\mathbb{D} \cap \mathbb{B}$.

The remaining surfaces which are $\boldsymbol{Q}_2^d$ for $d \in \mathbb{D}/(\mathbb{D} \cap \mathbb{B})$ become non-genuine lines $\widetilde{\mathcal{L}}^d$ upon the SymTFT compactification. This implies that the 0-form symmetry generators $D_2^b$ for $b \in \mathbb{D}/(\mathbb{D} \cap \mathbb{B})$ are isomorphic to the identity surface in the IR TQFT and therefore the 0-form symmetry subgroup $\mathbb{D}/(\mathbb{D} \cap \mathbb{B}) \subset \mathbb{A}/\mathbb{B}$ is preserved. The class $\nu$ restricts to $\mathbb{D}/(\mathbb{D} \cap \mathbb{B})$ and gives rise to the associators of the lines $\widetilde{\mathcal{L}}^d$ with $d \in \mathbb{D}/(\mathbb{D} \cap \mathbb{B})$. This class determines the SPT for the preserved 0-form symmetry.

Moving on to the order parameters obtained from the SymTFT lines, firstly the lines $\boldsymbol{Q}_1^{\hat{a}}$ for $\hat{a} \in (\widehat{\mathbb{A}/\mathbb{B}}) \cap (\widehat{\mathbb{A}/\mathbb{D}})$ can end on the symmetry boundary as well and therefore provides a topological local operator that is a 0-form symmetry breaking order parameter. The number of vacua are equal to the number of topological local operators which is

$$\left| (\widehat{\mathbb{A}/\mathbb{B}}) \cap (\widehat{\mathbb{A}/\mathbb{D}}) \right|. \tag{6.61}$$

Meanwhile the bulk lines

$$\boldsymbol{Q}_1^{\hat{a}}, \qquad \hat{a} \in \widehat{\mathbb{A}}/(\widehat{\mathbb{A}/\mathbb{B}}) \cap (\widehat{\mathbb{A}/\mathbb{D}}), \tag{6.62}$$

do not end on the symmetry boundary and therefore trivialize the corresponding 1-form symmetry. To summarize the SymTFT sandwich $(\mathfrak{B}_{\mathrm{Neu}(\mathbb{B}),\omega}, \mathfrak{B}_{\mathrm{Neu}(\mathbb{D}),\nu})$ describes a gapped phase where the 0-form symmetry is broken down to $\mathbb{D}/(\mathbb{D} \cap \mathbb{B})$ and each of these realizes an MTC which is the 3d DW $\mathbb{B} \cap \mathbb{D}$ theory with the topological action given by the restriction of $\omega \cdot \nu^{-1}$ restricted to $\mathbb{B} \cap \mathbb{D}$.

|  | Dir | $\mathrm{Neu}^{\mathbb{Z}_2}_+$ | $\mathrm{Neu}^{\mathbb{Z}_2}_-$ | $\mathrm{Neu}^{\mathbb{Z}_4}_1$ | $\mathrm{Neu}^{\mathbb{Z}_4}_\omega$ | $\mathrm{Neu}^{\mathbb{Z}_4}_{\omega^2}$ | $\mathrm{Neu}^{\mathbb{Z}_4}_{\omega^3}$ |
|---|---|---|---|---|---|---|---|
| Dir | $\mathbb{Z}^{(0)}_4$ SSB | $\mathbb{Z}^{(0)}_2$ SSB $\mathbb{Z}^{(0)}_2$ SPT$_+$ | $\mathbb{Z}^{(0)}_2$ SSB $\mathbb{Z}^{(0)}_2$ SPT$_-$ | $\mathbb{Z}^{(0)}_4$ SPT$_1$ | $\mathbb{Z}^{(0)}_4$ SPT$_\omega$ | $\mathbb{Z}^{(0)}_4$ SPT$_{\omega^2}$ | $\mathbb{Z}^{(0)}_4$ SPT$_{\omega^3}$ |
| $\mathrm{Neu}^{\mathbb{Z}_2}_+$ | $\mathbb{Z}^{(0)}_2$ SSB | $\mathbb{Z}_2$ DW$^+$ | $\mathbb{Z}_2$ DW$^-$ | $\mathbb{Z}^{(1)}_2$ SSB $\mathbb{Z}^{(0)}_2$ SPT $\boxtimes$ $\mathbb{Z}_2$ DW$^+$ | $\mathbb{Z}^{(1)}_2$ SSB $\mathbb{Z}^{(0)}_2$ SPT $\boxtimes$ $\mathbb{Z}_2$ DW$^{\omega^3}$ | $\mathbb{Z}^{(1)}_2$ SSB $\mathbb{Z}^{(0)}_2$ SPT $\boxtimes$ $\mathbb{Z}_2$ DW$^-$ | $\mathbb{Z}^{(1)}_2$ SSB $\mathbb{Z}^{(0)}_2$ SPT $\boxtimes$ $\mathbb{Z}_2$ DW$^{\omega}$ |
| $\mathrm{Neu}^{\mathbb{Z}_2}_-$ | $\mathbb{Z}^{(0)}_2$ SSB | $\mathbb{Z}_2$ DW$^-$ | $\mathbb{Z}_2$ DW$^+$ | $\mathbb{Z}^{(1)}_2$ SSB $\mathbb{Z}^{(0)}_2$ SPT $\boxtimes$ $\mathbb{Z}_2$ DW$^-$ | $\mathbb{Z}^{(1)}_2$ SSB $\mathbb{Z}^{(0)}_2$ SPT $\boxtimes$ $\mathbb{Z}_2$ DW$^{\omega}$ | $\mathbb{Z}^{(1)}_2$ SSB $\mathbb{Z}^{(0)}_2$ SPT $\boxtimes$ $\mathbb{Z}_2$ DW$^+$ | $\mathbb{Z}^{(1)}_2$ SSB $\mathbb{Z}^{(0)}_2$ SPT $\boxtimes$ $\mathbb{Z}_2$ DW$^{\omega^3}$ |
| $\mathrm{Neu}^{\mathbb{Z}_4}_1$ | $\mathbb{Z}^{(1)}_4$ Trivial | $\mathbb{Z}^{(1)}_2$ SSB $\mathbb{Z}_2$ DW$^+$ | $\mathbb{Z}^{(1)}_2$ SSB $\mathbb{Z}_2$ DW$^-$ | $\mathbb{Z}^{(1)}_4$ SSB $\mathbb{Z}_4$ DW$^1$ | $\mathbb{Z}^{(1)}_4$ SSB $\mathbb{Z}_4$ DW$^{\omega}$ | $\mathbb{Z}^{(1)}_4$ SSB $\mathbb{Z}_4$ DW$^{\omega^2}$ | $\mathbb{Z}^{(1)}_4$ SSB $\mathbb{Z}_4$ DW$^{\omega^3}$ |
| $\mathrm{Neu}^{\mathbb{Z}_4}_\omega$ | $\mathbb{Z}^{(1)}_4$ Trivial | $\mathbb{Z}^{(1)}_2$ SSB $\mathbb{Z}_2$ DW$^{\omega^3}$ | $\mathbb{Z}^{(1)}_2$ SSB $\mathbb{Z}_2$ DW$^{\omega}$ | $\mathbb{Z}^{(1)}_4$ SSB $\mathbb{Z}_4$ DW$^{\omega^3}$ | $\mathbb{Z}^{(1)}_4$ SSB $\mathbb{Z}_4$ DW$^1$ | $\mathbb{Z}^{(1)}_4$ SSB $\mathbb{Z}_4$ DW$^{\omega}$ | $\mathbb{Z}^{(1)}_4$ SSB $\mathbb{Z}_4$ DW$^{\omega^2}$ |
| $\mathrm{Neu}^{\mathbb{Z}_4}_{\omega^2}$ | $\mathbb{Z}^{(1)}_4$ Trivial | $\mathbb{Z}^{(1)}_2$ SSB $\mathbb{Z}_2$ DW$^-$ | $\mathbb{Z}^{(1)}_2$ SSB $\mathbb{Z}_2$ DW$^+$ | $\mathbb{Z}^{(1)}_4$ SSB $\mathbb{Z}_4$ DW$^{\omega^2}$ | $\mathbb{Z}^{(1)}_4$ SSB $\mathbb{Z}_4$ DW$^{\omega^3}$ | $\mathbb{Z}^{(1)}_4$ SSB $\mathbb{Z}_4$ DW$^1$ | $\mathbb{Z}^{(1)}_4$ SSB $\mathbb{Z}_4$ DW$^{\omega}$ |
| $\mathrm{Neu}^{\mathbb{Z}_4}_{\omega^3}$ | $\mathbb{Z}^{(1)}_4$ Trivial | $\mathbb{Z}^{(1)}_2$ SSB $\mathbb{Z}_2$ DW$^{\omega}$ | $\mathbb{Z}^{(1)}_2$ SSB $\mathbb{Z}_2$ DW$^{\omega^3}$ | $\mathbb{Z}^{(1)}_4$ SSB $\mathbb{Z}_4$ DW$^{\omega}$ | $\mathbb{Z}^{(1)}_4$ SSB $\mathbb{Z}_4$ DW$^{\omega^2}$ | $\mathbb{Z}^{(1)}_4$ SSB $\mathbb{Z}_4$ DW$^{\omega^3}$ | $\mathbb{Z}^{(1)}_4$ SSB $\mathbb{Z}_4$ DW$^1$ |

Table 2: Symmetries determined by the vertical axis while the physical boundary the horizontal axis. SPT = symmetry-protected topological phase, SSB = spontaneous symmetry breaking, DW$^\alpha$= Dijkgraaf-Witten with twist $\alpha$. The groups listed that are not SSB are the preserved symmetries. As $\mathbb{Z}_2$ is a subgroup of $\mathbb{Z}_4$, in this context, we have DW$^+$ = DW$^1$ and DW$^-$ = DW$^{\omega^2}$.

### 6.2.4  Example: $\mathbb{A} = \mathbb{Z}_4$

Again, we illustrate the general abelian group case with $\mathbb{A} = \mathbb{Z}_4$. Lattice realizations of these gapped phases and their gaugings were studied in [70].

**Gapped Phases for** $2\mathsf{Vec}_{\mathbb{Z}_4}$. We can use $\mathfrak{B}_{\mathrm{Dir}}$ as the symmetry boundary for $\mathbb{Z}_4$ 0-form symmetry. Using the three classes of boundaries as physical boundaries respectively, we obtain three classes of $(2+1)$d $\mathbb{Z}_4$ 0-form symmetric gapped phases which are minimal:

- For $\mathfrak{B}^{\mathrm{phys}} = \mathfrak{B}_{\mathrm{Dir}}$, we have a $\mathbb{Z}_4^{(0)}$ SSB phase,

$$
\begin{array}{ccc}
\mathfrak{B}_{\mathrm{Dir}} \quad \mathfrak{B}_{\mathrm{Dir}} & & \mathfrak{T}^{\mathcal{S}} \\
\underset{Q_1^{e^i}}{\boxed{\phantom{QQQ}}} & = & \big|\; D_0^{e^i}
\end{array}
\tag{6.63}
$$

with $i \in \{0, 1, 2, 3\}$, whose underlying 3d TFT can be expressed as

$$
\mathfrak{T} = \mathrm{Triv} \oplus \mathrm{Triv} \oplus \mathrm{Triv} \oplus \mathrm{Triv}.
\tag{6.64}
$$

- For $\mathfrak{B}^{\mathrm{phys}} = \mathfrak{B}_{\mathrm{Neu}(\mathbb{Z}_2),\omega}$ for $\omega \in \{0, 1\}$, the $\mathbb{Z}_4^{(0)}$ symmetry is broken to a $\mathbb{Z}_2^{(0)}$ subgroup,

$$
\begin{array}{ccc}
\mathfrak{B}_{\mathrm{Dir}} \quad \mathfrak{B}_{\mathrm{Neu}(\mathbb{Z}_2),\omega} & & \mathfrak{T}^{\mathcal{S}} \\
\underset{Q_1^{e^i}}{\boxed{\phantom{QQQ}}} & = & \big|\; D_0^{e^i}
\end{array}
\tag{6.65}
$$

with $i \in \{0, 2\}$. The resulting 3d TFT is

$$
\mathfrak{T} = \mathrm{Triv} \oplus \mathrm{Triv},
\tag{6.66}
$$

The broken $\mathbb{Z}_2$ subgroup acts to exchange the two trivial vacua.

- For $\mathfrak{B}^{\mathrm{phys}} = \mathfrak{B}_{\mathrm{Neu},\omega}$ for $\omega \in \{0, 1, 2, 3\}$, all of $\mathbb{Z}_4^{(0)}$ symmetry is unbroken and the SymTFT picture is

$$
\begin{array}{ccc}
\mathfrak{B}_{\mathrm{Dir}} \quad \mathfrak{B}_{\mathrm{Neu},\omega} & & \mathfrak{T}^{\mathcal{S}} \\
\boxed{\phantom{QQQ}} & = & \big|
\end{array}
\tag{6.67}
$$

The resulting 3d TFT is

$$
\mathfrak{T} = \mathrm{Triv}.
\tag{6.68}
$$

**Gapped Phases for $2\mathsf{Vec}^{\omega}_{\mathbb{Z}_2^{(0)} \times \mathbb{Z}_2^{(1)}}$.** The 3d phases obtained by choosing $\mathfrak{B}^{\mathrm{sym}} = \mathfrak{B}_{\mathrm{Neu}(\mathbb{Z}_2),\omega}$ have symmetry 0-form $\mathbb{Z}_2^{(0)}$ and 1-form $\mathbb{Z}_2^{(1)}$, with a mixed anomaly $\omega$. The minimal gapped phases are:

- For $\mathfrak{B}^{\mathrm{phys}} = \mathfrak{B}_{\mathrm{Dir}}$, the SymTFT picture is

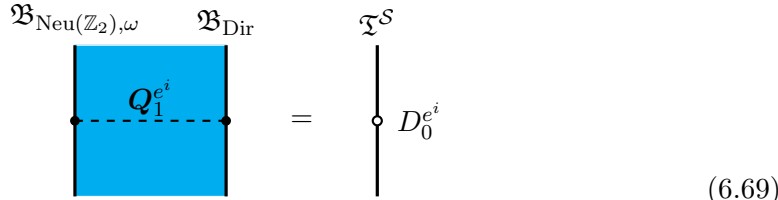

$$(6.69)$$

  with $i \in \{0,2\}$. The $\mathbb{Z}_2^{(0)}$ is spontaneously broken giving rise to two trivial vacua

$$\mathfrak{T} = \mathrm{Triv} \oplus \mathrm{Triv}. \tag{6.70}$$

  Inside each trivial vacuum the 1-form symmetry is generated by the identity line $D_1^{\mathrm{id}}$.

- For $\mathfrak{B}^{\mathrm{phys}} = \mathfrak{B}_{\mathrm{Neu}(\mathbb{Z}_2),omega)}$ for $\omega \in \{0,1\}$, the SymTFT picture is

$$(6.71)$$

  with $i,j \in \{0,2\}$. The $\mathbb{Z}_2^{(0)}$ is spontaneously broken giving rise to two vacua carrying a topological order

$$\mathfrak{T} = \mathfrak{Z}(\mathsf{Vec}^{(\omega')^{-1}}_{\mathbb{Z}_2} \boxtimes_{\mathbb{Z}_2} \mathsf{Vec}^{\omega}_{\mathbb{Z}_2}) \oplus \mathfrak{Z}(\mathsf{Vec}^{(\omega')^{-1}}_{\mathbb{Z}_2} \boxtimes_{\mathbb{Z}_2} \mathsf{Vec}^{\omega}_{\mathbb{Z}_2}). \tag{6.72}$$

  Inside each $\mathfrak{Z}(\mathsf{Vec}^{(\omega')^{-1}}_{\mathbb{Z}_2} \boxtimes_{\mathbb{Z}_2} \mathsf{Vec}^{\omega}_{\mathbb{Z}_2})$ vacuum the 1-form symmetry is generated by the line $D_1^-$ that projects to $D_1^{\mathrm{id}} \in \mathsf{Vec}^{(\omega')^{-1}}_{\mathbb{Z}_2} \boxtimes_{\mathbb{Z}_2} \mathsf{Vec}^{\omega}_{\mathbb{Z}_2}$, half-braids trivially with trivial grade of $\mathsf{Vec}^{(\omega')^{-1}}_{\mathbb{Z}_2} \boxtimes_{\mathbb{Z}_2} \mathsf{Vec}^{\omega}_{\mathbb{Z}_2}$ and half-braids by a non-trivial sign with the non-trivial grade of $\mathsf{Vec}^{(\omega')^{-1}}_{\mathbb{Z}_2} \boxtimes_{\mathbb{Z}_2} \mathsf{Vec}^{\omega}_{\mathbb{Z}_2}$. The $\mathbb{Z}_2^{(0)}$ symmetry fractionalizes on $D_1^-$, which is what captures the 't Hooft anomaly.

- For $\mathfrak{B}^{\mathrm{phys}} = \mathfrak{B}_{\mathrm{Neu},\omega}$ for $\omega \in \{0,1,2,3\}$, the $\mathbb{Z}_2^{(0)}$ symmetry is unbroken and the SymTFT picture is

$$(6.73)$$

with $j \in \{0, 2\}$. The resulting 3d TFT is

$$\mathfrak{T} = \mathfrak{Z}\left(\mathsf{Vec}_{\mathbb{Z}_2}^{(\omega')^{-1}} \boxtimes_{\mathbb{Z}_2} \mathsf{Vec}_{\mathbb{Z}_4}^{\omega}\right). \tag{6.74}$$

The $\mathbb{Z}_2^{(0)}$ symmetry is realized according to the extension of $\mathsf{Vec}_{\mathbb{Z}_2}^{(\omega')^{-1}}$ to $\mathsf{Vec}_{\mathbb{Z}_4}^{\omega}$, where $\mathsf{Vec}_{\mathbb{Z}_4}^{\omega}$ is now viewed as a $\mathbb{Z}_2$ graded fusion category whose trivial grade is $\mathsf{Vec}_{\mathbb{Z}_2}^{(\omega')^{-1}}$. The $\mathbb{Z}_2^{(1)}$ symmetry is realized by the line operator $D_1^-$ associated to the fusion category $\mathsf{Vec}_{\mathbb{Z}_2}^{(\omega')^{-1}}$. The $\mathbb{Z}_2^0$ symmetry is fractionalized on $D_1^-$, which realizes the 't Hooft anomaly.

**Gapped Phases for** $2\mathsf{Rep}(\mathbb{Z}_4)$. The $\mathbb{Z}_4$ 1-form symmetric 3d phases obtained by choosing $\mathfrak{B}^{\mathrm{sym}} = \mathfrak{B}_{\mathrm{Neu},\omega}$ are

- For $\mathfrak{B}^{\mathrm{phys}} = \mathfrak{B}_{\mathrm{Dir}}$, the SymTFT is simply

$$\tag{6.75}$$

and the $\mathbb{Z}_4^{(1)}$ is realized trivially in a trivial 3d TFT

$$\mathfrak{T} = \mathrm{Triv}. \tag{6.76}$$

- For $\mathfrak{B}^{\mathrm{phys}} = \mathfrak{B}_{\mathrm{Neu}(\mathbb{Z}_2),\omega}$ for $\omega \in \{0, 1\}$, the SymTFT picture is

$$\tag{6.77}$$

with $j \in \{0, 2\}$. The $\mathbb{Z}_4^{(1)}$ is spontaneously broken to $\mathbb{Z}_2^{(1)}$. The underlying 3d TFT is

$$\mathfrak{T} = \mathfrak{Z}(\mathsf{Vec}_{\mathbb{Z}_2}^{\alpha'} \boxtimes_{\mathbb{Z}_2} \mathsf{Vec}_{\mathbb{Z}_2}^{\omega}), \tag{6.78}$$

where $\alpha'$ is pullback to $\mathbb{Z}_2$ subgroup of the $\mathbb{Z}_4$ class $(\omega')^{-1}$. The 1-form symmetry is generated by the line $D_1^-$ described above in the case of $\mathfrak{B}^{\mathrm{sym}} = \mathfrak{B}_{\mathrm{Neu}(\mathbb{Z}_2),\omega}$.

- For $\mathfrak{B}^{\mathrm{phys}} = \mathfrak{B}_{\mathrm{Neu},\omega}$ for $\omega \in \{0, 1, 2, 3\}$,

$$\tag{6.79}$$

with $j \in \{0, 1, 2, 3\}$. The resulting 3d TFT is

$$\mathfrak{T} = \mathfrak{Z}(\mathsf{Vec}_{\mathbb{Z}_4}^{(\omega')^{-1}} \boxtimes_{\mathbb{Z}_4} \mathsf{Vec}_{\mathbb{Z}_4}^{\omega}). \tag{6.80}$$

The $\mathbb{Z}_4^{(1)}$ symmetry is generated by the line operator $D_1^{m^j}$ which projects to trivial line in $\mathsf{Vec}_{\mathbb{Z}_4}^{(\omega')^{-1}} \boxtimes_{\mathbb{Z}_4} \mathsf{Vec}_{\mathbb{Z}_4}^{\omega}$ and whose half-braiding with every object in $j$-th grade is $i^j$.

## 6.3 Non-Minimal Gapped Boundary Conditions

We now describe the structure of non-minimal boundary conditions for $(3+1)$d DW theory for a general Abelian group $\mathbb{A}$.

**Non-minimal Dirichlet Boundaries:** Let us again first start with Dirichlet type boundary conditions. The are classified by MTCs. Given an MTC $\mathcal{M}$ which describes the topological line defects of a TFT $\mathfrak{T}$, one can trivially stack the Dirichlet BC to obtain a new topological boundary condition

$$\mathfrak{B}_{\text{Dir}}^{\mathfrak{T}} = \mathfrak{B}_{\text{Dir}} \boxtimes \mathfrak{T}. \tag{6.81}$$

The resulting symmetry on this boundary is given by the genuine topological defects on the boundary which form

$$\mathcal{S}(\mathfrak{B}_{\text{Dir}}^{\mathfrak{T}}) = 2\mathsf{Vec}_{\mathbb{A}} \boxtimes \Sigma\mathcal{M}, \tag{6.82}$$

where $\Sigma\mathcal{M}$ is the fusion 2-category obtained from $\mathcal{M}$ by including all possible condensation surface defects, i.e. the Karoubi 2-completion of the MTC $\mathcal{M}$ [35, 71].

**Non-minimal Neumann Boundaries:** Now let us describe the non-minimal generalizations of $\mathfrak{B}_{\text{Neu},\omega}$, which we denote by $\mathfrak{B}_{\text{Neu},\omega}^{\mathfrak{T}_{\mathbb{A}}}$. Here $\mathfrak{T}_{\mathbb{A}}$ is some $\mathbb{A}$ symmetric TFT whose line defects form the MTC $\mathcal{M}$. An $\mathbb{A}$ symmetry can be implemented on $\mathfrak{T}$ by a collection of invertible surface defects in $\Sigma\mathcal{M}$ that satisfy $\mathbb{A}$ fusion rules. Notably, these surfaces don't quite form $2\mathsf{Vec}_{\mathbb{A}}$ as all such surface defects are constructible as condensations of lines in $\mathcal{M}$ and can therefore end along non-genuine lines [72]. In order to characterize the $\mathbb{A}$ symmetry of $\mathcal{M}$, we consider the lines in $\mathcal{M}$ along with the non-genuine lines at the ends of the condensations defects implementing $a \in \mathbb{A}$. Together these form

$$\mathcal{M}_{\mathbb{A}}^{\times} = \bigoplus_{a \in \mathbb{A}} \mathcal{M}_a, \tag{6.83}$$

the $\mathbb{A}$ crossed braided extension of $\mathcal{M}$. Here $\mathcal{M} \equiv \mathcal{M}_{\text{id}} \in \mathcal{M}_{\mathbb{A}}^{\times}$. The topological lines on $\mathfrak{B}_{\text{Neu},\omega}^{\mathfrak{T}_{\mathbb{A}}}$ can then be obtained by gauging $\mathbb{A}$ in $\mathcal{M}_{\mathbb{A}}^{\times}$ which furnishes a new MTC via the $\mathbb{A}$-equivariantization of $\mathcal{M}_{\mathbb{A}}^{\times}$ [3, 73]

$$\widetilde{\mathcal{M}} = (\mathcal{M}_{\mathbb{A}}^{\times})^{\mathbb{A}}. \tag{6.84}$$

Some of the lines in $\widetilde{\mathcal{M}}$ are non-genuine. In particular, $\widetilde{\mathcal{M}}$ is an $\mathbb{A}$-graded braided fusion category. The boundary projections of $\boldsymbol{Q}_1^{\hat{a}}$ play an important role. They generate a $2\mathsf{Rep}_\mathbb{A}$ symmetry on $\mathfrak{B}_{\mathrm{Neu},\omega}^{\mathfrak{T}_\mathbb{A}}$. Let us denote the boundary projections of $\boldsymbol{Q}_1^{\hat{a}}$ by $D_1^{\hat{a}}$. Then $\widetilde{\mathcal{M}}$ has the $\mathbb{A}$-graded structure

$$\widetilde{\mathcal{M}} = \bigoplus_{a\in\mathbb{A}} \widetilde{\mathcal{M}}_a \,, \tag{6.85}$$

such that any line in $\widetilde{\mathcal{M}}_a$ with $a \neq \mathrm{id}$ is non-genuine and in particular is attached to the bulk surface $\boldsymbol{Q}_2^a$. Since the $\mathsf{Rep}_\mathbb{A}$ symmetry on $\mathfrak{B}_{\mathrm{Neu},\omega}^{\mathfrak{T}_\mathbb{A}}$ is generated by the boundary projection of $\boldsymbol{Q}_1^{\hat{a}}$ the braiding of any line $\widetilde{\mathcal{L}}_a^x \in \widetilde{M}_a$ with $D_1^{\hat{a}}$ is constrained by the bulk braiding of $\boldsymbol{Q}_1^{\hat{a}}$ and $\boldsymbol{Q}_2^a$ such that

$$B(\widetilde{\mathcal{L}}_a^x, D_1^{\hat{a}}) = B(\boldsymbol{Q}_2^a, \boldsymbol{Q}_1^{\hat{a}}) = \hat{a} \,. \tag{6.86}$$

The Müger center [74] (or sub-category of transparent lines) of $\widetilde{\mathcal{M}}_{\mathrm{id}}$ is precisely the dual symmetry obtained upon gauging $\mathbb{A}$ in $\mathcal{M}_\mathbb{A}^\times$.

$$\mathcal{Z}_2(\mathcal{M}_\mathbb{A}^\times) = \mathsf{Rep}_\mathbb{A} = \{ D_1^{\hat{a}} \mid \hat{a} \in \mathsf{Rep}(\mathbb{A}) \ \} \,. \tag{6.87}$$

Finally, since $\mathfrak{B}_{\mathrm{Neu},\omega}^{\mathfrak{T}_\mathbb{A}}$ is obtained by gauging the Dirichlet boundary condition with a choice of discrete torsion $\omega \in H^3(\mathbb{A}, U(1))$, the non-genuine lines after gauging have a non-trivial associator controlled by $\omega$. The associator of any three lines $\widetilde{\mathcal{L}}_{a_1}^x \in \widetilde{M}_{a_1}$, $\widetilde{\mathcal{L}}_{a_2}^y \in \widetilde{M}_{a_2}$ and $\widetilde{\mathcal{L}}_{a_3}^z \in \widetilde{M}_{a_3}$ is

$$F(\widetilde{\mathcal{L}}_{a_1}^x, \widetilde{\mathcal{L}}_{a_2}^y, \widetilde{\mathcal{L}}_{a_3}^z) = \omega(a_1, a_2, a_3) \,. \tag{6.88}$$

Again the symmetry 2-category associated to the boundary $\mathfrak{B}_{\mathrm{Neu},\omega}^{\mathfrak{T}_\mathbb{A}}$ is given by the Karoubi completion of the subcategory of genuine line defects

$$\mathcal{S}(\mathfrak{B}_{\mathrm{Neu},\omega}^{\mathfrak{T}_\mathbb{A}}) = \Sigma\widetilde{\mathcal{M}}_{\mathrm{id}} \,. \tag{6.89}$$

**Non-Minimal Partial Neumann Boundary Conditions:**  We now move onto the non-minimal generalizations of the partial Neumann boundary condition $\mathfrak{B}_{\mathrm{Neu}(\mathbb{B}),\omega}$ which we denote as $\mathfrak{B}_{\mathrm{Neu}(\mathbb{B}),\omega}^{\mathfrak{T}_\mathbb{B}}$. Again the starting point for classifying such boundary conditions would be the Dirichlet boundary condition stacked with a TFT $\mathfrak{T}_\mathbb{B}$ which is symmetric under $\mathbb{B} \subset \mathbb{A}$. Let the lines on $\mathfrak{T}$ form the MTC $\mathcal{M}$. The discussion about the $\mathbb{B}$ symmetry implementation on $\mathcal{M}$ is identical to that described above. We first obtain the $\mathbb{B}$-crossed braided extension of $\mathcal{M}$ which is $\mathbb{B}$ graded

$$\mathcal{M}_\mathbb{B}^\times = \bigoplus_{b\in\mathbb{B}} \mathcal{M}_b \,, \tag{6.90}$$

such that $\mathcal{M}_b$ comprises of the lines that live at the end of the condensation defects implementing $b \in \mathbb{B}$ on $\mathfrak{T}$. Then the $\mathbb{B}$-equivariantization of $\mathcal{M}_\mathbb{B}^\times$ produces an MTC $\widetilde{\mathcal{M}}$. The boundary

projections of $\boldsymbol{Q}_1^{\hat{b}}$ generate a non-anomalous 1-form $\mathsf{Rep}(\mathbb{B})$ symmetry on $\mathfrak{B}_{\mathrm{Neu}(\mathbb{B}),\omega}^{\mathfrak{T}_\mathbb{B}}$. We denote the corresponding symmetry generators by $D_1^{\hat{b}}$. The category of lines $\widetilde{\mathcal{M}}$ on $\mathfrak{B}_{\mathrm{Neu}(\mathbb{B}),\omega}^{\mathfrak{T}_\mathbb{B}}$ is $\mathbb{B}$ graded

$$\widetilde{\mathcal{M}} = \bigoplus_{b \in \mathbb{B}} \widetilde{\mathcal{M}}_b \,, \tag{6.91}$$

such that the lines in $\widetilde{\mathcal{M}}_b$ are non-genuine lines that live at the ends of $\boldsymbol{Q}_2^b$. Consistency requires that any $\mathcal{L}_b^x \in \widetilde{\mathcal{M}}_b$ braids with $D_1^{\hat{b}}$ as

$$B(\mathcal{L}_b^x, D_1^{\hat{b}}) = B(\boldsymbol{Q}_2^b, \boldsymbol{Q}_1^{\hat{b}}) = \hat{b}(b) \,. \tag{6.92}$$

The Müger centre of $\widetilde{\mathcal{M}}_{\mathrm{id}}$ is $\mathsf{Rep}(\mathbb{B})$ and generates a non-anomalous 1-form symmetry on $\mathfrak{B}_{\mathrm{Neu}(\mathbb{B}),\omega}^{\mathfrak{T}_\mathbb{B}}$

$$\mathcal{Z}_2(\widetilde{\mathcal{M}}_{\mathrm{id}}) = \mathsf{Rep}(\mathbb{B}) = \{D_1^{\hat{b}} \mid \hat{b} \in \mathsf{Rep}(\mathbb{B})\} \,. \tag{6.93}$$

Furthermore, since $\mathfrak{B}_{\mathrm{Neu}(\mathbb{B}),\omega}^{\mathfrak{T}_\mathbb{B}}$ is obtained by gauging the Dirichlet boundary condition with a choice of discrete torsion $\omega \in H^3(\mathbb{B}, U(1))$, the non-genuine lines after gauging have a non-trivial associator controlled by $\omega$. The associator of any three lines $\widetilde{\mathcal{L}}_{b_1}^x \in \widetilde{M}_{b_1}$, $\widetilde{\mathcal{L}}_{b_2}^y \in \widetilde{M}_{b_2}$ and $\widetilde{\mathcal{L}}_{b_3}^z \in \widetilde{M}_{b_3}$ is

$$F(\widetilde{\mathcal{L}}_{b_1}^x, \widetilde{\mathcal{L}}_{b_2}^y, \widetilde{\mathcal{L}}_{b_3}^z) = \omega(b_1, b_2, b_3) \,. \tag{6.94}$$

Finally the surfaces $\boldsymbol{Q}_2^a$ with $a \in \mathbb{A}/\mathbb{B}$ do not end on $\mathfrak{B}_{\mathrm{Neu}(\mathbb{B}),\omega}^{\mathfrak{T}_\mathbb{B}}$ and therefore project as $D_2^a$ which generates a $2\mathsf{Vec}_{\mathbb{A}/\mathbb{B}}$ symmetry. There is a mixed-anomaly between the 0-form symmetry $\mathbb{A}/\mathbb{B}$ and the 1-form symmetry $\mathsf{Rep}_\mathbb{B}$ given by the anomaly action (6.21). Physically this anomaly encodes the fact that the fusion product $D_2^{a_1} \otimes D_2^{a_2}$ with $a_1, a_2 \in \mathbb{A}/\mathbb{B}$ is isomorphic to $D_2^{a_1 a_2}$ such that the isomorphism is implemented by the line $\widehat{D}_1^{\epsilon(a_1,a_2)}$ which braids non-trivially with the 1-form symmetry generator $D_1^{\hat{b}}$ with the braiding phase

$$B(\widehat{D}_1^{\epsilon(a_1,a_2)}, D_1^{\hat{b}}) = \hat{b}(\epsilon(a_1, a_2)) \,. \tag{6.95}$$

The symmetry category associated with $\mathfrak{B}_{\mathrm{Neu}(\mathbb{B}),\omega}^{\mathfrak{T}_\mathbb{B}}$ is

$$\mathcal{S}(\mathfrak{B}_{\mathrm{Neu}(\mathbb{B}),\omega}^{\mathfrak{T}_\mathbb{B}}) = \Sigma\mathcal{M}_{\mathrm{id}} \boxtimes 2\mathsf{Vec}_{\mathbb{A}/\mathbb{B}} + \text{anomaly}. \tag{6.96}$$

### 6.3.1 Example: $\mathbb{A} = \mathbb{Z}_4$

Applied to $\mathbb{A} = \mathbb{Z}_4$ these non-minimal boundary conditions are as follows:

**Dirichlet Boundaries.** These are again classified by MTCs

$$\mathfrak{B}_{\mathrm{Dir}}^{\mathcal{M}} := \mathfrak{B}_{\mathrm{Dir}} \boxtimes \mathfrak{T}_{\mathcal{M}} \,. \tag{6.97}$$

The symmetry is

$$\mathcal{S}_\mathcal{M} = 2\mathsf{Vec}_{\mathbb{Z}_4} \boxtimes \Sigma\mathcal{M} \,. \tag{6.98}$$

The boundaries admitting topological interfaces to $\mathfrak{B}_{\mathrm{Dir}}$ are classified by fusion categories.

**Neumann Boundaries.** These are classified by MTCs $\mathcal{M}$ with a choice of $\mathsf{Rep}(\mathbb{Z}_4)$ sub-category. Such an MTC is $\mathbb{Z}_4$ graded. The symmetry is

$$\mathcal{S}_\mathcal{M} = \Sigma\mathcal{M}^{(0)} \,, \tag{6.99}$$

where $\mathcal{M}^{(0)}$ is the braided fusion category formed by lines lying in trivial grade of $\mathcal{M}$. We denote such boundaries by $\mathfrak{B}_{\mathrm{Neu}}^\mathcal{M}$. The Müger center of $\mathcal{M}^{(0)}$ is $\mathsf{Rep}(\mathbb{Z}_4)$. The boundaries admitting topological interfaces to $\mathfrak{B}^e$ are classified by $\mathbb{Z}_4$-graded fusion categories.

**Neu($\mathbb{Z}_2$) Boundaries.** These are constructed by first stacking $\mathfrak{B}_{\mathrm{Dir}}$ with a $\mathbb{Z}_2$-symmetric 3d TFT and then gauging the diagonal $\mathbb{Z}_2$ symmetry. As such they are classified by MTCs $\mathcal{M}$ with a choice of $\mathsf{Rep}(\mathbb{Z}_2)$ sub-category. The symmetry contains $\Sigma\mathcal{M}^{(0)}$ and $2\mathsf{Vec}_{\mathbb{Z}_2}$ with the generator $D_2^m$ of $2\mathsf{Vec}_{\mathbb{Z}_2}$ fractionalized on the line $D_1^{e^2}$ generating $\mathsf{Rep}(\mathbb{Z}_2)$. We denote such boundaries by $\mathfrak{B}_{\mathrm{Neu}(\mathbb{Z}_2)}^\mathcal{M}$.

## 6.4  Non-Minimal Phases

Now we describe the gapped phases obtained by picking non-minimal gapped boundaries as the physical boundary and symmetry boundaries. For simplicity, we always pick boundaries with the trivial discrete torsion class. However boundaries with non-trivial discrete torsion can be readily included using the methods described previously.

**Non-Minimal Phases for $\mathfrak{B}^{\mathbf{sym}} = \mathfrak{B}_{\mathbf{Dir}}^\mathfrak{T}$:** Here the symmetry structure is

$$\mathcal{S}(\mathfrak{B}_{\mathrm{Dir}}^\mathfrak{T}) = 2\mathsf{Vec}_\mathbb{A} \boxtimes \Sigma\mathcal{M} \,. \tag{6.100}$$

For the different choices of $\mathfrak{B}^{\mathrm{phys}}$ we get:

- $\mathfrak{B}^{\mathrm{phys}} = \mathbb{B}\mathrm{Dir}^{\mathfrak{T}'}$: Since the bulk lines $\boldsymbol{Q}_1^{\hat{a}}$ can end on both the symmetry and physical boundaries, we get $|\mathbb{A}|$ local operators and hence $|\mathbb{A}|$ identical vacua. Each of these vacua realize the $\mathfrak{T} \otimes \mathfrak{T}'$ TFT.

- $\mathfrak{B}^{\mathrm{phys}} = \mathfrak{B}_{\mathrm{Neu}}^{\mathfrak{T}_\mathbb{A}}$: Let the underlying MTC of the theory $\mathfrak{T}'$ be denoted $\mathcal{M}'$ and its $\mathbb{A}$ equivariantization $\widetilde{\mathcal{M}'}$. With the Neumann boundary condition, we obtain order parameters from the bulk surfaces $\boldsymbol{Q}_2^a$. After compactifying the SymTFT these become

non-genuine lines in the grade $\widetilde{\mathcal{M}}'_a \in \widetilde{\mathcal{M}}$. Additionally, there are genuine lines in the MTC $\mathcal{M}$ contributed from the symmetry structure as well as additional genuine lines

$$\widetilde{\mathcal{M}}'_{\mathrm{id}}, \tag{6.101}$$

from the physical boundary. To summarize, we get a single vacuum theory whose underlying MTC is

$$\mathcal{M} \boxtimes \widetilde{\mathcal{M}}'_{\mathrm{id}}. \tag{6.102}$$

The $\mathbb{A}$ action is implemented such that $\mathbb{A}$ crossed braided extension coincides with $\mathcal{M} \boxtimes \widetilde{\mathcal{M}}'$.

- $\mathfrak{B}^{\mathrm{phys}} = \mathfrak{B}^{\mathfrak{T}_{\mathbb{A}}}_{\mathrm{Neu}(\mathbb{B})}$: In this phase the lines $Q_1^{\hat{a}}$ for $\hat{a} \in \widehat{\mathbb{A}/\mathbb{B}}$ can end on both boundaries and therefore become local operators after compactifying the SymTFT. These local operators are order parameters for symmetry breaking from $\mathbb{A} \to \mathbb{B}$. There are $|\mathbb{A}/\mathbb{B}|$ vacua, each of which realize a TFT whose underlying MTC is

$$\mathcal{M} \boxtimes \widetilde{\mathcal{M}}'_{\mathrm{id}}. \tag{6.103}$$

The $\mathbb{B}$ action is implemented such that $\mathbb{B}$ crossed braided extension of the MTC coincides with $\mathcal{M} \boxtimes \widetilde{\mathcal{M}}'$ restricted to $\mathbb{B}$.

**Non-Minimal Phases for $\mathfrak{B}^{\mathrm{sym}} = \mathfrak{B}^{\mathfrak{T}_{\mathbb{A}}}_{\mathbf{Neu}}$:** The symmetry 2-category for this choice of boundary is

$$\mathcal{S}(\mathfrak{B}^{\mathfrak{T}_{\mathbb{A}}}_{\mathrm{Neu}}) = \Sigma \widetilde{\mathcal{M}}_{\mathrm{id}}. \tag{6.104}$$

For the different choices of $\mathfrak{B}^{\mathrm{phys}}$ we get:

- $\mathfrak{B}^{\mathrm{phys}} = \mathbb{B}\mathrm{Dir}^{\mathfrak{T}'}$: In this phase, all the bulk lines $\boldsymbol{Q}_1^{\hat{a}}$ can end on both the boundaries. As a result, the non-anomalous 1-form symmetry $\mathsf{Rep}_{\mathbb{A}}$ is trivialized. The IR realizes a TFT whose underlying TFT is $\widetilde{\mathcal{M}}_{\mathrm{id}}^{(0)}$ which is obtained from $\widetilde{\mathcal{M}}_{\mathrm{id}}$ by forgetting or quotienting out the Müger center $\mathsf{Rep}_{\mathbb{A}}$ lines. Additionally, there is MTC $\mathcal{M}'$ which underlies the TFT $\mathfrak{T}'$ which is stacked onto the physical boundary. Therefore we obtain a gapped phase described by the MTC

$$\widetilde{\mathcal{M}}_{\mathrm{id}}^{(0)} \boxtimes \mathcal{M}'. \tag{6.105}$$

- $\mathfrak{B}^{\mathrm{phys}} = \mathbb{B}\mathrm{Neu}^{\mathfrak{T}_{\mathbb{A}}}$: In this SymTFT sandwich, the surface $\boldsymbol{Q}_2^a$ for all $a \in \mathbb{A}$ can end on both boundaries. Compactifying such makes the lines in the grade

$$\widetilde{\mathcal{M}}_a \otimes \widetilde{\mathcal{M}}'_a, \tag{6.106}$$

genuine lines. These lines braid non-trivially with the lines that generate the $\mathsf{Rep}_{\mathbb{A}}$ symmetry in the Müger center of $\widetilde{\mathcal{M}}$. The resulting gapped phase forms an MTC

$$\bigoplus_{a \in \mathbb{A}} \left[ \widetilde{\mathcal{M}}_a \otimes \widetilde{\mathcal{M}}'_a \right] . \tag{6.107}$$

This is a non-minimal 1-form symmetry breaking phase.

- $\mathfrak{B}^{\text{phys}} = \mathfrak{B}_{\text{Neu}}^{\mathfrak{T}'_{\mathbb{B}}}$: Since $\boldsymbol{Q}_1^{\hat{a}}$ for $\hat{a}$ in $\widehat{\mathbb{A}/\mathbb{B}}$ can end on the physical boundary, it plays the role of an order parameter. Stretched between the two boundaries, the line $\boldsymbol{Q}_1^{\hat{a}}$ provides an end for the symmetry operator $D_1^{\hat{a}}$ in the Müger center of $\widetilde{\mathcal{M}}_{\text{id}}$. The other family of order parameters correspond to the surfaces $\boldsymbol{Q}_2^b$ that can end on both boundaries. On the symmetry boundary this surface ends on

$$\widetilde{\mathcal{M}}_b \in \widetilde{\mathcal{M}} , \tag{6.108}$$

while on the physical boundary it ends on

$$\widetilde{\mathcal{M}}'_b \in \widetilde{\mathcal{M}}' . \tag{6.109}$$

Therefore the compactification of the $\boldsymbol{Q}_2^b$ provides in the IR TFT a set of genuine lines which corresponds to

$$\widetilde{\mathcal{M}}_b \otimes \widetilde{\mathcal{M}}'_b . \tag{6.110}$$

These lines braid non-trivially with the surviving $\mathsf{Rep}_{\mathbb{B}}$ lines. In summary, the MTC underlying the IR TFT is given by

$$\bigoplus_{b \in \mathbb{B}} \left[ \widetilde{\mathcal{M}}_b \otimes \widetilde{\mathcal{M}}'_b \right] . \tag{6.111}$$

**Non-Minimal Phases for $\mathfrak{B}^{\text{sym}} = \mathfrak{B}_{\text{Neu}(\mathbb{B})}^{\mathfrak{T}_{\mathbb{B}}}$:** The symmetry 2-category for this choice of boundary is given in (6.96). For the different choices of $\mathfrak{B}^{\text{phys}}$ we get:

- $\mathfrak{B}^{\text{phys}} = \mathbb{B}\text{Dir}^{\mathfrak{T}'}$: All the bulk lines can end on this choice of physical boundary. These play the role of order parameters. The lines

$$\boldsymbol{Q}_1^{\hat{a}} , \qquad \hat{a} \in \widehat{\mathbb{A}} / \widehat{\mathbb{A}/\mathbb{B}} \simeq \widehat{\mathbb{B}} , \tag{6.112}$$

provide ends $\mathsf{Rep}_{\mathbb{B}}$ 1-form symmetry generators upon compactification. Therefore the $\mathsf{Rep}_{\mathbb{B}}$ 1-form symmetry generators are all isomorphic to identity line and are therefore trivial in this phase. The MTC realized in the IR is given by the quotient

$$\widetilde{\mathcal{M}}_{\text{id}}^{(0)} \cong \widetilde{\mathcal{M}}_{\text{id}} / \mathcal{Z}_2 \left[ \widetilde{\mathcal{M}}_{\text{id}} \right] . \tag{6.113}$$

Meanwhile the lines

$$\boldsymbol{Q}_1^{\hat{a}}, \qquad \hat{a} \in \widehat{\mathbb{A}/\mathbb{B}}, \tag{6.114}$$

can end on both boundaries and therefore become symmetry breaking order parameters for the $\mathbb{A}/\mathbb{B}$ symmetry. To summarize, in this phase, the 0 form symmetry is spontaneously broken, the 1-form symmetry is spontaneously preserved. There are $|\mathbb{A}/\mathbb{B}|$ vacua, each of which realize a TFT whose underlying MTC is $\widetilde{\mathcal{M}}_{\mathrm{id}}^{(0)}$.

- $\mathfrak{B}^{\mathrm{phys}} = \mathfrak{B}_{\mathrm{Neu}}^{\mathfrak{T}_{\mathbb{A}}'}$: In this SymTFT sandwich, all the bulk surfaces can end on the physical boundary and therefore play the role of order parameters. The bulk surfaces $\boldsymbol{Q}_2^b$ with $b \in \mathbb{B}$ end on the lines in the grade $\widetilde{\mathcal{M}}_b \in \widetilde{\mathcal{M}}$ on the symmetry boundary and likewise on lines in the grade $\widetilde{\mathcal{M}}_b' \in \widetilde{\mathcal{M}}'$ on the physical boundary. After compactification of the SymTFT, this provides genuine lines

$$\bigoplus_{b \in \mathbb{B}} \left[ \widetilde{\mathcal{M}}_b \otimes \widetilde{\mathcal{M}}' \right], \tag{6.115}$$

such that grade lines in the grade $b \in \mathbb{B}$ have a braiding phase $\hat{b}(b)$ with the $\mathsf{Rep}_{\mathbb{B}}$ line $D_1^{\hat{b}}$. The remaining bulk surfaces

$$\boldsymbol{Q}_2^a, \qquad a \in \mathbb{A}/\mathbb{B}, \tag{6.116}$$

after SymTFT compactification become twisted sector lines of the unbroken $\mathbb{A}/\mathbb{B}$ 0-form symmetry.

- $\mathfrak{B}^{\mathrm{phys}} = \mathfrak{B}_{\mathrm{Neu}(\mathbb{D})}^{\mathfrak{T}_{\mathbb{D}}'}$: In this case, the surfaces $\boldsymbol{Q}_2^a$ with $a \in \mathbb{D} \cap \mathbb{B}$ can end on both boundaries. This provides the following MTC of genuine lines in the IR TFT corresponding to this gapped phase

$$\bigoplus_{a \in \mathbb{B} \cap \mathbb{D}} \left[ \widetilde{\mathcal{M}}_a \otimes \widetilde{\mathcal{M}}_a' \right]. \tag{6.117}$$

Additionally the bulk lines

$$Q_1^{\hat{a}}, \quad \hat{a} \in \widehat{\mathbb{A}/\mathbb{B}} \cap \widehat{\mathbb{A}/\mathbb{D}}, \tag{6.118}$$

can end on both boundaries and therefore become local operators after compactifying the SymTFT. The number of vacua is equal to the number of linearly independent topological operators which is

$$\left| \widehat{\mathbb{A}/\mathbb{B}} \cap \widehat{\mathbb{A}/\mathbb{D}} \right|. \tag{6.119}$$

The 0-form symmetry preserved in this phase is $\mathbb{D}/(\mathbb{D} \cap \mathbb{B})$. The non-genuine lines at the ends of the 0-form symmetry defect $D_2^a$ lies in $\widetilde{\mathcal{M}}_a$ with $a \in \mathbb{D}/(\mathbb{D} \cap \mathbb{B})$.

### 6.4.1  Example: $\mathbb{A} = \mathbb{Z}_4$

**Dir-Dir-Sandwich.**   We take

$$\mathfrak{B}^{\mathrm{sym}} = \mathfrak{B}_{\mathrm{Dir}}, \qquad \mathfrak{B}^{\mathrm{phys}} = \mathfrak{B}_{\mathrm{Dir}}^{\mathcal{M}} \tag{6.120}$$

The resulting 3d TFT is

$$\mathfrak{T}_{\mathcal{M}} \oplus \mathfrak{T}_{\mathcal{M}} \oplus \mathfrak{T}_{\mathcal{M}} \oplus \mathfrak{T}_{\mathcal{M}} \tag{6.121}$$

with the $\mathbb{Z}_4^{(0)}$ symmetry cyclically permuting the four $\mathfrak{T}_{\mathcal{M}}$ vacua.

**Dir-Neu-Sandwich.**   We take

$$\mathfrak{B}^{\mathrm{sym}} = \mathfrak{B}_{\mathrm{Dir}} \quad, \qquad \mathfrak{B}^{\mathrm{phys}} = \mathfrak{B}_{\mathrm{Neu}}^{\mathcal{M}_0^{\mathbb{Z}_4}} \tag{6.122}$$

The resulting 3d TFT is

$$\mathfrak{T}_{\mathcal{M}_0} \tag{6.123}$$

with the $\mathbb{Z}_4^{(0)}$ symmetry described by the extension $\mathcal{M}_0^{\mathbb{Z}_4}$.

**Dir-Neu($\mathbb{Z}_2$)-Sandwich.**   We take

$$\mathfrak{B}^{\mathrm{sym}} = \mathfrak{B}_{\mathrm{Dir}} \quad, \qquad \mathfrak{B}^{\mathrm{phys}} = \mathfrak{B}_{\mathrm{Neu}(\mathbb{Z}_2)}^{\mathcal{M}_0^{\mathbb{Z}_2}} \tag{6.124}$$

The resulting 3d TFT is

$$\mathfrak{T}_{\mathcal{M}_0} \oplus \mathfrak{T}_{\mathcal{M}_0} \tag{6.125}$$

with the generator of $\mathbb{Z}_4^{(0)}$ symmetry exchanging the two $\mathfrak{T}_{\mathcal{M}_0}$ vacua, and the $\mathbb{Z}_2$ subgroup of $\mathbb{Z}_4$ realized within each vacuum according to the $\mathbb{Z}_2$ crossed braided extension $\mathcal{M}_0^{\mathbb{Z}_2}$ of $\mathcal{M}_0$.

**Neumann-Dirichlet-Sandwich.**   We take

$$\mathfrak{B}^{\mathrm{sym}} = \mathfrak{B}_{\mathrm{Neu}}^{\mathcal{M}_0^{\mathbb{Z}_4}} \quad, \qquad \mathfrak{B}^{\mathrm{phys}} = \mathfrak{B}_{\mathrm{Dir}} \tag{6.126}$$

The resulting 3d TFT is

$$\mathfrak{T}_{\mathcal{M}_0} \tag{6.127}$$

with the $\mathcal{M}^{(0)}$ symmetry realized via the forgetful functor

$$\mathcal{M}^{(0)} \to \mathcal{M}_0 \,, \tag{6.128}$$

that outputs the object of $\mathcal{M}_0$ underlying a $\mathbb{Z}_4$ equivariant object in $\mathcal{M}^{(0)}$.

**Neumann-Neumann-Sandwich.** We take

$$\mathfrak{B}^{\text{sym}} = \mathfrak{B}_{\text{Neu}}^{\mathcal{M}}, \qquad \mathfrak{B}^{\text{phys}} = \mathfrak{B}_{\text{Neu}}^{\mathcal{M}'}, \tag{6.129}$$

where $\mathcal{M}$ and $\mathcal{M}'$ are MTCs with chosen $\mathsf{Rep}(\mathbb{Z}_4)$ subcategories. The resulting 3d TFT is associated to the MTC $\widetilde{\mathcal{M}}$ obtained by de-equivariantizing the braided fusion category $\mathcal{M} \boxtimes_{\mathbb{Z}_4} \mathcal{M}'$ with respect to the diagonal $\mathsf{Rep}(\mathbb{Z}_4)$ of the two $\mathsf{Rep}(\mathbb{Z}_4)$s inside $\mathcal{M}$ and $\mathcal{M}'$, which we can express as

$$\widetilde{\mathcal{M}} = \frac{\mathcal{M} \boxtimes_{\mathbb{Z}_4} \mathcal{M}'}{\mathbb{Z}_4^{(1)}}. \tag{6.130}$$

The $\mathcal{M}^{(0)}$ lines are part of $\mathcal{M} \boxtimes_{\mathbb{Z}_4} \mathcal{M}'$ and then are realized on the 3d TFT via the forgetful map $\mathcal{M} \boxtimes_{\mathbb{Z}_4} \mathcal{M}' \to \widetilde{\mathcal{M}}$ that outputs the object underlying a $\mathbb{Z}_4$ equivariant object.

**Neu-Neu($\mathbb{Z}_2$)-Sandwich.** We take

$$\mathfrak{B}^{\text{sym}} = \mathfrak{B}_{\text{Neu}}^{\mathcal{M}}, \qquad \mathfrak{B}^{\text{phys}} = \mathfrak{B}_{\text{Neu}(\mathbb{Z}_2)}^{\mathcal{M}'}, \tag{6.131}$$

where $\mathcal{M}$ is an MTC with a chosen $\mathsf{Rep}(\mathbb{Z}_4)$ subcategory, and $\mathcal{M}'$ is an MTC with a chosen $\mathsf{Rep}(\mathbb{Z}_2)$ subcategory. We can express $\mathcal{M}$ as $\mathbb{Z}_4$ equivariantization of a $\mathbb{Z}_4$ crossed braided category $\mathcal{M}_0^{\mathbb{Z}_4}$

$$\mathcal{M} = \mathcal{M}_0^{\mathbb{Z}_4}/\mathbb{Z}_4^{(0)}. \tag{6.132}$$

Let the grade decomposition of $\mathcal{M}_0^{\mathbb{Z}_4}$ be

$$\mathcal{M}_0^{\mathbb{Z}_4} = \mathcal{M}_0 \oplus \mathcal{M}_1 \oplus \mathcal{M}_2 \oplus \mathcal{M}_3. \tag{6.133}$$

$\mathcal{M}_0^{\mathbb{Z}_2} := \mathcal{M}_0 \oplus \mathcal{M}_2$ combine to form a $\mathbb{Z}_2$ crossed braided extension of $\mathcal{M}_0$, equivariantizing which we obtain an MTC

$$\mathcal{M}_{0,2} := \mathcal{M}_0^{\mathbb{Z}_2}/\mathbb{Z}_2^{(0)} \tag{6.134}$$

with a $\mathsf{Rep}(\mathbb{Z}_2)$ subcategory. Hence $\mathcal{M}_{0,2}$ is $\mathbb{Z}_2$ graded.

The resulting 3d TFT is $\mathfrak{T}_{\widetilde{\mathcal{M}}}$ with

$$\widetilde{\mathcal{M}} = \frac{\mathcal{M}_{0,2} \boxtimes_{\mathbb{Z}_2} \mathcal{M}'}{\mathbb{Z}_2^{(1)}} \tag{6.135}$$

In order to describe the $\mathcal{M}^{(0)}$ symmetry, note that $\mathcal{M}_{0,2}$ is a de-equivariantization of $\mathcal{M}$ with respect to the $\mathbb{Z}_2^{(1)}$ subgroup of $\mathsf{Rep}(\mathbb{Z}_4) \subset \mathcal{M}$.

$$\mathcal{M}_{0,2} = \mathcal{M}/\mathbb{Z}_2^{(1)} \tag{6.136}$$

We thus have a map $\mathcal{M}^{(0)} \to \mathcal{M}_{0,2}^{(0)}$ outputting the underlying object of a $\mathbb{Z}_2$ equivariant object. Combining it with a similar map $\mathcal{M}_{0,2}^{(0)} \to \widetilde{\mathcal{M}}$ we obtain the realization of $\mathcal{M}^{(0)}$ symmetry on this 3d TFT.

**Neu($\mathbb{Z}_2$)-Dir-Sandwich.** We take

$$\mathfrak{B}^{\text{sym}} = \mathfrak{B}^{\mathcal{M}_0^{\mathbb{Z}_2}}_{\text{Neu}(\mathbb{Z}_2)}, \qquad \mathfrak{B}^{\text{phys}} = \mathfrak{B}_{\text{Dir}}. \tag{6.137}$$

The resulting 3d TFT is

$$\mathfrak{T}_{\mathcal{M}_0} \oplus \mathfrak{T}_{\mathcal{M}_0} \tag{6.138}$$

with the $\mathcal{M}^{(0)}$ symmetry being realized according to the functor

$$\mathcal{M}^{(0)} \to \mathcal{M}_0 \tag{6.139}$$

outputting the object underlying a $\mathbb{Z}_2$ equivariant object. The generator of $\mathbb{Z}_2^{(0)}$ symmetry exchanges the two $\mathfrak{T}_{\mathcal{M}_0}$ vacua, and the isomorphism of $D_1^{e^2}$ with identity line carries a fractional charge $1/2$ under this $\mathbb{Z}_2$, i.e. transforms by phase $\pm i$ under the $\mathbb{Z}_2$ action.

**Neu($\mathbb{Z}_2$)-Neu-Sandwich.** We take

$$\mathfrak{B}^{\text{sym}} = \mathfrak{B}^{\mathcal{M}'}_{\text{Neu}(\mathbb{Z}_2)}, \qquad \mathfrak{B}^{\text{phys}} = \mathfrak{B}^{\mathcal{M}}_{\text{Neu}}, \tag{6.140}$$

where $\mathcal{M}$ is an MTC with a chosen $\text{Rep}(\mathbb{Z}_4)$ subcategory, and $\mathcal{M}'$ is an MTC with a chosen $\text{Rep}(\mathbb{Z}_2)$ subcategory. The resulting 3d TFT is described by the MTC (6.135). The $\mathcal{M}'^{(0)}$ symmetry is realized according to the (by now obvious) maps

$$\mathcal{M}'^{(0)} \to \mathcal{M}_{0,2} \boxtimes_{\mathbb{Z}_2} \mathcal{M}' \to \widetilde{\mathcal{M}}. \tag{6.141}$$

The $\mathbb{Z}_2^{(0)}$ symmetry is realized by the $\mathbb{Z}_2$ crossed braided extension $\mathcal{M}_{0,2}^{\mathbb{Z}_2}$ whose non-trivial grade is the equivariantization

$$(\mathcal{M}_1 \oplus \mathcal{M}_3)/\mathbb{Z}_2^{(0)}. \tag{6.142}$$

**Neu($\mathbb{Z}_2$)-Neu($\mathbb{Z}_2$)-Sandwich.** We take

$$\mathfrak{B}^{\text{sym}} = \mathfrak{B}^{\mathcal{M}}_{\text{Neu}(\mathbb{Z}_2)}, \qquad \mathfrak{B}^{\text{phys}} = \mathfrak{B}^{\mathcal{M}'}_{\text{Neu}(\mathbb{Z}_2)}, \tag{6.143}$$

where $\mathcal{M}$ and $\mathcal{M}'$ are MTCs with chosen $\text{Rep}(\mathbb{Z}_2)$ subcategories. The resulting 3d TFT is

$$\mathfrak{T}_{\widetilde{\mathcal{M}}} \oplus \mathfrak{T}_{\widetilde{\mathcal{M}}}, \tag{6.144}$$

where

$$\widetilde{\mathcal{M}} = \frac{\mathcal{M} \boxtimes_{\mathbb{Z}_2} \mathcal{M}'}{\mathbb{Z}_2^{(1)}}. \tag{6.145}$$

The $\mathcal{M}^{(0)}$ symmetry is realized by maps

$$\mathcal{M}^{(0)} \to \mathcal{M} \boxtimes_{\mathbb{Z}_2} \mathcal{M}' \to \widetilde{\mathcal{M}}. \tag{6.146}$$

The $\mathbb{Z}_2^{(0)}$ symmetry exchanges the two $\mathfrak{T}_{\widetilde{\mathcal{M}}}$ vacua such that the square of the $\mathbb{Z}_2$ symmetry generator is fractionalized on $D_1^{e^2} \in \mathcal{M}^{(0)}$.

**Acknowledgmenets**

We thank T. Decoppet, S. Huang, K. Inamura, F. Moosavian, H. Moradi, M. Yu for discussions. We thank the author of [75] for coordinating submission on related results. LB is funded as a Royal Society University Research Fellow through grant URF\R1\231467. The work of SSN and AW is supported by the UKRI Frontier Research Grant, underwriting the ERC Advanced Grant "Generalized Symmetries in Quantum Field Theory and Quantum Gravity". The work of AT is funded by Villum Fonden Grant no. VIL60714.

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
