# Peer review of "Gapped Phases in (2+1)d with Non-Invertible Symmetries: Part I"

_SciPost Physics_

## Round 2 · Referee Report · Anonymous (Referee 2) · 2025-3-19

Strengths
2-The treatment is very explicit.
3-Many concrete examples are provided.
Weaknesses
2-The content of section 3 is by now quite standard, both in the Math and Physics literatures.
3-The preprint is fairly long.
Report
Requested changes
0-There is a typo in the first line of the second paragraph of page 5.
1-Footnote 5 is incorrect.
2-The question of which boundary conditions admit a topological interface admit a well-known mathematical description in terms of so-called Witt equivalence. This ought to be mentioned. Please see the article ''On the structure of the Witt group of braided fusion categories'' by Davydov-Nikshych-Ostrik. In particular, the question of which boundary conditions are not gauge related to the Dirichlet boundary is intimately related to fact that the Witt group associated to $\mathrm{Rep}(\mathbb{Z}_2)$ is non-trivial.
3-The notation $\boxtimes_{\mathbb{Z}_2}$ is potentially confusing: I find that it clashes somewhat with the relative tensor product. I would recommend modifying it, or at least introducing it more carefully.
4-The theta construction on page 45 is missing some data. Namely, in the presence of an anomaly $\tau$, there is no \textit{canonical} choice of diagonal action. (Unlike the case when the anomaly is trivial.)
4.5-Relatedly, for ease of comparison with the other treatments of similar questions that have appeared, it may be worth emphasizing that even when the anomaly $\tau$ is trivial, there is still a choice of anomaly $\omega$ for the $G$-symmetric 3d TFT. This is brought up later, but I think the general reader would appreciate this point being discussed explicitly near the beginning of section 5.
5-I believe that the categories on page 53 have also been extensively analyzed by Delcamp-Tiwari and Decoppet-Yu.
6-I fail to understand why Tables 1 and 2 are not symmetric along the diagonal. This should be explained.
7-On page 70, the class $\omega$ is already determined by $\widetilde{\mathcal{M}}$. It can therefore not be chosen.
8-Equation (6.96) is not quite correct. I suppose that the authors are trying to say that the left hand-side is some kind of extension?
Recommendation
Ask for minor revision
We thank the referee for their positive comments, helpful comments and pointing out various typos. 1. We thank the referee for pointing out the typo. 2. We thank the referee for pointing that out. We have clarified the statement. 3. We have added a definition and reference regarding this point. 4. We clarified the notation, which was adopted from \cite{Cui:2021lyi}. 5. Thanks, we have now added these references. 6. These tables are not symmetric about the diagonal as exchanging the rows and columns exchanges the symmetry and physical boundary of the SymTFT. While the resulting 3d TFT remains invariant, the phase changes as the reference symmetry changes. 7. We thank the referee for this important observation. We agree that the associators are part of the data of $\widetilde{\mathcal M}$ and are determined internally. These can however typically be modified by stacking an SPT. We have now clarified this part. 8. We agree. Eq 6.96 is meant to be schematic. There is a mixed anomaly between the 0-form symmetry and the non-minimal 1-form symmetry which has been explained in words in the preceding paragraph.

---

## Round 2 · Referee Report · Anonymous (Referee 1) · 2025-3-19

Weaknesses
Report
The title claims that the symmetries can be non-invertible, however, in the details and examples only invertible symmetries are studied. In the referee's opinion, the semi-categorical, semi-field-theoretical approach taken in this work is far from sufficient for dealing with genuine non-invertible higher symmetries. Neither the examples with invertible higher symmetries nor the use of SymTFT is novel to the community. Therefore, this submission hardly meets the criteria of SciPost Physics but does meet those of SciPost Physics Core, where it could be published.
Requested changes
- Footnote 5 on page 6 is incorrect. It is $2\mathrm{Vec}_G^\tau$ which should be replaced by a fermionic strongly fusion 2-category (a group extension of 2$\mathrm{SVec}$) while $\mathcal M$ remains to be non-degenerate (See Theorem 4.1.6. in arXiv:2211.04917).
- Section 3.5 the "only if" direction under eq. (3.43) is not logically sound. Why it is not possible to gauge something both in $\mathfrak{B}_\mathrm{Dir}$ and $\mathfrak{T}$ to obtain $\mathfrak{B}_\mathrm{Dir}$? It may be proved based on a more general result (see Theorem 3.29. and Corollary 3.30. in arXiv:2312.15958) but for now the argument is not strong enough.
Recommendation
Accept in alternative Journal (see Report)
We will reply to each of the points in turn: 1. We thank the referee for pointing that out. We have clarified the statement. 2. Equation (3.43) and the statement below it are correct as is. A gapped boundary condition has a topological interface to $\mathfrak B_{\rm Dir}$, if it is obtainable by gauging an algebra object in $2\rm Vec_{G}$. Algebras in $2\rm Vec_{G}$ are classified by $H$-graded fusion categories where $H<G$. Let us denote such an algebra as $A=\oplus_{h}A_{h}$. Then in equation 3.43, $\mathfrak T=\mathcal Z(A_{\rm id})$, as claimed.
About the comment ``no example with non-invertible symmetry'': we would like to reiterate that all fusion 2-categories of bosonic type are gauge related (or Morita equivalent) to $2\mathrm{Vec}^\tau_{G}$, upto decoupled invertible factors which are (condensation completions of) MTCs. In particular this means that the SymTFT is the same for all symmetries of this type to the SymTFT for $2\mathrm{Vec}_{G}^\tau$, which is a Dijkgraaf Witten theory based on group $G$ and twist $\tau$. Therefore even though $G$ is always a (invertible) group, our construction covers all possible symmetries described by fusion 2-categories of bosonic type, including both invertible and non-invertibles ones. In the present manuscript we restricted to the case of $G$ being an Abelian group which was generalized to non-Abelian groups in part-II of this series (https://arxiv.org/abs/2502.20440). While minimal gaugings of Abelian groups always give invertible symmetry categories (upto condensations), non-minimal gaugings embody an infinte family of generically non-invertible symmetry structures.

Author: Jingxiang Wu on 2025-06-25 [id 5596]
(in reply to Report 3 on 2025-05-09)We thank the referee for the points 1-5 and we have clarified/fixed the items in those points. Regarding the last point:

---

## Round 2 · Referee Report · Anonymous (Referee 3) · 2025-5-9

Strengths
Weaknesses
-
Several statements contain mathematical inaccuracies that could mislead readers.
-
The treatment of gapped boundaries in the 4d TQFT is largely physical in tone. Some notations clash with standard notations in math and mathematical physics literature and might cause confusion.
4.Several arguments and examples recur in slightly different guises. Consolidating overlapping material will sharpen the narrative and reduce the overall length without sacrificing clarity.
Report
Overall the manuscript is clearly written and meets SciPost's publication criteria. However there are several mathematical inaccuracies and other issues that should be addressed before publication, as detailed in the requested changes below.
Requested changes
-
On page 5, there is a typo “the existence of s that”.
-
On page 6, the authors discuss the classification of fusion 2-categories up to Morita equivalence. There are two locations where the statements are incorrect. 2.1 “In short the main result is that all fusion 2-categories are gauge related to $2\text{Vec}_G^\tau$”. The authors might mean “bosonic fusion 2-categories” or “fusion 2-categories of bosonic type”.
2.2 “A similar statement for the fusion 2-categories of fermionic type also holds by working with $\text{SVec}$ and allowing $M$ to have $\mathcal{Z}_2=\text{SVec}$”. The phrase “working with $\text{SVec}$” is ambiguous. It is also not true that all fusion 2-categories of fermionic type is Morita equivalent to $2\text{Vec}_G^\tau \boxtimes \Sigma M$ for a slightly degenerate $M$, as that would imply any fermionic fusion 2-category $\mathcal{C}$ satisfies $\mathcal{Z}_2\Omega \mathcal{C}=\text{SVec}$. The general case is that$\mathcal{Z}_2\Omega C$ is super-Tannakian. The correct variant of (2.1) for the fermionic case is that any fermionic fusion 2-category is Morita equivalent to $\mathcal{C}\boxtimes \Sigma M$ where $\mathcal{C}$ is a graded extension of $2\text{SVec}$, and $M$ is an MTC. This is 2211.04917 theorem 4.1.6.
-
On page 7. The authors describe the gapped symmetry boundary as being specified by bulk operators that end on the boundary, and mention that in 1+1D the analog are Lagrangian algebras. Notably condensable and Lagrangian algebras in higher dimensional TFTs had been defined and studied in 2105.15167, 2307.02843, 2403.07813, and provide a unified mathematical description for gapped boundaries of TFTs in any dimensions. It seems odd to say the 1+1d analog are Lagrangian algebras, as it suggests the 2+1d case is described by something else.
-
On page 46. The authors give a classification of gapped boundaries of 4d DW theories. This classification is incorrect. If the 4d DW theory has nontrivial twist, then a gapped boundary is given by a subgroup H<G, together with a $\textit{twisted}$ H-crossed extension of an MTC. When the $\mathcal{H}^4$ obstruction is nontrivial, there is no (untwisted)H-crossed extension.
-
On page 70. Eq.(6.87), the LHS should be $\mathcal{Z}2(\widetilde{M)$}}.
-
Section 6.2 seems redundant given the more general discussion in section 5.1. E.g. (6.87)(when corrected) is simply (5.10) when H=A and Eq. (6.83) is simply (5.9) when H=A, etc.
Recommendation
Ask for minor revision

---

## Editorial Decision

unknown